# FANCD2–FANCI surveys DNA and recognizes double- to single-stranded junctions

Pablo Alcón[1], Artur P. Kaczmarczyk[2,3], Korak Kumar Ray[2,3], Themistoklis Liolios[4], Guillaume Guilbaud[1], Tamara Sijacki[1], Yichao Shen[1], Stephen H. McLaughlin[1], Julian E. Sale[1], Puck Knipscheer[4], David S. Rueda[2,3✉] & Lori A. Passmore[1✉]

DNA crosslinks block DNA replication and are repaired by the Fanconi anaemia pathway. The FANCD2–FANCI (D2–I) protein complex is central to this process as it initiates repair by coordinating DNA incisions around the lesion[1]. However, D2–I is also known to have a more general role in DNA repair and in protecting stalled replication forks from unscheduled degradation[2–4]. At present, it is unclear how DNA crosslinks are recognized and how D2–I functions in replication fork protection. Here, using single-molecule imaging, we show that D2–I is a sliding clamp that binds to and diffuses on double-stranded DNA. Notably, sliding D2–I stalls on encountering single-stranded–double-stranded (ss–ds) DNA junctions, structures that are generated when replication forks stall at DNA lesions[5]. Using cryogenic electron microscopy, we determined structures of D2–I on DNA that show that stalled D2–I makes specific interactions with the ss–dsDNA junction that are distinct from those made by sliding D2–I. Thus, D2–I surveys dsDNA and, when it reaches an ssDNA gap, it specifically clamps onto ss–dsDNA junctions. Because ss–dsDNA junctions are found at stalled replication forks, D2–I can identify sites of DNA damage. Therefore, our data provide a unified molecular mechanism that reconciles the roles of D2–I in the recognition and protection of stalled replication forks in several DNA repair pathways.

DNA interstrand crosslinks (ICLs) are toxic lesions that block DNA replication and transcription, and are detected and repaired by the Fanconi anaemia pathway. The FANCD2–FANCI (D2–I) complex is key to Fanconi anaemia DNA repair as it binds dsDNA and is required for recruitment of nucleases that excise crosslinks[1]. D2–I is also implicated more broadly in human immunodeficiency virus integration, clustered regularly interspaced short palindromic repeats (CRISPR)-mediated DNA repair and general DNA replication stress, where it plays a role in stabilizing stalled replication forks or other DNA repair intermediates[3,6–8]. Monoubiquitination of D2–I is required for DNA crosslink repair, but the role of monoubiquitination in replication fork protection is less clear[2].

Recent in vitro reconstitution and structural studies have revealed three states of the D2–I clamp: open, closed and locked[9–12]. Free D2–I is in an open conformation in which the principal DNA-binding site in FANCI is accessible. Both DNA binding and phosphorylation of FANCI by the DNA damage-activated ataxia telangiectasia and Rad3-related (ATR) kinase promote closure of D2–I. Closed D2–I is clamped around DNA. A pronounced kink in the D2–I-bound DNA[9,10,12] led to the suggestion that closed D2–I might sense changes in the double helix, for example because of an ICL. The monoubiquitination sites on D2–I are located at the dimerization interface and are exposed in the closed (but not in the open) state. The closed state enables efficient D2–I monoubiquitination to activate Fanconi

anaemia DNA repair. As ubiquitin is wedged into the dimerization interface, ubiquitinated D2–I is locked on DNA and is not able to transition back to the open state and release DNA until ubiquitin is removed.

It has been assumed that D2–I directly recognizes DNA crosslinks. However, D2–I has no clear binding preference for any particular DNA sequence or structure tested so far, and any dsDNA stimulates ubiquitination in vitro[13–15]. Moreover, in cryogenic electron microscopy (cryo-EM) structures, a crosslink was not visible in D2–I clamped on DNA[12]. Finally, a direct function in recognizing crosslinks cannot explain D2–I's more general role in replication stress. Thus, the mechanism of how DNA damage is specifically recognized and the molecular roles of D2–I have remained a mystery, both in the Fanconi anaemia pathway and in other types of DNA repair.

Given the lack of evidence for direct recognition of DNA crosslinks, we suggested that D2–I recognizes an alternative DNA structure that would unify its roles in crosslink repair and replication fork protection. Here, we test this using single-molecule imaging and cryo-EM to directly visualize vertebrate D2–I interacting with DNA. We show that D2–I is a proficient DNA sliding clamp that, notably, stalls at and specifically recognizes ds to ss junctions, a role that is required for efficient repair of DNA crosslinks. Our data suggest that D2–I identifies and protects the structures surrounding stalled replication forks, reconciling its multiple roles in DNA repair.

[1]MRC Laboratory of Molecular Biology, Cambridge, UK. [2]Department of Infectious Disease, Faculty of Medicine, Imperial College London, London, UK. [3]MRC Laboratory of Medical Sciences, London, UK. [4]Oncode Institute, Hubrecht Institute–KNAW and University Medical Center Utrecht, Utrecht, The Netherlands. ✉e-mail: david.rueda@imperial.ac.uk; passmore@mrc-lmb.cam.ac.uk

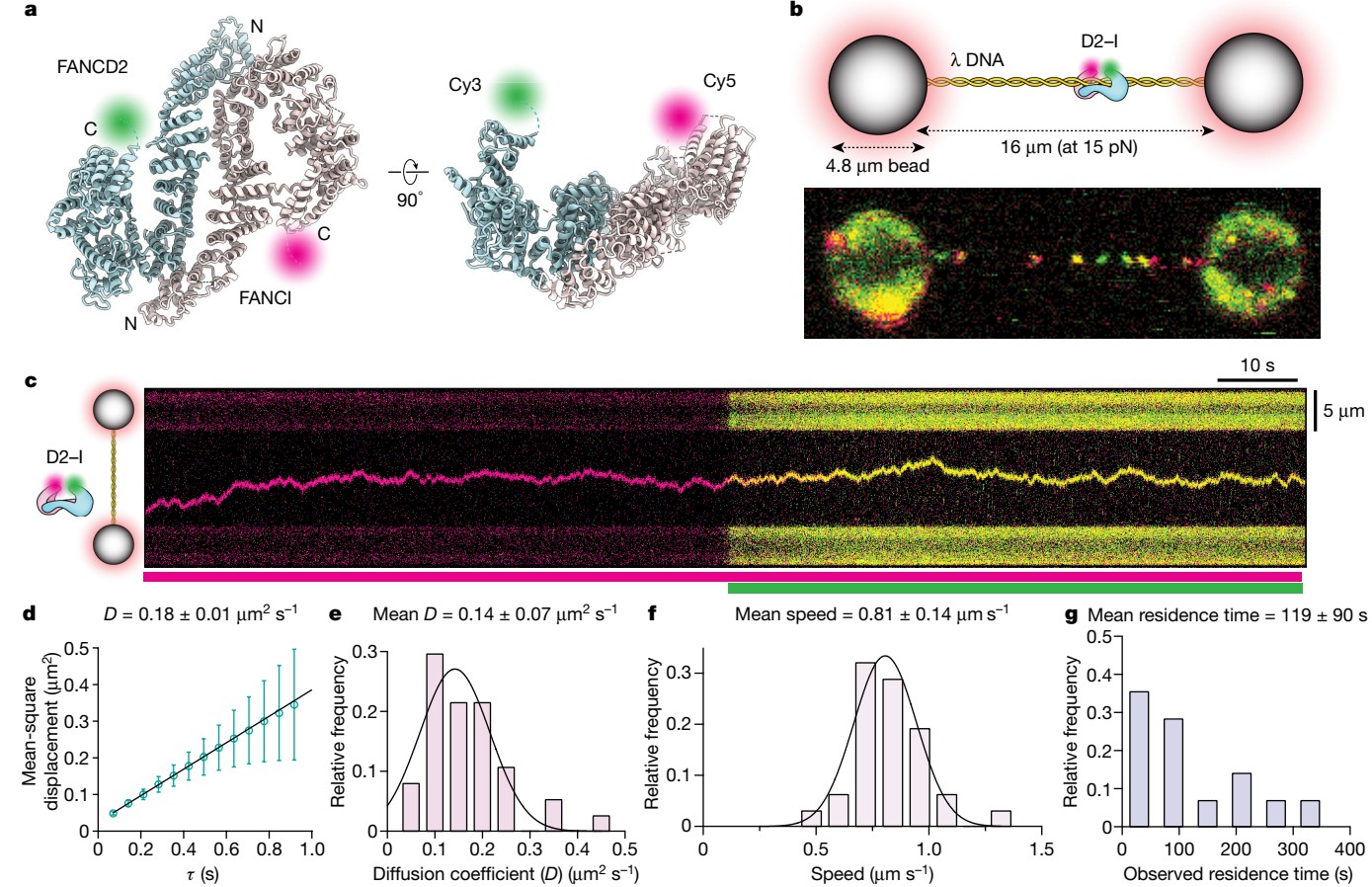

**Fig. 1 | Single-molecule imaging of the D2–I clamp sliding on DNA.**
**a**, Orthogonal views of the model of the open D2–I clamp (PDB 6TNG) indicating the positions of the fluorescent dyes. C, C terminus; N, N terminus. **b**, Top, schematic illustration of the single-molecule experimental set-up. Bottom, confocal scan (merged green and red channels) showing labelled D2–I bound to λ DNA. **c**, Representative kymograph showing one-dimensional random diffusion by D2–I. The coloured bars below indicate the excitation laser used (magenta, Cy5; green, Cy3). A gallery of representative kymographs is shown in Supplementary Fig. 2. **d**, Mean square displacement (MSD) of the single

trajectory extracted from the kymograph in **c**, yielding a diffusion coefficient ($D$) of $0.18 \pm 0.01\ \mu m^2\ s^{-1}$. Error bars represent variance of MSD at given time intervals. **e**, Distribution of calculated $D$ of individual D2–I particles and Gaussian fit, yielding a mean $D$ of $0.14 \pm 0.07\ \mu m^2\ s^{-1}$ ($n = 37$). **f**, Distribution of speed of individual D2–I molecules on λ DNA yields a mean speed of $0.81 \pm 0.14\ \mu m\ s^{-1}$ ($n = 31$). **g**, Distribution of observed residence times of trajectories of D2–I on λ DNA ($n = 28$). All confidence intervals are standard deviation of the fit. For **e**–**g**, data points were divided into bins and the bars of the histograms are centred at the midpoint of each bin.

## D2–I slides on DNA

To examine DNA binding by D2–I, we directly visualized it at single-molecule resolution and in real time. First, we prepared fluorescently labelled *Gallus gallus* D2–I complexes (Fig. 1a and Extended Data Fig. 1a–d). We then used optical tweezers coupled with confocal microscopy and microfluidics[16–18] to trap the 48.5 kb λ bacteriophage DNA in a dumbbell-like configuration. The tethered DNA molecule was incubated in a microfluidic channel containing D2–I and subsequently imaged in the protein-free channel (Extended Data Fig. 1e). This showed distinct fluorescent D2–I puncta that moved along the λ DNA over time (Fig. 1b and Supplementary Video 1). These data therefore suggest that D2–I slides on DNA, consistent with the previous observation that D2–I has a slower dissociation rate from circular or end-blocked DNAs than from linear DNA[11,12].

To resolve the dynamic movement of D2–I on DNA, we recorded kymographs along the DNA axis and observed that individual D2–I complexes move bidirectionally on DNA (Fig. 1c). The presence of both FANCD2 and FANCI were confirmed through the detection of both green (FANCD2-Cy3) and red (FANCI-Cy5; shown as magenta in the figures) fluorophores within the same particle (Fig. 1c). Photobleaching was observed as a single-step event for each fluorophore, consistent with the

analysed trajectories comprising single D2–I heterodimers (Extended Data Fig. 1f). Analysis of these tracked pixel positions shows that D2–I moves along the DNA duplex by means of thermal energy-driven random one-dimensional diffusion (Extended Data Fig. 1g).

D2–I slides on DNA with a mean diffusion coefficient (mean $D$) of $0.14 \pm 0.07\ \mu m^2\ s^{-1}$ (which corresponds to $1.2 \times 10^6\ bp^2\ s^{-1}$), and a fast apparent mean speed ($0.81 \pm 0.14\ \mu m\ s^{-1}$) (Fig. 1d–f). The diffusion coefficient of D2–I on DNA is slower than that of proliferating cell nuclear antigen (PCNA)[19] and XRCC4-XLF[20], but faster than the Cdc45-MCM-GINS (CMG) replicative helicase[21–23]. It matches the diffusion coefficient for a protein the size of D2–I undergoing both rotational and translational diffusion ($0.12\ \mu m^2\ s^{-1}$)[19], suggesting that it slides along the helical pitch of dsDNA. D2–I has a long mean residence time on DNA ($119 \pm 90$ s; Fig. 1g), with individual molecules having residence times of up to 10 min, which were limited by photobleaching (Extended Data Fig. 1h). D2–I was often engaged in long-range scanning, traversing across almost the entire length of λ DNA (about 16 μm or 48.5 kbp) (Fig. 1c).

The initial loading of D2–I on DNA is influenced by DNA tension, as it loads less efficiently onto DNA stretched at higher forces (greater than 15 pN) than it does onto DNA held at lower forces (less than 5 pN) (Extended Data Fig. 1i). In addition, we found that D2–I loading onto

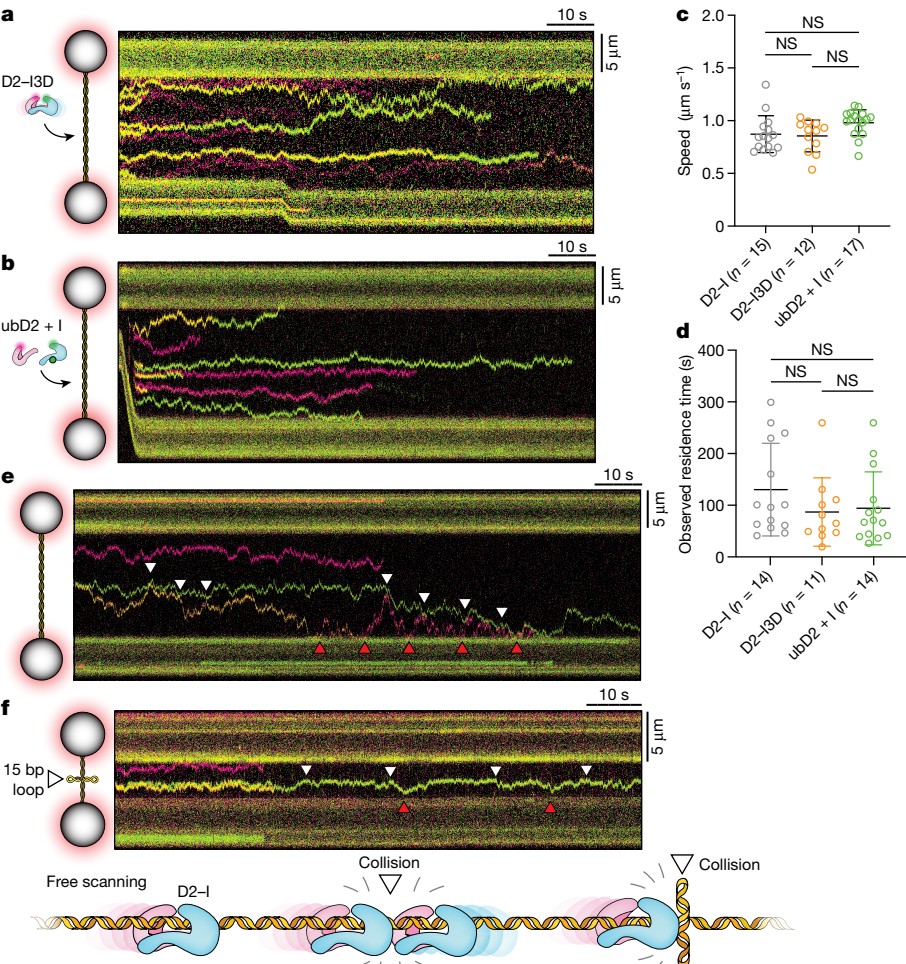

**Fig. 2 | Characterization of D2–I sliding on DNA. a,b,** Schematic representation of experimental set-up (left) and representative kymographs (right) of phosphomimetic D2–I3D (**a**) and monoubiquitinated ubD2–I (**b**) complexes on λ DNA, acquired at 15 pN. **c,d,** Comparison of the distributions of the speeds (**c**) and observed residence times (**d**) of D2–I, D2–I3D and ubD2–I. Individual values, mean, standard deviation and number of traces analysed in the comparison (*n*) are shown. Unpaired *t*-test pairwise comparisons showed no statistically significant differences (NS) between the three complexes. **e,** Kymograph of D2–I on λ DNA at 15 pN. Molecular collisions with other D2–I molecules are marked with white arrows; collisions between D2–I and the trapped beads are marked with red arrows. **f,** Kymograph showing D2–I on a 15 kb DNA that contains a four-way junction at a central position. Molecular collisions between D2–I and the 15 bp DNA loop or D2–I and the beads are marked with white and red arrows, respectively. These kymographs are representative of observations from ten independent experiments. A schematic representation of encounters between individual sliding D2–I complexes and D2–I with a DNA loop is shown at the bottom.

DNA is more efficient at lower salt concentrations (Extended Data Fig. 2a,b). This indicates that DNA binding by D2–I involves predominantly ionic interactions, in agreement with its sequence-non-specific nature. We observed only small changes in the diffusion coefficient over different ionic strengths, suggesting that D2–I maintains continuous contact with DNA to diffuse by sliding, rather than by a hopping mechanism (Extended Data Fig. 2c).

FANCI alone binds DNA with comparable affinity to the D2–I heterodimer, whereas FANCD2 can form a closed homodimer that does not bind DNA[9]. We found that FANCI binds DNA at a lower density, has significantly shorter observed residence times and slides on DNA with a wider range of diffusion coefficients than D2–I (Extended Data Fig. 2d–f). By contrast, FANCD2 showed very inefficient DNA binding (Extended Data Fig. 2d). These observations agree with the structures of DNA-bound FANCI and D2–I, where most protein–DNA contacts are located on the concave surface of FANCI, with more limited contacts to FANCD2 (refs. 9–12,24,25).

Previous work proposed that the open, closed and locked states could have altered sliding properties that facilitate sensing or signalling of the DNA lesion[12]. We observed anti-correlated green and red fluorescence in single-molecule traces (Extended Data Fig. 3a–f). This is indicative of single-molecule fluorescence resonance energy transfer (FRET) and reflects changes in the distance between the two fluorescent dyes. These data therefore suggest that D2–I is in an equilibrium between open (low-FRET) and closed (high-FRET) conformations, not only in solution[9,12,24] but also while it is diffusing on DNA.

To directly compare the sliding properties of D2–I in different states, we analysed fluorescently labelled phosphomimetic D2–I (D2–I3D), which is more likely to occupy the closed state than unmodified D2–I[11], and the locked monoubiquitinated complex (ubD2–I) (Extended Data Fig. 3g). Both 'closed' D2–I3D and 'locked' ubD2–I complexes slide on λ DNA with speeds and observed residence times comparable to those of unmodified D2–I (Fig. 2a–d). Thus, phosphorylation and monoubiquitination do not seem to play major roles in one-dimensional DNA scanning by D2–I in vitro.

## D2–I bounces off non-specific obstacles

Next, we tested D2–I behaviour in the presence of obstacles to understand how it might respond to different situations present after DNA

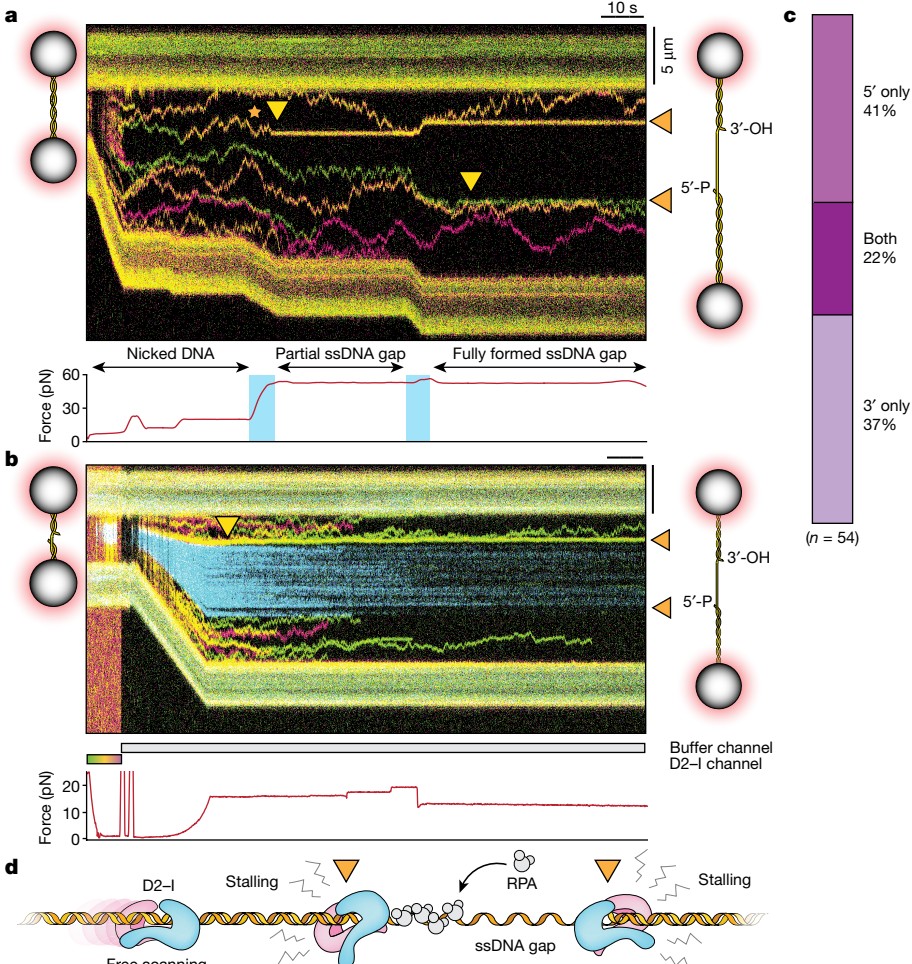

**Fig. 3 | Sliding D2–I complexes stall at ss–dsDNA junctions. a**, Kymograph of D2–I bound to λ DNA with a site-defined ssDNA gap. At the start of the kymograph, the λ DNA is nicked at two specific sites. Force is applied to generate a 17.8 knt ssDNA gap resulting from a two-step dissociation of the ssDNA fragment between the two nicks. Force is plotted under the kymograph and shaded in blue at stretching points. Schematic representations of the DNA before and after applying force are shown on the left and right sides of the kymograph, respectively. Orange arrows, ss–dsDNA junctions; yellow arrows, static binding events at ss–dsDNA junctions; orange star, a mobile trace that becomes static on ssDNA gap generation. **b**, Kymograph of D2–I on gapped λ

DNA where the ssDNA gap has been coated by blue eGFP-labelled RPA. The yellow arrow marks the site of static D2–I at the ss–dsDNA junction. Coloured bars under the kymograph indicate the channel position (grey, buffer channel; green–yellow–red, D2–I channel). Force measurement is plotted under the kymograph. These are representative of 80 observations in independent experiments. **c**, Quantification of kymographs containing stalled D2–I complexes at 3′-ended, 5′-ended, or both ends of ss–dsDNA junctions. **d**, Schematic representation of D2–I sliding on dsDNA and stalling at the RPA-bound ss–dsDNA junctions (orange arrows) that flank the ssDNA gap.

damage. When the sliding complexes reached the ends of DNA in our single-molecule experiments, they bounced off the streptavidin-coated polystyrene beads and continued diffusing along the DNA in the opposite direction (Fig. 2e). Owing to random photobleaching events, a given D2–I particle may have green fluorescence, red fluorescence or both. This allowed us to differentiate complexes from each other. We found that converging D2–I molecules did not bypass one another, stall, dissociate from DNA or aggregate with one another. Instead, colliding D2–I complexes reversed direction and continued sliding (Fig. 2e). The absence of bypass events is consistent with D2–I encircling the dsDNA helix, as seen in cryo-EM structures[9,10,12].

To further test the behaviour of sliding D2–I molecules on encountering a physical roadblock, we used a linear dsDNA that contained a central four-way junction comprising a double 15 nt loop, similar to a Holliday junction[26] (Fig. 2f). All observed D2–I molecules showed efficient sliding that was confined to one half of the Holliday junction DNA, suggesting that the DNA loop poses a physical roadblock to sliding D2–I. Molecules that reached the central DNA loop bounced off and did not bypass, dissociate or stall (Extended Data Fig. 4a), similar to the

protein–protein molecular collisions, suggesting that a non-specific barrier constrains DNA scanning by D2–I.

## D2–I stalls at ss–dsDNA junctions

During S phase, replisomes converge and stall at DNA crosslinks. This results in unloading of the CMG helicase, followed by removal of the DNA lesion (unhooking) and repair[1]. D2–I accumulates at stalled replication forks during crosslink repair[1] and replication stress[2,4], and FANCD2 prevents resection of nascent DNA and stabilizes stalled replication forks[2]. Thus, we reasoned that D2–I may recognize a DNA structure commonly found in these processes. ssDNA gaps are present on both the leading and lagging strands at stalled replication forks. We therefore tested how D2–I responds to ssDNA gaps[27].

We used the RNA-guided CRISPR–Cas9 system to generate site-specific nicks that, on force extension, result in a defined 17.8 knt ssDNA gap in the ds λ DNA[27,28]. D2–I readily binds and slides along the dsDNA regions, but not in the ssDNA gaps (Fig. 3a). This is consistent with D2–I

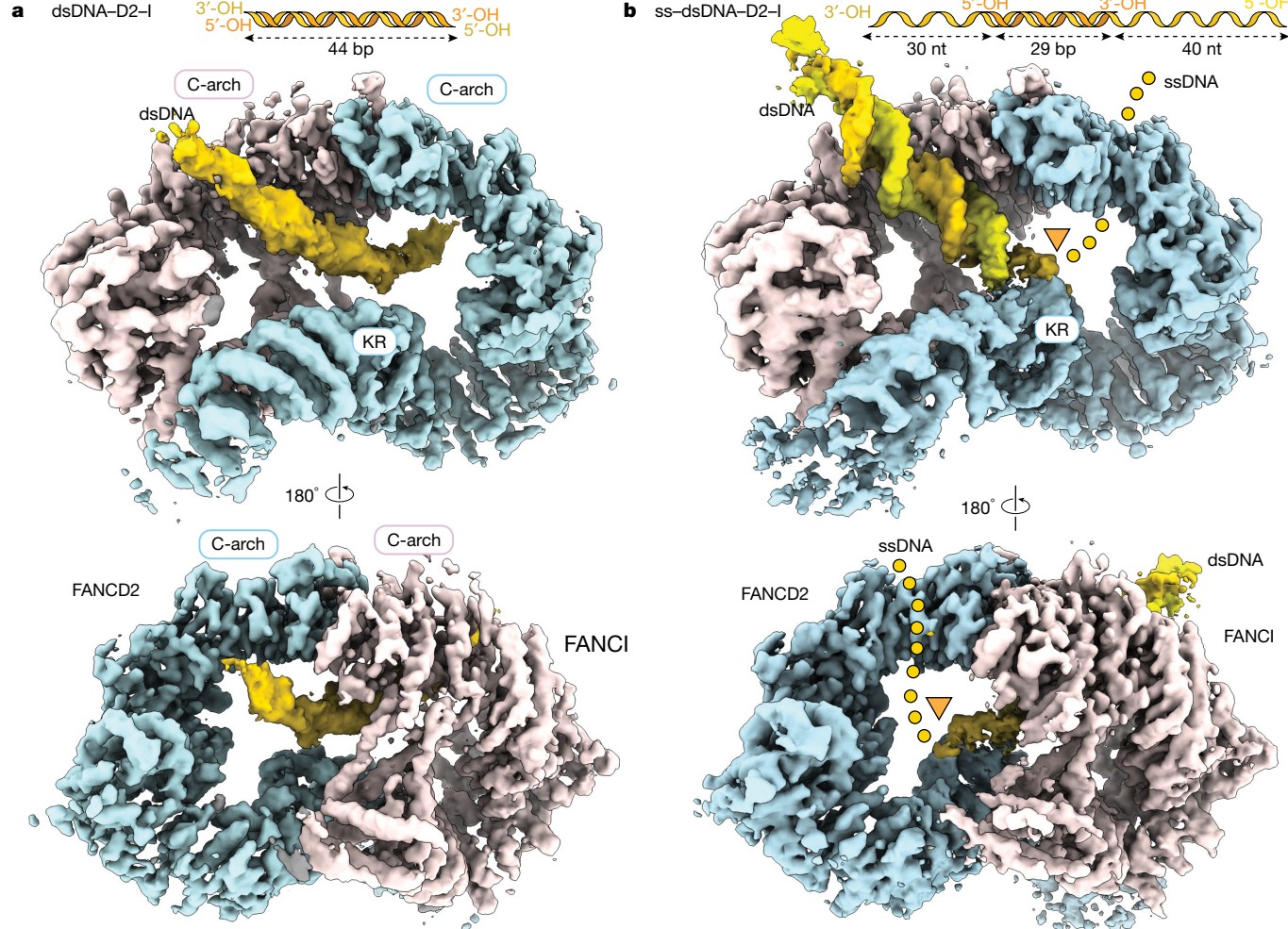

**Fig. 4 | Structures of D2–I with dsDNA or an ss–dsDNA junction. a**, Composite cryo-EM map of dsDNA–D2–I. The unsharpened DNA density is segmented and coloured in gold. The sharpened map of D2–I is segmented into FANCD2 (light blue) and FANCI (pink). A schematic of the dsDNA used is shown on top. The C-terminal arches (C-arch) and the Lys- and Arg-rich helix (KR) are labelled.

**b**, Sharpened cryo-EM map of ss–dsDNA–D2–I. Segmented and coloured as in **a**. The ss–dsDNA junction site is marked by an orange triangle and the putative path of ssDNA, determined from the unsharpened DNA density (Extended Data Fig. 8b), is marked with a dotted line.

interacting with dsDNA, but not ssDNA, in gel shift assays (Extended Data Fig. 4b). Notably, we observed static D2–I molecules close to the borders of the ssDNA gaps, suggesting that the complex stalls at ss–dsDNA junctions (Fig. 3a).

To more precisely map the location of the ss–dsDNA junction and how this relates to the static D2–I traces, we used the ssDNA-binding replication protein A (RPA) labelled with enhanced green fluorescent protein (eGFP)[28]. We incubated DNA containing a single-strand gap with RPA, and then with D2–I. Static D2–I particles were located at the edges of the RPA-labelled ssDNA (Fig. 3b), showing that D2–I stalls at ss–dsDNA junctions. By contrast, neither FANCD2 nor FANCI alone stall at ss–dsDNA junctions (Extended Data Fig. 5).

We observed two different types of D2–I stall (Extended Data Fig. 6a–d and Supplementary Fig. 3): approximately 50% of stalled D2–I molecules remain static at the ss–dsDNA junction (persistent stalls) with a lifetime of $52 \pm 3$ s (probably limited by photobleaching). The remaining stalled D2–I complexes are static at the ss–dsDNA junction for a shorter time (transient stalls), with a lifetime of $7.3 \pm 0.1$ s. Quantitation of stalled complexes in the single-molecule data and measurement of DNA-binding kinetics on different DNA substrates show that D2–I does not have a strong preference for the 5′ or 3′ end of the junction (Fig. 3c and Extended Data Fig. 6e,f).

Together, these data show that D2–I slides freely on dsDNA but, notably, stalls at ss–dsDNA junctions (Fig. 3d). Because ss–dsDNA junctions are generated at stalled replication forks, they may represent a universal binding site for D2–I in crosslink repair and replication stress, unifying the recognition of DNA structures by D2–I across several pathways.

## Structure of a sliding D2–I on DNA

To understand the molecular basis for how D2–I stalls at ss–dsDNA junctions, we determined cryo-EM structures of D2–I bound to a linear dsDNA duplex, and compared this with D2–I bound to an ss–dsDNA junction (Fig. 4, Extended Data Figs. 7 and 8, Extended Data Table 1 and Supplementary Fig. 4). First, we obtained a 3.7 Å resolution structure of D2–I bound to a 44 bp dsDNA (Fig. 4a). In this structure, the D2–I clamp is fully closed around DNA. Similarly to previously reported closed structures of monoubiquitinated and phosphomimetic D2–I[9–12], the C termini of FANCD2 and FANCI are well-ordered, and stabilized by contacts that form the clamp and embrace DNA. The N-terminal regions of both FANCD2 and FANCI are more flexible (Extended Data Fig. 7c).

We imaged human D2–I and found that it also closes onto 44 bp dsDNA (Extended Data Fig. 7e,f). Previous structures of unmodified D2–I bound to DNA showed a range of conformations from fully open

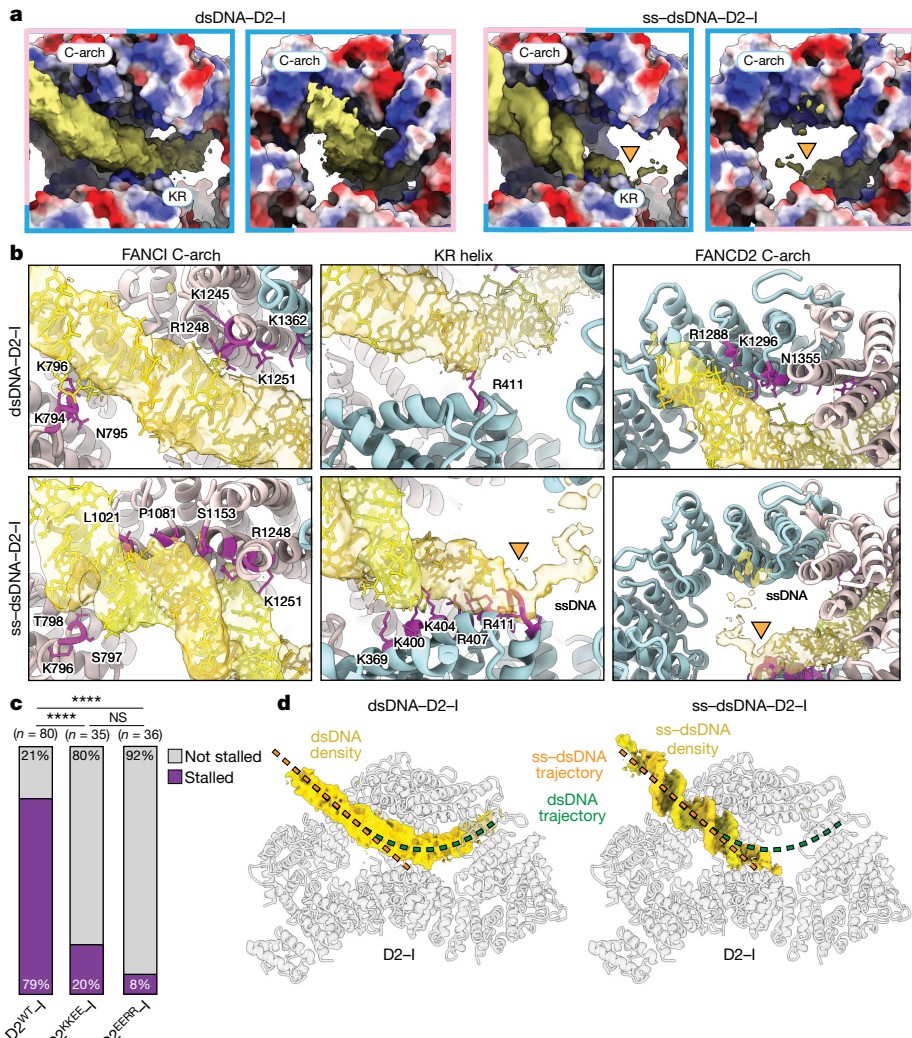

**Fig. 5 | Details of interaction of D2–I with dsDNA or ss–dsDNA junction.**
**a**, Electrostatic potential of D2–I shown in surface representation from −10 (red) to +10 (blue) kT; regions with a neutral net charge are shown in white. The C-arches of FANCD2 (label outlined in light blue) and FANCI (label outlined in light pink), and the KR helix on FANCD2 are labelled. **b**, The interaction of D2–I with dsDNA (top) and ss–dsDNA (bottom) differs. FANCD2 is coloured in light blue, FANCI in light pink and DNA in yellow and gold. The residues in D2–I that are within a 4.5 Å distance of the DNA model are shown in purple, and their side chains are shown. The ss–dsDNA junction site is marked by an orange triangle. **c**, Quantification of the number of kymographs showing stalled D2–I traces at ss–dsDNA junctions. The results of two-sided pairwise comparisons using Fisher's exact test are shown (****$P < 0.0001$; not significant (NS) $P = 0.5059$). $n$, number of independent kymographs. **d**, DNA trajectories in D2–I are different for dsDNA and ss–dsDNA. Segmented DNA densities on transparent D2–I model to show the trajectories followed by dsDNA (left) and ss–dsDNA (right). A line tracing the central axis of the duplex for each substate is shown for comparison (dsDNA in green and ss–dsDNA in orange).

to fully closed, but they were predominantly in the open state[9–12]. However, most of the previous cryo-EM structures were determined using branched (nonlinear) dsDNA substrates that contained only short regions of continuous duplexes. Therefore, it is likely that the longer 44 bp dsDNA used here allows efficient formation of the closed state. By contrast, the dsDNA segments (≤29 bp) in previous samples also contained branched structures that did not allow D2–I to close efficiently, resulting in open or partially closed conformations in cryo-EM and reflecting the equilibrium between these states in solution[9–12].

The 44 bp dsDNA comes into close contact with two interaction sites on D2–I (Fig. 5a,b). First, the dsDNA contacts a basic arch in the C-terminal domain (C-arch) of FANCI that was previously described as the primary DNA binding site[12]. This is followed by a substantial kink in the dsDNA before it contacts a second basic arch within a C-terminal region of FANCD2.

The DNA is poorly ordered, probably because it represents an average of different binding positions within the D2–I clamp (Fig. 4a and

Extended Data Fig. 8a). In agreement with this, signal subtraction and focused classification of the dsDNA region showed that the duplex adopts several positions within the closed D2–I clamp (Extended Data Fig. 8c). Given the diversity of dsDNA positions, this structure probably represents sliding D2–I. We propose that sliding is mediated by the electrostatic, sequence-non-specific nature of binding to the two C-arches.

## ss–dsDNA junction recognition by D2–I

To visualize the interaction with an ss–dsDNA junction, we used a DNA with a 29 bp ds region flanked by 30 nt and 40 nt ssDNA overhangs (ss–dsDNA, Supplementary Table 1). We reasoned that the flanking ssDNA regions would trap D2–I on the ss–dsDNA junction. We determined a cryo-EM reconstruction of D2–I bound to this DNA at 3.6 Å resolution (Fig. 4b).

The complex of D2–I with ss–dsDNA was also in a closed configuration, clamped on DNA. However, in contrast to the structure with

fully dsDNA, the ds region of the ss–dsDNA is very well defined, with clearly visible major and minor grooves (Fig. 4b, Extended Data Figs. 7 and 8 and Extended Data Table 1). This suggests that the ss–dsDNA junction occupies a preferred binding position on D2–I with a specific base register. We propose that this represents a stalled D2–I complex.

We modelled a 24 bp DNA duplex into the map (Fig. 5a,b and Extended Data Fig. 7g). Like the dsDNA, the ds region of the ss–dsDNA first makes direct contact with the basic C-arch of FANCI. There are several contacts within this region, but one loop in FANCI (residues 793–798) seems to play a principal role in stabilizing the position of the DNA duplex by approaching the minor groove (Fig. 5b).

The well-defined density for the dsDNA region then extends straight into the base of the clamp and ends by contacting a lysine- and arginine-rich (KR) helix in the N-terminal domain of FANCD2 (residues 397–412; Fig. 5a,b and Supplementary Video 2). The dsDNA end is in close proximity to the kink in the dsDNA from the sliding complex. Finally, a weaker, more fragmented density, consistent with conformationally heterogeneous ssDNA, is visible (Fig. 5a,b and Extended Data Fig. 8b). Although it extends towards the C-arch of FANCD2, the fragmented nature of the putative ssDNA density precluded identification of any direct contacts. Given the asymmetry in D2–I, the stalled complex has directionality. This directionality could explain the presence of persistent and transient stalls (Extended Data Fig. 6d) and may have implications for the role of D2–I in DNA repair.

Overall, our structure of stalled D2–I shows that FANCI binds to dsDNA through its C-arch, but the FANCD2 KR helix specifically recognizes the ss–dsDNA junction by means of direct interactions with the DNA. All of these interactions are largely mediated by basic residues and the DNA backbone, consistent with a lack of DNA sequence specificity.

To test the importance of this binding mode, we made point mutations in two lysine and two arginine residues of the KR helix (to glutamic acid) and assessed their behaviour with our single-molecule assay. Charge reversal mutants diffuse on dsDNA but have a major defect in stalling at ss–dsDNA junctions (fourfold reduction in arginine mutants; tenfold reduction in lysine mutants; Fig. 5c). We did not observe robust DNA interaction for a variant with all four residues mutated in single-molecule assays. The reduction in stalling is consistent with corresponding reductions in DNA binding affinity (Extended Data Fig. 6e,f). Overall, these data are in agreement with the KR helix mediating specific recognition of ss–dsDNA junctions. Our data thereby provide a molecular mechanism for the localization of D2–I at stalled replication forks.

The trajectory of the DNA is different in the sliding (dsDNA) and in the stalled (ss–dsDNA) D2–I complexes; this provides insight into the mechanism of sliding versus stalling (Fig. 5d). The dsDNA in the sliding complex is kinked through contacting the C-arches of FANCI and FANCD2. Notably, these charge-mediated interactions probably facilitate the one-dimensional diffusion of D2–I along the DNA double helix[29]. By contrast, in the stalled complex, the dsDNA region is straight and extends from the C-arch in FANCI to the KR helix in FANCD2. The KR helix therefore directly stabilizes the interaction between D2–I and an ss–dsDNA junction, probably to inhibit sliding. The lower charge density on ssDNA prevents interaction with the FANCD2 C-arch, also preventing sliding. Moreover, binding to the junction may be more energetically favourable because the dsDNA region does not need to kink and ssDNA bends more readily than dsDNA. Together, these factors are likely to account for the notable stall of D2–I on ss–dsDNA junctions.

## The KR helix is important for DNA repair

To understand the importance of D2–I-mediated recognition of the ss–dsDNA junction in DNA repair, we tested whether FANCD2 KR helix mutants could rescue cellular survival after cisplatin treatment in chicken DT40 cells lacking FANCD2. Mutation of basic residues in the KR helix to alanine or glutamic acid compromised the ability of FANCD2 to rescue cell viability after treatment with cisplatin (Fig. 6a and Extended Data Fig. 9a). Thus, the KR helix contributes to an efficient cellular response to cisplatin.

To further test our model that recognition of ss–dsDNA junctions by the KR helix in FANCD2 is required for DNA ICL repair, we used a *Xenopus* egg extract system that supports repair of a site-specific ICL on a plasmid[30,31]. We depleted egg extracts of FANCD2 and complemented back recombinant wild-type or mutant D2–I complexes. Addition of wild-type D2–I supported ICL repair, but repair was inhibited in the presence of KR helix mutants (Fig. 6b and Extended Data Fig. 9b,c). Mutation of all four residues (D2$^{EEEE}$–I) resulted in severe inhibition, whereas the double mutants (D2$^{KREE}$–I and D2$^{EERR}$–I) retained some repair efficiency.

FANCD2 ubiquitination promotes FANCD2 localization to ICLs and is essential for repair. In the egg extract system, we observed no ubiquitination of the D2$^{EEEE}$–I mutant, whereas the two double mutants showed low levels of the modification (Fig. 6c and Extended Data Fig. 9b,c). Consistent with this, the D2$^{KREE}$–I and D2$^{EERR}$–I mutant complexes were recruited to ICL-containing plasmids during repair, albeit inefficiently (Fig. 6c). Thus, the ability of D2–I to stall at ss–dsDNA junctions correlates with DNA binding efficiency and ICL repair, indicating that the KR helix is important for ICL repair in *Xenopus* extracts.

## Discussion

In this work, we show that an unmodified D2–I clamp can close onto and scan long stretches of dsDNA. D2–I cannot bypass another D2–I clamp or a DNA obstacle, thereby constraining D2–I scanning to open chromatin. Consistent with this, we propose that D2–I surveys the nucleosome-depleted regions that occur around replication forks[32]. If the replisome stalls, exposed ss–dsDNA junctions would be recognized by scanning D2–I (Fig. 6d and Supplementary Video 2). Closed D2–I positioned at ss–dsDNA junctions would thus identify stalled replication forks, providing a unified model for the role of D2–I in several types of DNA repair and DNA replication stress.

Our model also reconciles how D2–I could protect stalled replication forks. By physically holding the duplex together at ss–dsDNA junctions, it could prevent the dsDNA region from fraying and from further resection. In agreement with this model, loss of FANCD2 in mice results in an increased number and size of chromosomal deletions[33]. These chromosomal deletions could be the consequence of nuclease sensitivity at ss–dsDNA junctions in the absence of FANCD2. Although FANCI alone can bind dsDNA, it cannot form a closed DNA clamp, and therefore would not protect replication forks in the absence of FANCD2. Further testing of this hypothesis will be the subject of future work.

D2–I interacts directly with RAD51 and promotes the formation of RAD51 filaments at stalled replication forks[34]. In the case of DNA crosslinks, ubiquitination of D2–I would lock it in position on the ss–dsDNA junction to recruit other downstream factors in the Fanconi anaemia pathway. Together, these data suggest that D2–I both identifies and protects the ss–dsDNA junction by clamping down on it, and facilitates initiation of homologous recombination-directed DNA repair by recruitment of downstream factors (Fig. 6d). Other DNA clamps also function as platforms to recruit repair enzymes to sites of DNA damage (for example 9-1-1 complex and PCNA). These clamps are dependent on adenosine triphosphate (ATP)-driven loaders to assemble on DNA[35,36], whereas D2–I loads and clamps dsDNA on its own, but is regulated by post-translational modifications.

It has long been a mystery as to what recognizes DNA crosslinks and why D2–I has been implicated in many other types of DNA repair and replication stress. Our work suggests that D2–I specifically localizes at structures around stalled or reversed replication forks, and it therefore

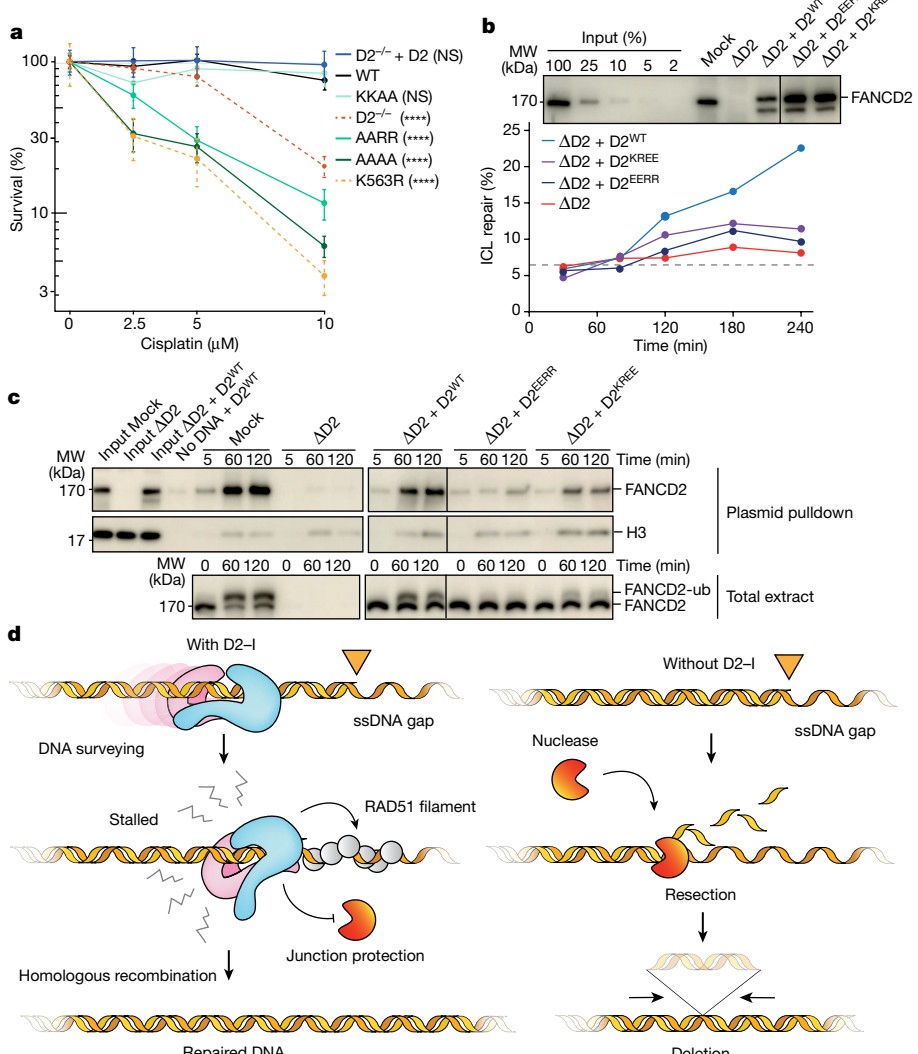

**Fig. 6 | The KR helix is important for DNA repair. a**, Clonogenic survival assay assessing cisplatin sensitivity of DT40 FANCD2$^{-/-}$ cells complemented with FANCD2$^{WT}$ (WT), FANCD2$^{K563R}$ (ubiquitination deficient) or KR helix variants (KKAA, AARR and AAAA). Results are based on two independent clones of each mutant and six replicates per clone, with mean and standard deviation plotted. *P* values (two-way analysis, Supplementary Table 2) enclosed in brackets are for samples compared with WT (****$P < 0.000001$). **b**, KR helix mutants have impaired ICL repair. Mock-depleted (Mock) and FANCD2-depleted (ΔD2) nucleoplasmic *Xenopus* egg extract were complemented with wild-type (D2$^{WT}$) or mutant FANCD2 (D2$^{EERR}$ or D2$^{KREE}$). FANCD2 levels were analysed by western blot (top panel). Lines indicate where irrelevant lanes were removed. Extracts were used to replicate an ICL-containing plasmid (pICL) and absolute ICL repair efficiency is plotted (bottom panel). Dotted line indicates background caused by uncrosslinked plasmid. This is representative of experiments performed twice (duplicate in Extended Data Fig. 9c). **c**, DNA binding and FANCD2 ubiquitination during ICL repair. During pICL replication, plasmids were isolated by LacR pull-down. Bound proteins (top blots, right) and a 1% input sample (top blots, left) or total extract (bottom blot) were analysed by western blot. This is representative of three independent experiments. **d**, Model for D2–I scanning dsDNA and stalling on ss–dsDNA junctions. D2–I loads onto chromatin and surveys the integrity of dsDNA by random one-dimensional diffusion. Exposed ssDNA gaps that occur at stalled replication forks cause D2–I to stall at the ss–dsDNA junctions. Closed D2–I clamps down on the ss–dsDNA junction, preventing further resection and minimizing DNA damage. In the absence of D2–I (right), unprotected junctions are more extensively resected, resulting in more frequent and longer chromosomal deletions. For gel source data, see Supplementary Fig. 1.

identifies the consequences of crosslinks. Because ss–dsDNA junctions and stalled replication forks are general features in replication stress and DNA damage response, this mechanism provides a unified molecular function for D2–I, reconciling its more general role in DNA repair.

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

## Methods

### Cloning and protein purification

A codon-optimized construct of full-length *Gallus gallus FANCD2* including a carboxyl-terminal double StrepII tag was synthesized (GeneArt) and cloned into pACEBac1. The ybbR protein tag comprising the sequence DSLEFIASKLA was fused to the C terminus of FANCD2 by polymerase chain reaction (PCR). A similar strategy was used for the generation of ybbR-tagged FANCI, except the *FANCI* gene contained a C-terminal extension with a 3C protease site and 6×His-tag before the ybbR tag (protein sequences in Supplementary Data 1). The resulting pACEBac1 vectors containing either tagged FANCD2 or tagged FANCI were transformed into EMBacY *Escherichia coli* competent cells for bacmid generation. The purified bacmids were transfected into Sf9 cells (Oxford Expression Technologies Ltd, Catalogue No. 600100) and the viruses passaged twice before large-scale cultures were infected using 5 ml of P2 virus in 500 ml of Sf9 cells at $1.5 \times 10^6$ cells per millilitre. Cells were harvested on growth arrest, typically 2 or 3 days after infection. Sf9 cells were not authenticated but were tested regularly for *Mycoplasma*.

Purification of ybbR-tagged FANCD2 was carried out as previously described for FANCD2 (ref. 9): cells were lysed by sonication in lysis buffer (100 mM HEPES (pH 7.5), 300 mM NaCl, 1 mM TCEP, 5% glycerol, EDTA-free protease inhibitor (Roche), 5 mM benzamidine hydrochloride and 100 U ml$^{-1}$ Benzonase). The clarified cell lysate was incubated with Strep-Tactin Sepharose high-performance resin (GE Healthcare Life Sciences) for 60 min. The loaded resin was poured into a glass column and washed twice with lysis buffer before elution with 8 mM D-desthiobiotin. The elution was then diluted to roughly 100 mM NaCl and loaded onto a HiTrap Heparin HP affinity column (GE Healthcare Life Sciences). Using a shallow NaCl gradient, ybbR-tagged FANCD2 eluted at about 500 mM NaCl. FANCD2 was concentrated and run on a Superdex 200 26/60 column (GE Healthcare Life Sciences) in 50 mM HEPES (pH 7.5), 150 mM NaCl and 1 mM TCEP. Fractions containing pure ybbR-tagged FANCD2 were pooled, concentrated to roughly 10 mg ml$^{-1}$ and flash frozen for storage at −80 °C or used immediately for fluorescence labelling.

Similarly, purification of ybbR-tagged FANCI was performed as previously described for FANCI[9]: clarified cell lysate produced as for FANCD2 was loaded onto a HisTrap HP column (GE Healthcare Life Sciences). Using an imidazole gradient, FANCI eluted at about 200 mM imidazole. Collected fractions containing FANCI were diluted to 100 mM NaCl and loaded onto a HiTrap Heparin HP affinity column (GE Healthcare Life Sciences). Using a shallow NaCl gradient, FANCI eluted at roughly 500 mM NaCl. FANCI was then run on a Superdex 200 26/60 column (GE Healthcare Life Sciences) in 50 mM HEPES (pH 7.5), 150 mM NaCl and 1 mM TCEP. Fractions containing FANCI were pooled and concentrated to about 10 mg ml$^{-1}$ and flash frozen for storage at −80 °C or used immediately for fluorescence labelling.

Human FANCD2 and FANCI, which both carry a carboxyl-terminal double StrepII tag, were synthesized and cloned into pACEBac1 (Epoch Life Science). Human FANCD2 and FANCI were expressed and purified separately as individual proteins in the same way as chicken FANCD2 and FANCI with two adjustments. Both human FANCD2 and FANCI were incubated with Strep-Tactin resin (IBA), eluted with 8 mM D-desthiobiotin and incubated with the 3C protease (1:100 ratio of protease to the protein of interest) to cleave the carboxyl-terminal tag for 16 h at 4 °C. Following heparin affinity, human FANCD2 and FANCI were concentrated and run on a Superdex 200 16/60 column (GE Healthcare Life Sciences). Fractions containing the protein of interest were finally concentrated to about 4.5 mg ml$^{-1}$ and flash frozen for storage at −80 °C.

Sample purity was assessed at each step of the purification by sodium dodecyl sulfate (SDS)–polyacrylamide gel electrophoresis (PAGE) using 4–12% NuPAGE Bis-Tris gels (Thermo Fisher Scientific), and the Gel Doc XR+ system (Bio-Rad) was used for gel imaging. Throughout purification, we routinely monitored the absorbance at 260 nm and 280 nm.

The A260 nm to A280 nm ratio showed that there is no substantial nucleic acid contamination in the protein purifications.

To generate the FANCD2 KR helix mutants, the respective lysines or arginines were mutated to alanine or glutamic acid by replacing the KR helix in wild-type FANCD2 with the respective gBlocks (Integrated DNA Technologies (IDT); Supplementary Table 1) using Gibson assembly. Sanger sequencing confirmed the correct substitutions. Mutant FANCD2 proteins were expressed and purified as described for wild-type FANCD2.

Biotinylated LacR protein was prepared as previously described[37].

### Fluorescent protein labelling

To generate site-specific fluorescently labelled proteins, we used SFP synthase (gift from J. Rhodes, Medical Research Council (MRC) Laboratory of Molecular Biology (LMB)) to conjugate CoA-activated fluorophores to the carboxyl-terminal ybbR tag. We incubated ybbR-tagged full-length FANCD2 with SFP synthase and a modified Cy3 (LD555-CoA, Lumidyne Technologies) at a 1:5:5 molar ratio for 16 h at 4 °C in 50 mM HEPES (pH 7.5), 150 mM NaCl, 1 mM TCEP and 10 mM MgCl$_2$. Similarly, ybbR-tagged full-length FANCI was incubated with SFP synthase and a modified Cy5 (LD655-CoA, Lumidyne Technologies) using the same conditions. Labelled protein was purified from excess free dye by gel filtration using a Superdex 200 10/300 column (GE Healthcare Life Sciences) equilibrated in 50 mM HEPES (pH 7.5), 150 mM NaCl, 1 mM TCEP and 1 mM MgCl$_2$. Fractions containing labelled protein (either FANCD2-Cy3 or FANCI-Cy5) were pooled, concentrated to 1–2 µM and aliquoted, flash frozen in liquid N$_2$ and stored at −80 °C until further use. The labelling efficiency was estimated using the extinction coefficients of ybbR-tagged FANCD2 or FANCI and either Cy3 or Cy5, respectively. All labelling reactions yielded an estimated 90–95% efficiency.

### DNA binding experiments

To assess the DNA-binding activity of fluorescently labelled D2–I, we carried out electrophoretic mobility shift assays (EMSAs) as previously described[9,11]. Fluorescently labelled dsDNA (purchased from IDT) was prepared by incubating complementary oligonucleotides P1 (3′ FAM-labelled) and P7 (Supplementary Table 1) at 95 °C for 5 min and slowly cooling down to room temperature over roughly 2 h. For EMSAs, a 20 µl reaction containing 20 nM DNA was incubated with the indicated concentration of protein in the presence of 50 mM HEPES (pH 8.0), 75 mM NaCl and 1 mM TCEP for 30 min at 22 °C. After incubation, a 5 µl aliquot was directly loaded onto a native polyacrylamide gel (6% DNA Retardation, Thermo Fisher) and run at 4 °C in 0.5× TBE buffer for 60 min. The gel was then visualized using a Typhoon Imaging System (GE Healthcare). Each binding experiment was repeated three times (Extended Data Fig. 1c).

### SwitchSENSE

The interactions of D2–I$^{WT}$, D2$^{EEEE}$–I, D2$^{KKEE}$–I and D2$^{EERR}$–I with dsDNA were analysed on a DRX2 instrument (Dynamic Biosensors GmbH) using a MPC2-48-2-G1R1-S chip equilibrated with SwitchSENSE buffer (20 mM HEPES (pH 8.0), 75 mM NaCl, 1 mM TCEP) at 25 °C. Before each kinetic analysis, 30-48-cNLB, 38-cNLB, 38P-cNLB or cNLB, together with cNLA (Supplementary Table 1), were annealed to DNA strands attached to the chip surface (NLB and NLA, respectively) by flowing 500 nM oligonucleotide over the chip for 4 min in a buffer of 10 mM Tris-HCl, 40 mM NaCl, 0.05% (v/v) Tween-20, 50 µM EDTA and 50 µM EGTA. As a result, either 48 bp dsDNA nanolevers with a 30 bp ss 5′ overhang were formed or a 38 bp dsDNA nanolever with a 10 bp 3′ overhang with a phosphate at the 5′ end was formed. Proteins in a 1:3 dilution series, starting at 100 nM protein, were injected at 50 µl min$^{-1}$ for 5–10 min followed by dissociation in running buffer after the highest concentration for 80 min at 50 µl min$^{-1}$. Dynamic switching data were analysed using the supplied switchANALYSIS v.1.9.0.33 software using a 1:1 kinetic model to give values for the association rate constant

$k_{on}$ and the dissociation rate constant $k_{off}$, and to calculate the kinetic dissociation constant $K_d = k_{off}/k_{on}$.

## Ubiquitination assays

To confirm that fluorescent labelling did not affect the activity of the D2–I complex, we performed FANCL-mediated monoubiquitination assays as previously described[38]. The reaction is based on previously described ubiquitination assays[9,11,14,38,39]: 75 nM human (hs) E1 ubiquitin activating enzyme (Boston Biochem), 0.8 µM E2 (hsUbe2Tv4), 1 µM E3 (hsFANCL[109–375]), 1 µM D2–I, 5 µM dsDNA (oligos P1 and P7; Supplementary Table 1) and 20 µM His-tagged ubiquitin (Enzo Life Sciences) were used in a total volume of 10 µl with a reaction buffer of 50 mM HEPES (pH 7.5), 64 mM NaCl, 4% glycerol, 5 mM MgCl$_2$, 2 mM ATP and 0.5 mM DTT. The reaction was incubated at 30 °C for 90 min and samples were analysed by SDS–PAGE (Extended Data Fig. 1d).

## Preparation of monoubiquitinated FANCD2

To prepare Cy3 labelled monoubiquitinated FANCD2 for single-molecule imaging, an in vitro reaction was performed by mixing 75 nM hsE1 ubiquitin activating enzyme (Boston Biochem), 0.8 µM hsUbe2Tv4 (ref. 38), 3 µM hsFANCL[109–375], 1 µM *Gallus gallus* FANCD2_ybbR and 20 µM His-tagged ubiquitin (Enzo Life Sciences) in a reaction buffer of 50 mM HEPES (pH 7.5), 64 mM NaCl, 4% glycerol, 5 mM MgCl$_2$, 2 mM ATP and 0.5 mM DTT in a total of 500 µl. The reaction was incubated at 30 °C for 90 min before applying it to 50 µl of Ni–NTA agarose resin (Qiagen) pre-equilibrated in W25 buffer (20 mM HEPES (pH 7.5), 150 mM NaCl, 1 mM TCEP and 25 mM imidazole) in a 1.5 ml centrifuge tube at 4 °C for 60 min. The resin was washed twice with 100 µl of W25 buffer (20 mM HEPES (pH 7.5), 150 mM NaCl, 1 mM TCEP and 25 mM imidazole). Each wash was performed for 30 min at 4 °C under rotation. The Ni–NTA-bound ubFANCD2 was eluted with W100 buffer (20 mM HEPES (pH 7.5), 150 mM NaCl, 1 mM TCEP and 100 mM imidazole) as previously described for the purification of ubD2–I[9]. We incubated ubFANCD2_ybbR with SFP synthase and Cy3 (LD555-CoA, Lumidyne Technologies) at a 1:5:5 molar ratio for 16 h at 4 °C in 50 mM HEPES (pH 7.5), 150 mM NaCl, 1 mM TCEP and 10 mM MgCl$_2$. Labelled ubFANCD2 was further purified from free dye by gel filtration using a Superdex 200 10/300 column (GE Healthcare Life Sciences) equilibrated in 50 mM HEPES (pH 7.5), 150 mM NaCl, 1 mM TCEP and 1 mM MgCl$_2$ (Extended Data Fig. 3g). Fractions containing labelled ubFANCD2 were pooled, concentrated to 1–2 µM and aliquoted before being flash frozen on liquid N$_2$, and stored at −80 °C until further use. The labelling efficiency was estimated using the extinction coefficients of ybbR-tagged FANCD2 and Cy3. All labelling reactions yielded an estimated 90–95% efficiency.

## DNA for single-molecule studies

Bacteriophage λ DNA (Thermo Scientific) was labelled at both ends with biotin using Klenow polymerase exo⁻ (New England Biolabs). The linear substrate (4 nM) was incubated with 100 µM dGTP, 100 µM dTTP, 80 µM biotin-14-dATP and 80 µM biotin-14-dCTP and the enzyme (0.5 U from stock of 50 U µl⁻¹) in NEB2 buffer at 37 °C for 30 min, then at 70 °C for 15 min and subsequently cooled on ice. The product was purified using Qiagen PCR clean-up kit.

To prepare λ DNA with a site-specific ssDNA gap, a λ DNA construct was treated with CRISPR–Cas9$_{D10A}$ nickase (nCas9) (IDT) as described in ref. 27. In brief, λ DNA and two ssDNA biotinylated oligos (cap 1 and cap 2; Supplementary Table 1) were phosphorylated and subsequently ligated together using T4 ligase at 37 °C for 1 h to form λ DNA with closed, biotinylated ends[40]. To quench the reaction, T4 ligase enzyme was inactivated at 65 °C for 20 min. Next, two nCas9–RNA complexes (complex 1: nCas9 + tracrRNA + crRNA 1, complex 2: nCas9 + tracrRNA + crRNA 2; Supplementary Table 1) were incubated in Cas9 digestion buffer (50 mM Tris-HCl (pH 8), 100 mM NaCl and 10 mM MgCl$_2$) with the biotinylated λ DNA at 37 °C to induce two site-specific nicks.

crRNA 1 was labelled with a Cy3 fluorescent dye to generate the fiduciary static nCas9 marker. For D2–I experiments, this nicked λ DNA was incubated with proteinase K enzyme for 15 mins at 56 °C to remove the nCas9 complexes bound to the DNA. The ssDNA gap was generated in situ by force-induced melting during the single-molecule experiment. The ssDNA gap in the Atto647N-labelled 17.8-kb-long dsDNA with site-specific nicks (LUMICKS) used for characterizing the spatial resolution of the single-molecule assay (Supplementary Fig. 3) was also generated in a similar manner in situ.

The DNA with a four-way junction was prepared by ligating 7.5 kb handles with phosphorylated, hairpin-forming, synthetic oligonucleotides, as described in ref. 26. The handles were prepared by PCR using λ DNA as a template and two modified primers. Primer 1 contains four biotin residues at its 5′ end. Primer 2 is 5′-phosphorylated and contains a single abasic site 10 nt from the 5′ end (Supplementary Table 1). A proofreading polymerase Pfu Ultra II Fusion HS (Agilent) was used to generate a 10 nt overhang beyond the abasic site, which enables ligation of the hairpin-forming oligonucleotides. The PCR products were purified on a QIAquick PCR purification column (Qiagen). The hairpin-forming, phosphorylated oligonucleotides with 10 nt 5′ overhangs were annealed in a hybridization buffer (10 mM Tris-HCl (pH 7.5), 50 mM NaCl) that was slow-cooled from 80 °C to room temperature. The annealed product was ligated to 7.5 kb handles by incubating equimolar ratio (3 pmol) with 400 U of T4 DNA ligase (New England Biolabs) in a ligase buffer at 16 °C for 8 h. To stop the reaction, 20 mM EDTA, 0.017% SDS was added and the solution was further incubated at 65 °C for 10 min. The product was purified by electrophoresis in a 1x TAE, 0.6% agarose, 1x TAE gel followed by electroelution of the excised product band and ethanol precipitation.

## Optical tweezers experiments

Single-molecule experiments were performed on a C-trap (LUMICKS) integrating optical tweezers, confocal fluorescence microscopy and microfluidics. The five-channel laminar flow cell was passivated using 0.5% (w/v) Pluronics F128 in phosphate buffered saline (PBS), and subsequently with bovine serum albumin (BSA) (1 mg ml⁻¹). Streptavidin-coated polystyrene beads, 0.005% w/v (4.8 µm or 4.35 µm, Spherotech), were injected into channel 1. Biotin-labelled DNA molecules (about 2 pM) were flowed into channel 2. Buffer A, containing 20 mM HEPES (pH 7.5), 75 mM NaCl, 0.5 mg ml⁻¹ BSA and 1 mM TCEP, was injected into channel 3. D2–I was diluted to 5 nM in buffer A and injected into channel 4. For experiments containing RPA, eGFP-RPA was diluted to 800 pM in buffer A and injected into channel 5. For ubD2–I experiments, the complex was formed by incubating purified ubFANCD2 and FANCI with relaxed, optically trapped λ DNA (less than 1 pN in the protein channel). The ubD2–I complex on DNA was then moved to the protein-free channel and kymographs were acquired at 15 pN.

The optical trap was calibrated to achieve a trap stiffness of 0.2–0.3 pN nm⁻¹. After optically trapping two beads, the DNA molecule was suspended between the beads in channel 2. The presence of the DNA tether was verified by measuring a force–extension curve in channel 3 using a constant pulling rate of 0.2 µm s⁻¹ and the acquisition rate of 60 Hz. Subsequently, the DNA tether was moved to the protein channel (channel 4) and incubated for 10–30 s. In most of the experiments, this loading step was performed using DNA held at very low force (less than 1 pN). After protein loading, the sliding experiment was carried out in the same buffer (channel 3) but without free protein. The confocal images were acquired in the absence of flow, using 488, 532 and 638 nm lasers (eGFP, Cy3 and Cy5 excitation, respectively) at a laser power of less than 3 µW. Fluorescence emission was recorded using blue (512/25 nm), green (585/75 nm) and red (640LP) filters. Kymographs were acquired by scanning the DNA contour with a pixel dwell time of 0.1–0.2 ms px⁻¹, resulting in frame rates in the range of about 100 ms, depending on the DNA tether length. All experiments were performed at room temperature (22 °C).

## Single-molecule data interpretation

Raw data exported from LUMICKS Bluelake as .h5 files were processed with custom-written Jupyter Notebooks in Python 3.9 using LUMICKS Pylake v.1.2.1, numpy v.1.26.0, matplotlib v.3.7.2, scipy v.1.11.3 and peakutils v.1.3.4 (https://github.com/singlemoleculegroup). Additional adjustments (colour contrast, cropping) were performed in Fiji[41]. Final graphs were generated in Prism.

The particle localization and the mean square displacement (MSD) analysis were incorporated in the same Jupyter Notebook workspace using a custom tracking algorithm. The script generates a list of points that correspond with the localization of the fluorescent molecule in each time frame. Simultaneously, a list of photon counts (intensities) in each pixel in the red and green channels was generated.

MSDs from the resolved, unprocessed trajectory were calculated using the formula below:

$$\text{MSD}(n, N) = \sum_{i=1}^{N-n} \frac{(X_{i+n} - X_i)^2}{N - n} = 2D\tau + b, \tag{1}$$

where $N$ is the number of frames in the kymograph, $n$ is the size of the moving window (corresponding to the lag time $\tau$) ranging from 1 to $N-1$, $X_i$ is the particle position at the frame $i$ and $b$ is the offset. The average diffusion coefficient ($D$) of the particle was obtained from the slope of the linear fit of MSD as a function of $\tau$ between $0.15 < \tau < 1$ s. To estimate the speed of the protein, the total displacement of the particle over time was calculated. Before the speed calculation, the trajectory was smoothed using a Savitzky–Golay filter.

For the rolling diffusion analysis, the resolved, unsmoothed trajectory was divided into several rolling windows (with a window size of 16 time points, corresponding to roughly 1 s). MSDs were calculated for each window (using equation (1) above). The MSDs (for the first three lag times or $\tau$s) were fit to a straight line using the relation

$$\text{MSD}(\tau) = 2D\tau, \tag{2}$$

to yield the rolling diffusion coefficient ($D_{\text{roll}}$). Given the small number of sample points used for each window, the offset parameter used for the average $D$ analysis was omitted for the $D_{\text{roll}}$ analysis. The threshold $D_{\text{roll}}$ used to distinguish between stalled and diffusing D2–I molecules was set to $6.4 \times 10^{-3}$ μm$^2$ s$^{-1}$, which was thrice the average $D_{\text{roll}}$ for the static nCas9 (Cy3) molecule on λ DNA (Supplementary Fig. 3).

The intensity of the trajectory at each time frame was calculated by integrating the total intensities of six pixels in the vicinity of the centre of the Gaussian peak (3 and −3). The intensity analysis from FRET quantification was done in IGOR. The intensities of donor and acceptor (Cy3- and Cy5-labelled units, respectively), $I_{\text{D\_raw}}$ and $I_{\text{A\_raw}}$, were filtered using the smoothing BOX function over six data points.

Subsequently, fluorescence bleed-through correction was performed. The average filtered intensities of the green ($I_{\text{D\_avg}}$) and red channels ($I_{\text{A\_avg}}$) after acceptor photobleaching were calculated. The correction factor $\alpha$ was calculated as:

$$\alpha = \frac{I_{\text{A\_avg}}}{I_{\text{D\_avg}}}. \tag{3}$$

The corrected acceptor intensity was therefore:

$$I_{\text{A}}^* = I_{\text{A}} - I_{\text{D}} \times \alpha. \tag{4}$$

FRET was calculated using the corrected acceptor intensity:

$$\text{FRET} = \frac{I_{\text{A}}^*}{I_{\text{A}}^* + I_{\text{D}}}. \tag{5}$$

For the lifetime analysis, dwell times of D2–I stalled at the ss–dsDNA junctions were estimated directly from the kymographs. The survival probability ($S$) for the dwell-time distribution was calculated as

$$S(d) = \frac{\text{Number of dwells greater than } d}{\text{Total number of dwells}} = 1 - \text{CDF}, \tag{6}$$

where CDF is the cumulative density function of the distribution. The survival probability was subsequently fit to a double exponential decay to the form

$$S(d) = A_1 e^{-\frac{d}{t_1}} + A_2 e^{-\frac{d}{t_2}}, \tag{7}$$

yielding the average lifetimes of the two kinetic phases, $t_1$ and $t_2$.

Final plots were generated using Prism, Wavemetrics IGOR 8 or in Jupyter Notebooks using matplotlib v.3.7.2.

## Estimation of protein density on λ DNA

To compare the loading efficiency of D2–I on λ DNA (Extended Data Fig. 2a,b) we estimated the number of bound D2–I molecules at different NaCl concentrations (150, 100, 75 and 37 mM). We used Fiji[41] to plot the two-dimensional profile along λ DNA (30 s after starting the kymograph acquisition). To estimate the number of bound D2–I molecules, we calculated the total area delimited by the distance ($x$ axis) and the pixel intensity ($y$ axis) and divided it by the area corresponding to a single peak (a single molecule). This process was performed for three representative kymographs for each NaCl concentration.

## Electron microscopy and image processing

For cryoEM, untagged D2–I was prepared as described[9]. Briefly, a pBIG1a vector containing Gallus gallus FANCD2 and FANCI was used for bacmid and virus preparation, protein expression, cell lysis and clarification, as described above for tagged FANCD2 and FANCI. For purification of untagged D2–I, the clarified lysate was applied to a HiTrap SP HP cation exchange chromatography column (GE Healthcare Life Sciences) to remove impurities. The flow-through, containing the D2–I complex, was diluted to 150 mM NaCl and loaded onto a HiTrap Heparin HP affinity column (GE Healthcare Life Sciences). Using a shallow NaCl gradient, the D2–I complex eluted at ~500 mM NaCl concentration. The complex was run on a Superdex 200 26/60 column (GE Healthcare Life Sciences) in 50 mM HEPES, pH 7.5, 150 mM NaCl and 1 mM TCEP. The peak fractions containing pure D2–I were concentrated to ~10 mg ml$^{-1}$ and flash frozen until use.

ss–dsDNA was prepared by incubating oligos J1 and J2 (both containing 3′-OH and 5′-OH ends; purchased from IDT; Supplementary Table 1) at 95 °C for 5 min and slowly cooling down to room temperature over 3 h. For cryo-EM analysis of chicken D2–I bound to ss–dsDNA, we incubated 1 μM D2–I complex with 3 μM ss–dsDNA in imaging buffer (used for single-molecule experiments: 20 mM HEPES (pH 7.5), 75 mM NaCl and 1 mM TCEP) for 15 min at 22 °C. A total of 3 μl of sample was applied onto plasma-cleaned Quantifoil 1.2/1.3 grids for 3.5 s before blotting and vitrification in liquid ethane using a Vitrobot Mark IV (Thermo Fisher) at 4 °C and 100% humidity. The grids were imaged using EPU v.3.4.0 on a Titan Krios (Thermo Fisher) operated at 300 keV using a Gatan K3 detector in super-resolution model at the Electron Bio-Imaging Centre (eBIC). About 20,000 videos were collected at a pixel size of 0.831 Å. We used a defocus range spanning −1.2 to −2.8 in 0.3 μm steps and a total dose of roughly 40 e/Å$^2$ per image.

All image processing was performed using RELION v.4.0 (ref. 42) unless otherwise stated (Supplementary Fig. 4). Multiframe videos were drift corrected using 5 × 5 patches in MotionCorr2 (ref. 43) and defocus was estimated using CTFFIND4 (ref. 44). Particle picking was performed using crYOLO[45] in trained mode after manual picking of about 3,000 particles. Around 6 million particles were initially extracted with a pixel size of roughly 2.9 Å, with intentional overpicking to prevent missing

rare views of the complex. After initial two-dimensional classification, about 2.4 million particles were subjected to three-dimensional classification, using as a reference a low-pass filtered (60 Å) map generated ab initio in RELION. Classes showing clear DNA density were selected and refined after re-extraction with the original pixel size of 0.831 Å. The defocus values of the consensus reconstruction were further refined using CTF refinement, followed by Bayesian polishing and a further round of CTF refinement. The resulting consensus refinement was subjected to three-dimensional classification without image alignment. The best classes were combined and, after refinement, postprocessed using a soft mask to correct for modulation transfer function of the detector and sharpened with a $B$ factor of −120, as determined using RELION. Fourier shell correlation curves generated after postprocessing using a cut-off value of 0.143 yielded a map with an estimated resolution of 3.58 Å. Local resolution was calculated in RELION and displayed onto the consensus refinement in ChimeraX[46]. Map sharpening of the refined maps using DeepEMhancer[47] yielded improved densities in the N termini of FANCD2 and FANCI, and aided in model building.

Linear dsDNA was prepared by incubating oligos P1 and P7 (purchased from IDT; Supplementary Table 1) at 95 °C for 5 min and slowly cooling down to room temperature over 3 h. The dsDNA−D2−I structures for both chicken and human were obtained using the same procedures in grid preparation, data acquisition and essentially the same processing strategy as described above (detailed in Supplementary Fig. 4 for chicken dsDNA−D2−I).

### Structure modelling
The structure of *Gallus gallus* D2−I bound to dsDNA (PDB 8A2Q)[11] was rigidly fitted into the map of D2−I bound to ss−dsDNA using Chimera[48]. After manual inspection and adjustment in Coot[40] and ISOLDE[49], the model was iteratively refined in Coot and Phenix[50]. An idealized dsDNA of 24 bp was placed and refined into the duplex density using ISOLDE with distance restraints, followed by refinement in Coot and Phenix. All models and maps were inspected in ChimeraX, which was also used to prepare figures.

### DT40 cell culture and mutant generation
DT40 cells (obtained from Dr J. Young at the Institute of Animal Health, Compton, Berkshire, UK) were cultured as previously described[51]. The DT40 cell line was authenticated by Bu-1a and b expression and immunoglobulin gene sequences. No *Mycoplasma* contamination was detected. The DT40 FANCD2 knockout cell line was generated by Yamamoto et al.[52]. FANCD2 point mutations were generated using Quick Change II XL Site-Directed Mutagenesis Kit (Agilent), according to the manufacturer's instructions, in pcDNA3.1 GFP-chFANCD2 plasmid[52]. Primers for site-directed mutagenesis were designed using the Agilent QuikChange Primer Design online tool, and sequences are reported in Supplementary Table 1. Each mutant plasmid was then checked by Nanopore30 sequencing (Source Bioscience). Stable non-targeted transfections of DT40 FANCD2[−/−] were performed as described in ref. 53 and puromycin-resistant clones selected.

### Colony survival assay
Colony survival assays were performed as described in ref. 54 replacing D-MEM by RPMI Medium 1640 (Gibco), under penicillin/streptomycin selection, and Puromycin (1 μg ml[−1]) for FANCD2[−/−] mutants. For each mutant, two independent clones and six replicates per clone were performed. Cisplatin (Merck) was freshly prepared in a 0.9% saline solution, and cells were treated for an hour with the indicated doses. Cells were then washed, plated on methylcellulose and cultured until colonies were visible (about 2 weeks). The relative clonogenic survival, compared with untreated cells, was then calculated for each replicate. Replicates were then pooled, and statistical testing was performed by the R package CFAssay using two-way analysis of the cell survival data

by fitting a linear quadratic model with maximum likelihood (*P* values reported in Supplementary Table 2).

### Preparation of *Xenopus* egg extracts
The eggs of *Xenopus laevis* female frogs (aged more than 2 years, purchased from Nasco) were used for extract preparation. Nucleoplasmic extract (NPE) and high-speed supernatant (HSS) were prepared as previously described[55]. All animal procedures were performed in accordance with national animal welfare laws and were reviewed by the Animal Ethics Committee of the Royal Netherlands Academy of Arts and Sciences (KNAW). All animal experiments were conducted under a project licence granted by the Central Committee Animal Experimentation (CCD) of the Dutch government and approved by the Hubrecht Institute Animal Welfare Body (IvD), with project licence number AVD80100202216633. Sample sizes were chosen on the basis of previous experience; randomization and blinding are not relevant to this study.

### DNA repair assay in *Xenopus* extracts
DNA replication and preparation of *Xenopus* egg extracts (HSS and NPE) were performed as described previously[31,56,57]. Preparation of plasmid with a site-specific cisplatin ICL (pICL) and ICL repair assays were performed as described[30,31,58]. Briefly, pICL was incubated with HSS containing [32]P-a-dCTP for 20 min, following addition of two volumes of NPE (*t* = 0). Aliquots of replication reaction were stopped at various times with ten volumes of Stop Solution II (0.5% SDS, 10 mM EDTA and 50 mM Tris (pH 7.5)). Samples were incubated with proteinase K (0.5 μg μl[−1]) for 1 h at 37 °C each. DNA was extracted using phenol/chloroform, ethanol-precipitated in the presence of glycogen (30 mg ml[−1]) and resuspended in TE buffer. ICL repair was analysed by digesting 1 μl of extracted DNA with HincII, or HincII and SapI, separation on a 0.8% native agarose gel and quantification using autoradiography. Absolute ICL repair efficiency was calculated as described[31]. For reconstitutions with D2−I proteins, replication reactions were supplemented with approximately 60 nM of recombinant *Xenopus laevis* D2[WT]−I or D2[EEEE]−I or D2[EERR]−I or D2[KREE]−I.

### Antibodies and immunodepletions
The antibody used to deplete and blot for *xl*FANCD2 was previously described[30,31]. Histone H3 antibody was purchased from Abcam (ab1791) and used in 1:4,000 dilution. The *xl*FANCD2 antibody was affinity purified against the antigen used to raise the antibody. To deplete *Xenopus* egg extracts of FANCD2, Dynabeads Protein A beads (Thermo Fisher Scientific) were incubated with the affinity-purified antibody to their maximum binding capacity. One and a half volumes of the antibody-coated beads were then mixed with one volume of pre-cleared HSS or NPE and incubated for 30 min at room temperature. Mock depletions were performed using non-specific immunoglobulin G from rabbit serum (Sigma-Aldrich). Depleted extracts were collected and immediately used for replication assays.

### Plasmid pull-down
Plasmid pull-downs were performed as previously described[59]. Briefly, streptavidin-coupled magnetic beads (Dynabeads M-280, Invitrogen; 6 μl per pull-down) were washed with 50 mM Tris (pH 7.5), 150 mM NaCl, 1 mM EDTA (pH 8) and 0.02% Tween-20. Biotinylated LacR was added to the beads (4 pmol per microlitre of beads) and incubated at room temperature for 45 min. The beads were washed with 10 mM HEPES (pH 7.7), 50 mM KCl, 2.5 mM MgCl$_2$, 250 mM sucrose, 0.25 mg ml[−1] BSA and 0.02% Tween-20 and resuspended in 40 μl of the same buffer. The bead suspension was stored on ice until needed. At indicated time points, 8 μl of the replication reaction was gently mixed with LacR−streptavidin Dynabeads. The suspension was incubated at 0 °C for 30 min. The beads were washed with 10 mM HEPES (pH 7.7), 50 mM KCl, 2.5 mM MgCl$_2$, 0.25 mg ml[−1] BSA, 100 mM NaCl and 0.5% Triton X-100.

All residual buffer was removed and the beads were resuspended in 2× SDS sample buffer (150 mM Tris-HCl (pH 6.8), 5% SDS, 0.05% bromophenol blue and 20% glycerol). DNA-bound proteins were then resolved by SDS–PAGE and visualized by western blotting with the indicated antibodies (anti-FANCD2 at 1:6,000; anti-Histone-H3 at 1:4,000). The recombinant D2–I complexes used for plasmid pull-down experiments were incubated with PreScission Protease (Cytiva) for 1 h at room temperature before being added to extract. This allows for the cleavage of the strep-tag from the recombinant FANCD2 to prevent direct binding of the protein to the streptavidin beads.

## Reporting summary

Further information on research design is available in the Nature Portfolio Reporting Summary linked to this article.

## Data availability

Models and maps have been deposited to PDB and EMDB and assigned the following accession codes: PDB ID 9FFF, EMD-50355 for dsDNA–D2–I; PDB ID 9FFB, EMD-50353 for ss–dsDNA–D2–I. Raw data are available at Zenodo (https://doi.org/10.5281/zenodo.11521474)[60]. The code for analysis of single-molecule data is freely available (https://github.com/singlemoleculegroup). Correspondence and requests for materials should be addressed to L.A.P. or D.S.R. All unique materials are available on request with completion of a standard Materials Transfer Agreement. Source data are provided with this paper.

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

**Acknowledgements** We thank J. Rhodes (MRC LMB) for providing SFP synthase and advice on protein labelling. We thank G. Moore (Imperial) for assistance and expert advice. We thank M. Maric, T. Stanage, O. Belan, R. Anand and S. Boulton (Crick) for labelled RPA protein and advice, and D. Lilley (Dundee) for Holliday junction DNA. We thank J. G. Shi (MRC LMB), the MRC LMB EM facility, J. Grimmett and T. Darling (MRC LMB scientific computation) for support. We thank K. J. Patel (Oxford), C. J. Russo, C. Johnson, S. Chaaban, J. T. P. Yeeles, S. S. H. W. Scheres (MRC LMB) and all members of the Passmore and Rueda groups for useful discussions and advice. This work was supported by the MRC as part of UK Research and Innovation, MRC file reference number MC_U105192715 (L.A.P.), U105178808 (J.E.S.) and MC-A658-5TY10 (D.S.R.); a Wellcome Trust Collaborative Grant 206292/Z/17/Z (D.S.R.); an ERC Consolidator Grant (ERCCOG 101003210-XlinkRepair, to P.K.); and an EMBO Long-Term Fellowship ALTF 692–2018 (P.A.). We acknowledge Diamond Light Source for access to eBIC and excellent support (proposals BI23268 and BI31336) funded by the Wellcome Trust, MRC and the Biotechnology and Biological Sciences Research Council. For the purpose of open access, the MRC LMB has applied a CC BY public copyright licence to any Author Accepted Manuscript version arising.

**Author contributions** P.A., L.A.P. and D.S.R. conceived the study and designed the experiments with input from A.P.K. P.A. purified and labelled proteins, performed DNA binding and ubiquitination assays, prepared cryo-EM samples and determined the structures. P.A., A.P.K. and K.K.R. performed optical tweezers experiments and analysed the data. A.P.K. and K.K.R. wrote analysis scripts and processed data. T.S. purified human D2–I. T.S. and P.A. prepared samples and determined structures of human D2–I. Y.S. and P.A. purified FANCD2 mutants. T.L. and P.K. performed the *Xenopus* egg extract experiments. G.G. and J.E.S. generated the DT40 mutants and performed cisplatin sensitivity assays. S.H.M. performed SwitchSENSE. P.A. wrote the initial draft of the manuscript and prepared figures with input from all authors. D.S.R. and L.A.P. supervised the project.

**Competing interests** A.P.K. was an employee of LUMICKS during review of this manuscript. The other authors declare no competing interests.

**Additional information**
**Correspondence and requests for materials** should be addressed to David S. Rueda or Lori A. Passmore.

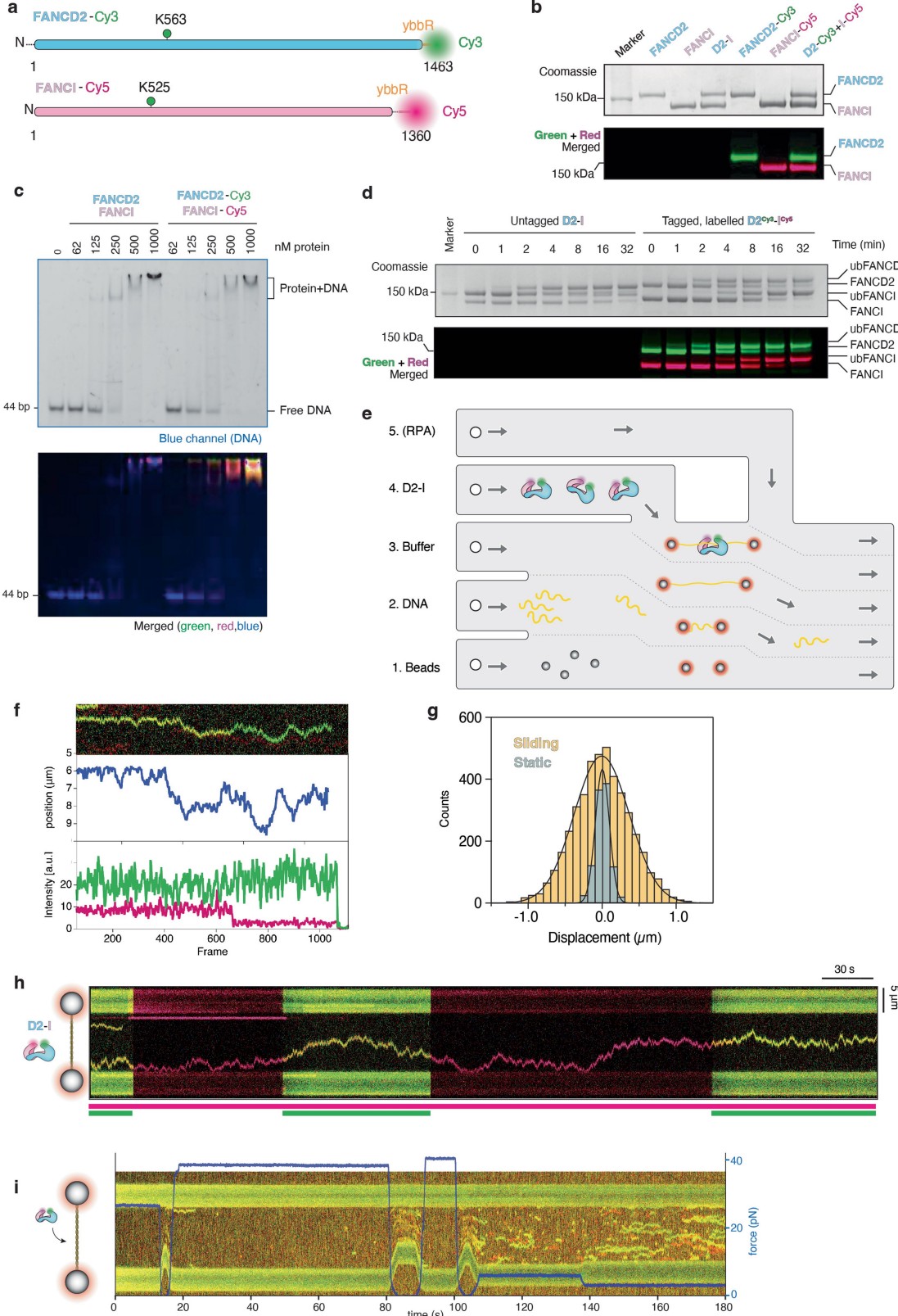

**Extended Data Fig. 1** | See next page for caption.

**Extended Data Fig. 1 | Preparation of labelled D2-I and single-molecule imaging of D2-I on DNA. a**, Schematic representation of the FANCD2-ybbR and FANCI-ybbR constructs used to generate doubly-labelled D2-I complexes. The fluorescent labels are indicated at the carboxyl termini of FANCD2-ybbR (Cy3) and FANCI-ybbR (Cy5). Residue numbers and the positions of the ubiquitinated lysines are indicated. **b**, SDS-PAGE of purified FANCD2, FANCI, D2-I, FANCD2-Cy3, FANCI-Cy5 and D2-Cy3 + I-Cy5. Top panel shows Coomassie-stained gel and bottom panel shows merged scan in the green (Cy3) and red (Cy5) channels. This is a representative gel of 5 independent preparations. **c**, Fluorescently-labelled D2-I binds to DNA with comparable affinity to non-labelled D2-I. FAM-labelled 44-bp dsDNA (20 nM) was incubated with increasing concentrations of either non-labelled or double-labelled D2-I for 20 min at 20 °C. Samples were run on 6% polyacrylamide gels in 0.5X TBE buffer at 4 °C for 60 min. The scan using the blue channel (FITC) is shown in the top panel and the overlay of the blue (FITC), green (Cy3) and red (Cy5) channels is shown in the bottom panel. These data are representative of experiments performed three times. **d**, Time-course ubiquitination assay of untagged D2-I and ybbR-tagged, fluorescently labelled D2-I. Ubiquitination reactions were set up as described in Methods, samples were taken at the indicated time points and loaded on a 3–8% NuPAGE Tris-Acetate gel (Invitrogen). Top panel shows Coomassie-stained gel and bottom panel shows merged scan in the green (Cy3) and red (Cy5) channels. This is a representative gel of an experiment repeated three times. For gel source data (panels b-d) see Supplementary Fig. 1. **e**, Schematic representation of the microfluidics chamber employed for single-molecule experiments. Arrows show flow directions. RPA was only used when stated. **f**, A cropped kymograph of a D2-I complex diffusing on λ DNA (top panel). The trajectory of the mobile molecule (middle) and the corresponding pixel intensity at each timeframe in the green and red channels (bottom) are shown. The photobleaching of a single double-labelled heterodimer occurs as a two-step event: Initially, Cy5 dye photobleaches (~650[th] frame) and it is followed by Cy3 dye photobleaching after ~1050 frames were acquired. **g**, Distribution of displacement per time window of 5 frames for one static and one stalled D2-I trace on DNA. The symmetric distribution reflects a random diffusion mechanism. **h**, Representative kymograph of D2-I sliding on λ DNA over several minutes. The colored bar below indicates the excitation laser (Cy5, red; Cy3, green). **i**, Relationship between force and DNA binding. Incubation at low forces allowed D2-I to readily bind to DNA, whereas high forces hindered DNA binding. Kymographs were normally collected at 15 pN (unless otherwise stated). Representative kymograph from three independent experiments.

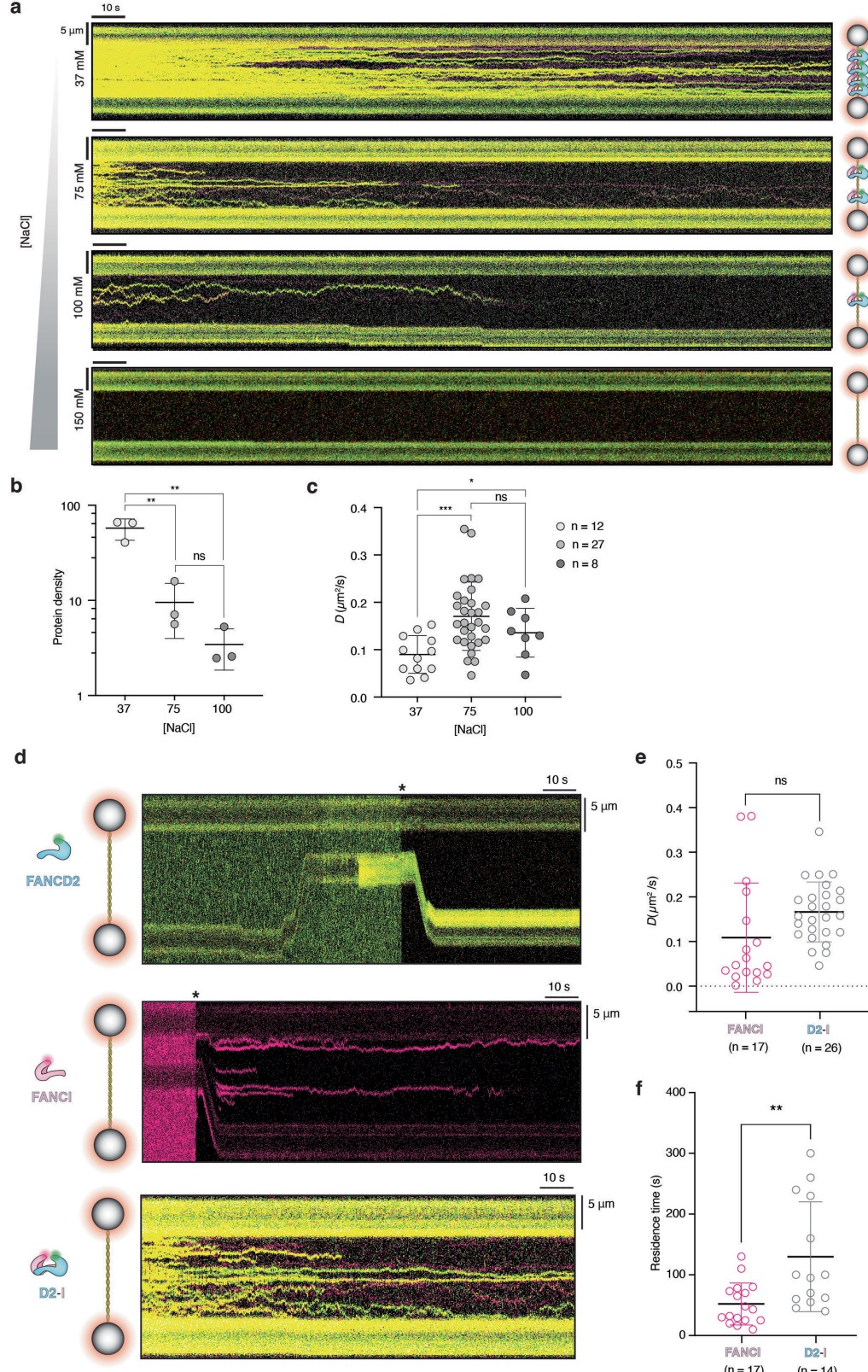

**Extended Data Fig. 2** | See next page for caption.

**Extended Data Fig. 2 | Effect of salt on D2-I binding and sliding, and single-molecule analysis of D2-I, FANCD2 alone and FANCI alone. a-b,** To estimate the efficiency of the DNA loading step, we analyzed the initial number of D2-I complexes bound to a single trapped DNA after incubation for 15 s in the protein channel. Representative kymographs of D2-I on λ DNA at different NaCl concentrations are shown in panel **a** and quantitated in panel **b.** The kymographs at 37 mM NaCl show more binding events than those at 75 mM NaCl. At 100 mM NaCl, D2-I shows even lower binding density and shorter traces, while very sparse binding or sliding was observed at 150 mM NaCl. In panel **b,** individual points show the number of observed traces per kymograph at each NaCl concentration. The mean, standard deviation and unpaired t-test pairwise comparisons are shown (n = 3). **, p = 0.006 (35 mM vs 75 mM); **, p = 0.003 (75 mM vs 100 mM); ns, p = 0.141. **c,** Comparison of the calculated diffusion coefficients (*D*) for D2-I at different NaCl concentrations. Individual values of *D* for each trace and the mean and standard deviation for each NaCl concentration are shown. Unpaired t-test pairwise comparisons showed that there is a statistically significant difference in the average *D* with 37 mM NaCl. However, this difference is small and likely a result of crowding on the DNA at 37 mM NaCl (see panel a). ***, p = 0.0008; *, p = 0.036; ns, not significant (p = 0.21 for 75 vs. 100 mM NaCl). Since the diffusion coefficient of D2-I is largely insensitive to changes in ionic strength, it exhibits a sliding diffusion mechanism, where the diffusing protein maintains constant contact with DNA[61]. In the case of a hopping mechanism where rapid dissociation and reassociation events take place, there would be more dramatic changes in the diffusion rate with varying salt concentrations[61-63]. **d,** DNA binding and sliding by FANCD2, FANCI and D2-I. *Top panel:* representative kymograph showing the absence of FANCD2 binding to λ DNA. Optically-trapped DNA was incubated in the protein channel containing 5 nM FANCD2 and then moved to the imaging channel. We observed only two traces in 25 kymographs. *Middle panel:* representative kymograph showing FANCI readily binding and sliding on trapped λ DNA (n = 17). *Bottom panel:* a representative kymograph of D2-I readily binding and sliding on trapped λ DNA (n = 26). Asterisks mark the change from protein to buffer channel. **e,** Distribution of the diffusion coefficients (*D*) of FANCI and D2-I. Individual values, mean, standard deviation and unpaired t-test pairwise comparisons are shown. **f,** Distribution of the observed residence times for FANCI and D2-I. Individual values, mean, standard deviation and unpaired t-test pairwise comparisons are shown. FANCI shows binding and sliding on DNA although with significantly shorter observed residence times (52 s average) compared to D2-I (129 s average). **, p = 0.0027; ns, not significant (p = 0.054).

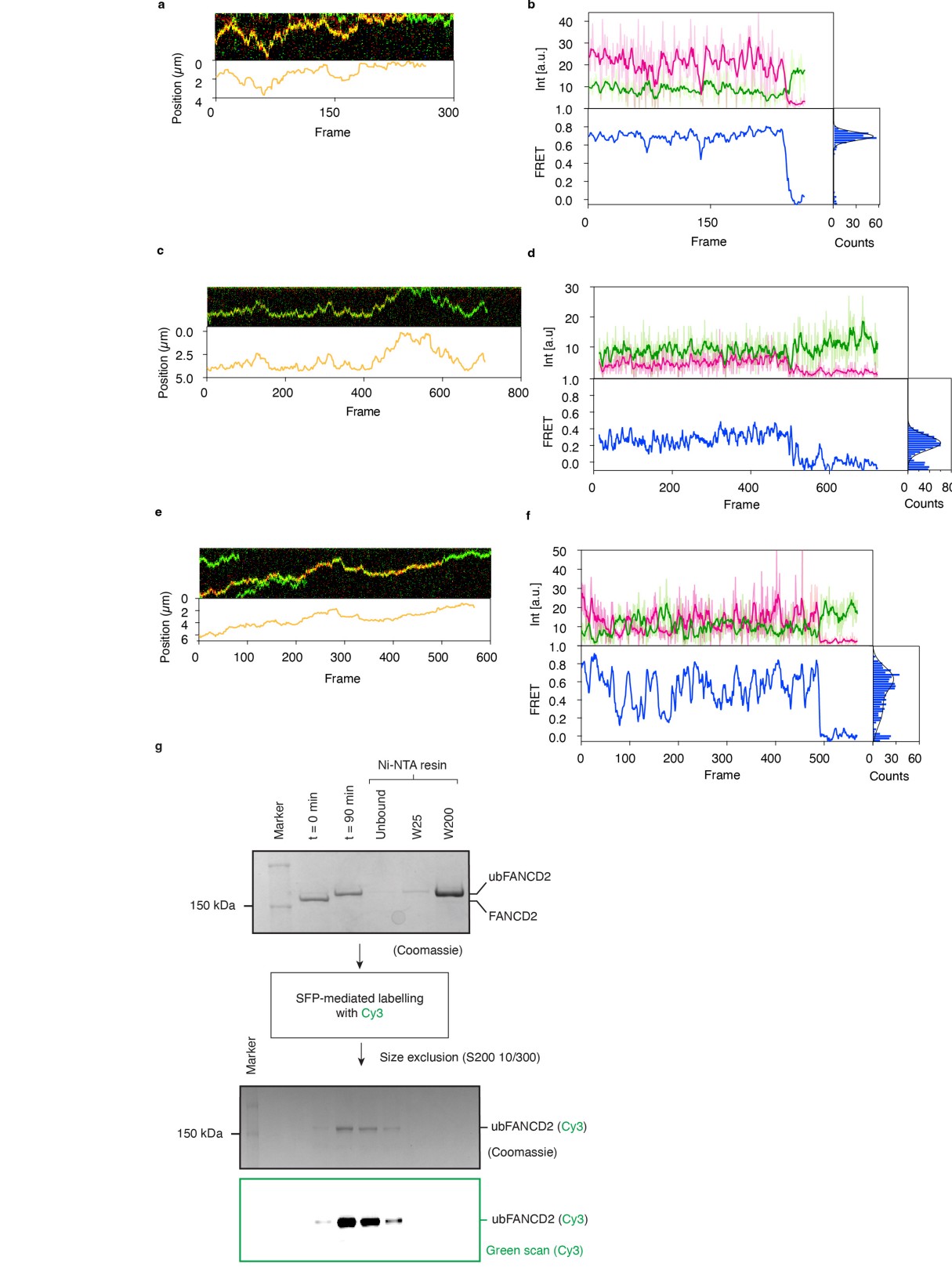

**Extended Data Fig. 3** | See next page for caption.

**Extended Data Fig. 3 | Single-molecule Fluorescence Resonance Energy Transfer (FRET) within diffusing D2-I complexes. a,c,e:** Cropped kymographs (top) and the corresponding trajectories (bottom) of selected D2-I complexes sliding on λ DNA. The images were recorded using only green laser excitation. **b,d,f:** Pixel intensity at each timeframe of the trajectory in the green and red channel (top) and the calculated FRET (bottom) with histograms of the FRET signal on the right. The complexes exhibit either high FRET (0.83 ± 0.02; panel b), low FRET (0.27 ± 0.01; panel d) or alternating FRET within a single trajectory (panel f). These likely represent predominantly closed, open or dynamic D2-I, respectively. FRET values are given as the peak centre of the FRET distribution ± S.E.M. **g**, SDS-PAGE analysis of the preparation of labelled monoubiquitinated FANCD2. FANCD2 was ubiquitinated with His-Ub in a 90 min reaction and purified on Ni-NTA resin (top). It was then labeled with Cy3 and purified using size exclusion chromatography (bottom). This was subsequently assembled with FANCI on DNA (see Fig. 2b). W25 and W200 indicate washes with 25 and 200 mM imidazole, respectively. Representative of an experiment performed independently three times. For gel source data, see Supplementary Fig. 1.

**a**   **No stalling of D2^WT-I at Holliday junctions (N = 10)**

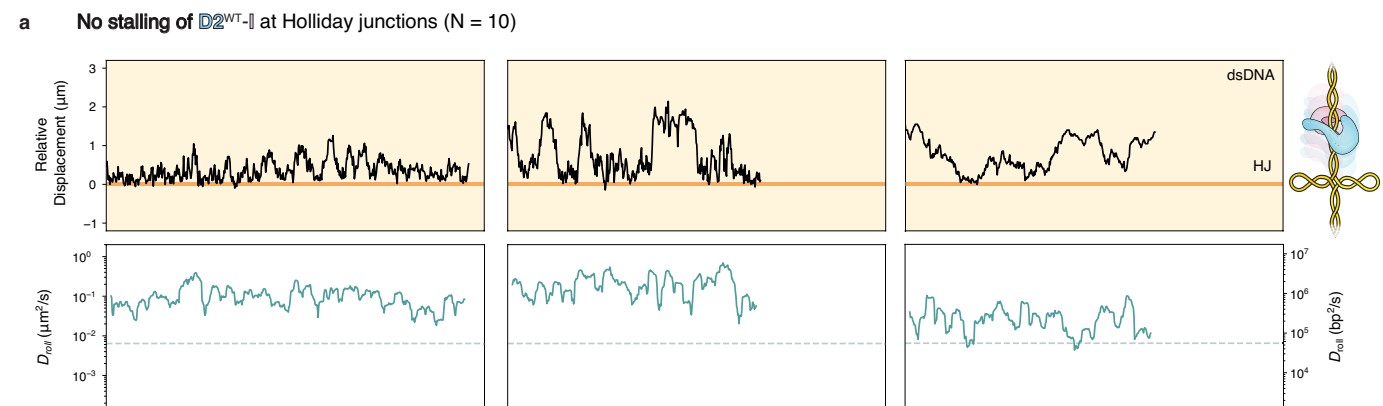

**b**

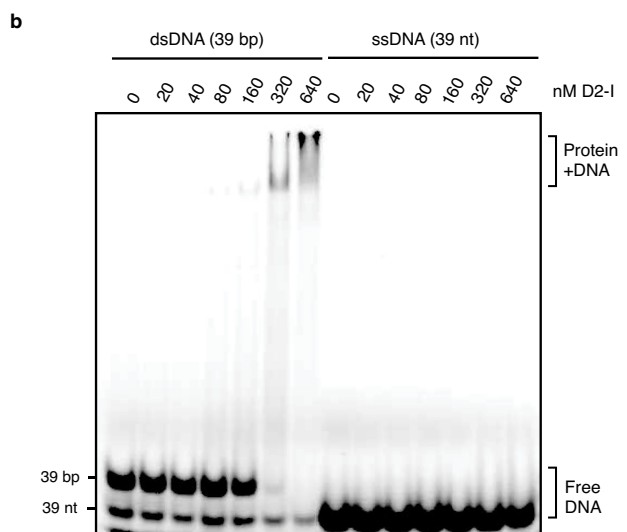

**Extended Data Fig. 4 | Analysis of D2-I on HJ DNA and ssDNA. a**, D2-I does not stall at HJ DNA. Rolling diffusion analysis of D2-I on HJ, showing that $D_{roll}$ stays above $D_{th}$ (dashed line) and is therefore not classified as stalled. Analysis for three separate traces representative of 10 kymographs are shown. **b**, D2-I binds to dsDNA (39 bp) but not to ssDNA (39 nt) on an electrophoretic mobility shift assay (EMSA). FAM-labelled dsDNA or ssDNA (20 nM) were incubated with increasing concentrations of D2-I for 20 min at 20 °C. Samples were run on 6% polyacrylamide gels in 0.5X TBE buffer at 4 °C for 60 min. These data are representative of experiments performed three times. For gel source data, see Supplementary Fig. 1.

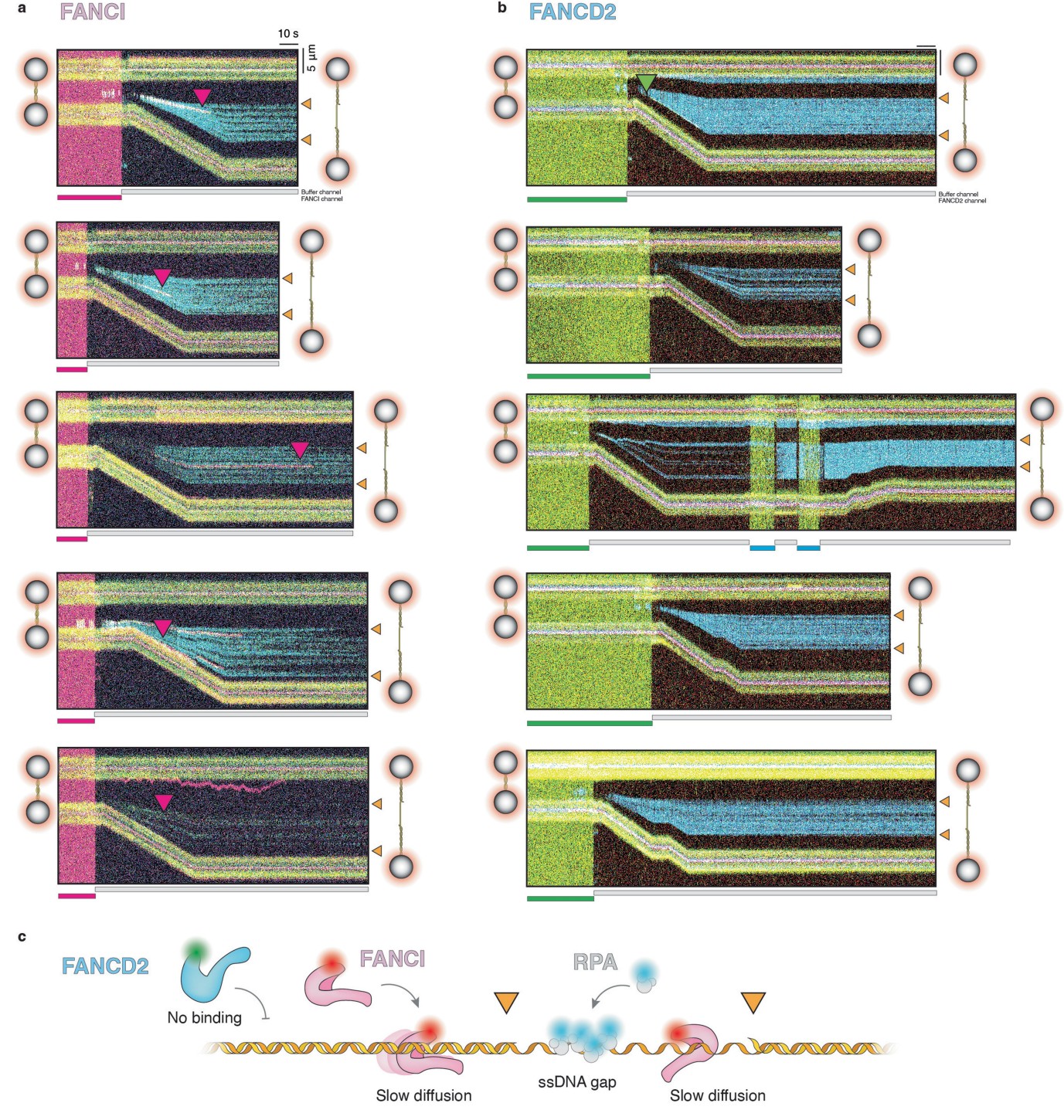

**Extended Data Fig. 5 | Analysis of FANCI and FANCD2 on λ DNA containing a defined ssDNA gap. a**, Representative kymographs from 10 independent experiments, showing FANCI on λ DNA containing a defined ssDNA gap. Red triangles indicate instances where FANCI is observed within the ssDNA gap. **b**, Representative kymographs from 10 independent experiments, showing FANCD2 on λ DNA containing a defined ssDNA gap. A small number of FANCD2 traces were observed on DNA (green triangle). In panels a and b, the colored bars indicate the channel position (grey=buffer channel, red=FANCI channel, green=FANCD2 channel, blue=RPA channel) and the orange triangles indicate the ss-dsDNA junctions. **c**, Schematic representation of FANCD2 and FANCI monomers and their interaction with DNA. FANCD2 is labelled in green, FANCI is in magenta and RPA is in blue.

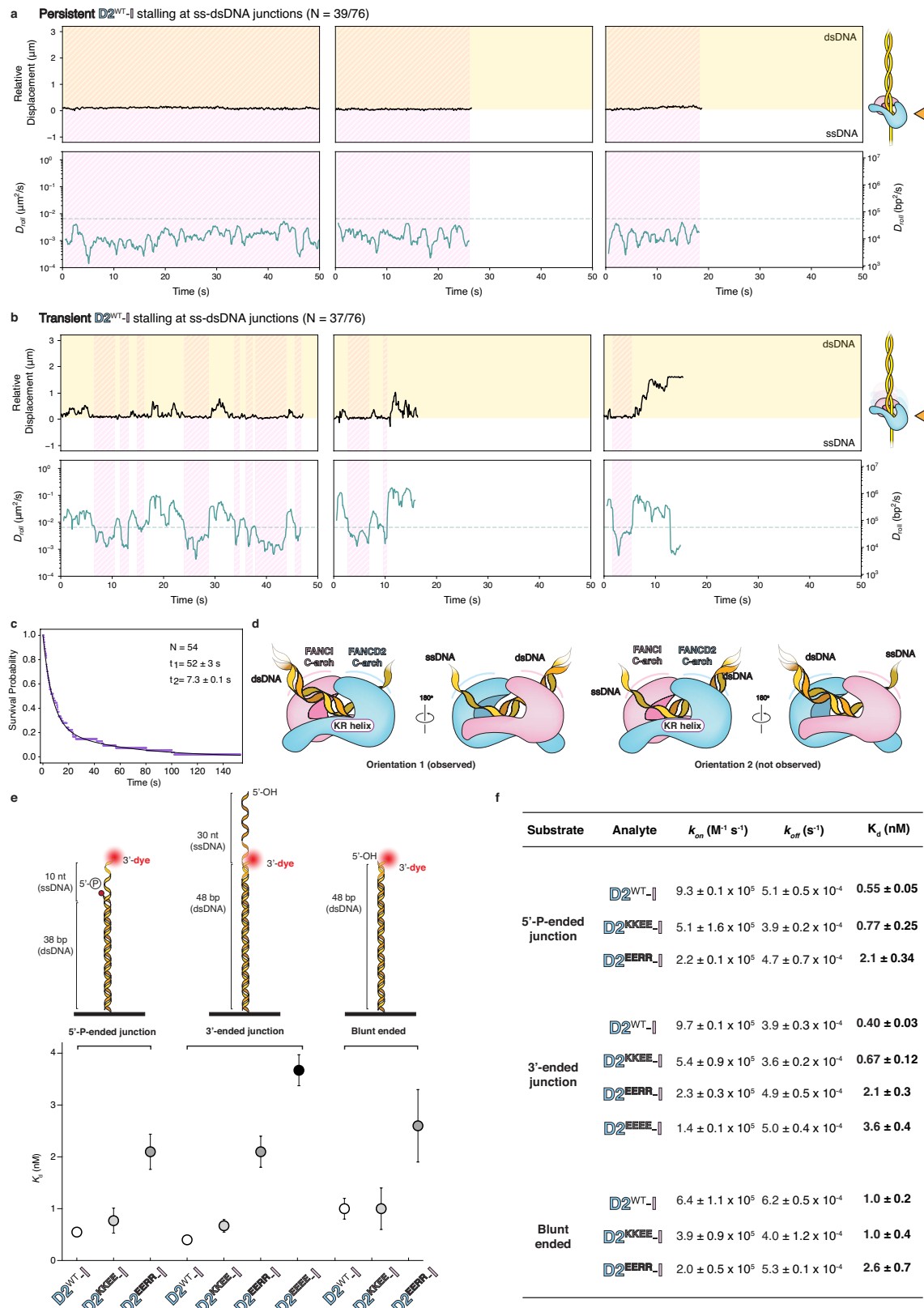

**Extended Data Fig. 6 |** See next page for caption.

**Extended Data Fig. 6 | Persistent and transient stalling of D2-I at ss-dsDNA junctions. a-b**, Rolling diffusion analysis of D2-I on dsDNA. D2-I tracks on dsDNA with a ssDNA gap (a–b, top) shows a rolling diffusion coefficient, $D_{roll}$ (below) less than the threshold of $6.4 \times 10^{-3} \, \mu m^2 s^{-1}$ (dashed line). These experiments reveal two different types of D2-I stalls. About half (n = 39/76 or 51%) of D2$^{WT}$-I molecules stalled at ss-dsDNA junctions are essentially immobile, remain static and do not diffuse away from the junction (a). We classify these as "persistent stalls". The remaining 49% of stalled complexes remained static for a shorter time, before diffusing away from the ss-dsDNA junction. We classify these as "transient stalls" (b). All stalls are shown with pink shading. Three representative traces are shown for each. **c**, Quantitative analysis of the lifetime of D2-I stalls. The survival probability plot for the distribution of D2-I dwells from 54 D2-I molecules (purple) along with the corresponding double exponential decay fit (black) shows that at least two kinetic phases were needed to describe the observed kinetics of D2-I stalling at junctions. The longer-lived dwells, which had a lifetime of $52 \pm 3$ s (which is likely to be photobleaching limited), could be attributed to persistent D2-I stalls, while the comparatively shorter-lived dwell, with a lifetime of $7.3 \pm 0.1$ s, could be attributed to transient D2-I stalls. Transient stalls may represent instances where D2-I molecules sense the ss-dsDNA junction but fail to engage stably. Error in lifetime is the error of the exponential fit to the survival probability. **d**, The reason that approximately half of the analysed molecules had persistent stalls and the other half had transient stalls is not clear, but one possibility is related to the orientation of D2-I when approaching the ss-dsDNA junction. D2-I moves bidirectionally on DNA, but the cryoEM structure of D2-I stalled at a ss-dsDNA junction is directional: In our structure, the dsDNA is always bound to the C-terminal arch of FANCI and not to the C-terminal arch of FANCD2 (orientation 1, observed). Since D2-I loads onto dsDNA in a random orientation, it could approach the ss-dsDNA junction with FANCD2 leading, or with FANCI leading. It is possible that approaching the junction with FANCD2 results in a persistent stall (as seen in cryoEM) whereas approaching the junction with FANCI (orientation 2, not observed) results in a transient stall that is unable to fully engage the KR helix. This hypothesis is consistent with approximately half of the stalls being persistent and half being transient. **e,f**, DNA binding kinetics of D2-I measured by SwitchSENSE. Mean values and error (standard deviation) are plotted (e) and their values shown in (f). All KR helix mutants have $K_d$s in the nanomolar regime, likely because they can bind dsDNA with their C-arches even if binding to ss-dsDNA junctions by the KR helix is impaired.

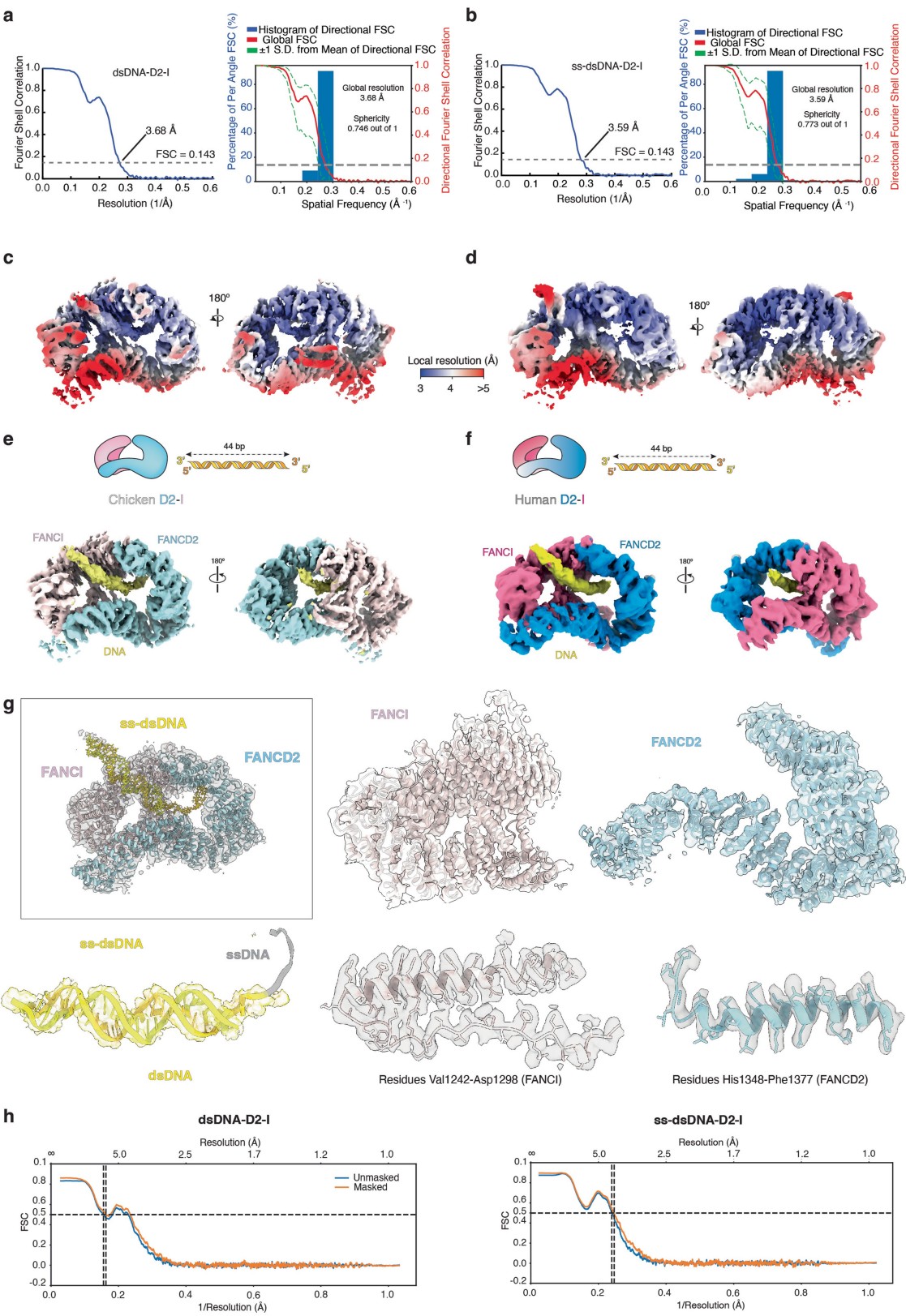

**Extended Data Fig. 7 | Resolution estimation and modelling into cryoEM maps. a-b**, Gold-standard Fourier shell correlation (FSC) curves (from RELION) and orientation distribution plot and sphericity (calculated with the 3DFSC server) for the final reconstructions of D2-I with dsDNA (a) and D2-I with ss-dsDNA (b). **c-d**, Unsharpened cryoEM maps of D2-I with dsDNA (c) and ss-dsDNA (d) colored by local resolution estimated in RELION. **e-f**, Comparison

of cryoEM structures of chicken (as in Fig. 4a) and human D2-I bound to a 44-bp dsDNA. **g**, Overall fit of model of D2-I bound to ss-dsDNA into the sharpened cryoEM map (boxed) and detailed fit of FANCI, FANCD2, DNA and representative regions. **h**, Map-to-model FSC for the dsDNA-D2-I and ss-dsDNA-D2-I structures, calculated in Phenix.

**a**  Unsharpened dsDNA-D2-I map

DNA
FANCI    FANCD2    DNA

dsDNA fit into unsharpened map

dsDNA

↓ Map sharpening
(DeepEMhancer)

**Sharpened dsDNA-D2-I map**

DNA
FANCI    FANCD2    DNA

dsDNA overlayed onto fragmented DNA density of
sharpened map due to conformational heterogeneity

dsDNA

**b**  Unsharpened ss-dsDNA-D2-I map

DNA
FANCI    FANCD2    DNA

ss-dsDNA fit into unsharpened map

dsDNA    Junction    ssDNA

↓ Map sharpening
(DeepEMhancer)

**Sharpened ss-dsDNA-D2-I map**

DNA
FANCI    FANCD2    DNA

ss-dsDNA fit into sharpened map

dsDNA    Junction    ssDNA

**c**  Focused classification on DNA region of dsDNA-D2-I map

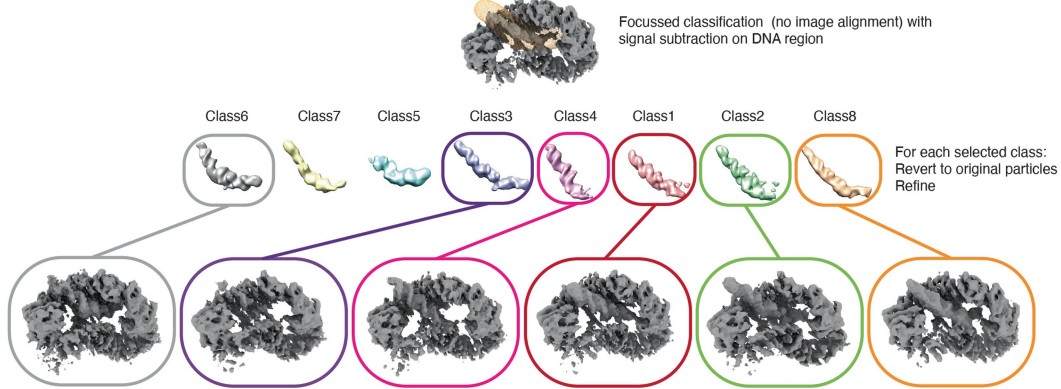

Focussed classification (no image alignment) with
signal subtraction on DNA region

Class6   Class7   Class5   Class3   Class4   Class1   Class2   Class8

For each selected class:
Revert to original particles
Refine

**Extended Data Fig. 8** | See next page for caption.

**Extended Data Fig. 8 | Comparison of maps of D2-I bound to dsDNA or ss-dsDNA. a-b**, CryoEM maps (unsharpened and sharpened with DeepEMhancer) of D2-I bound to dsDNA (a) and ss-dsDNA (b). The map of D2-I bound to dsDNA shows a blurry, undefined density in the DNA region, which is consistent with different translational positions while sliding with respect to D2-I. Indeed, the DNA density becomes fragmented after sharpening with a single B-factor (RELION postprocessing) or locally scaling the sharpening with DeepEMhancer. In contrast, the DNA density in the map with ss-dsDNA is well defined, owing to the ss-dsDNA being predominantly constrained to a single position after D2-I stalling on the ss-dsDNA junction (marked with an orange triangle). **c**, Image processing pipeline of signal subtraction and 3D classification without image alignment for the DNA region of the dsDNA-D2-I map. The best classes based on DNA density quality and number of particles were selected (colored circles) and refined separately, showing dsDNA binding to D2-I at different positions. Interestingly, class 4 shows that the blunt dsDNA end can contact the KR helix in a similar mechanism to ss-dsDNA junction. However, this class accounts for only 14% of the total number of particles subjected to this 3D classification. Moreover, all published structures of D2-I with DNA contain blunt dsDNA termini but they never show a preferred recognition of the end. Therefore, unlike ss-dsDNA, dsDNA is more likely to contact the C-arches of FANCI and FANCD2 rather than the C-arch of FANCI and the KR helix.

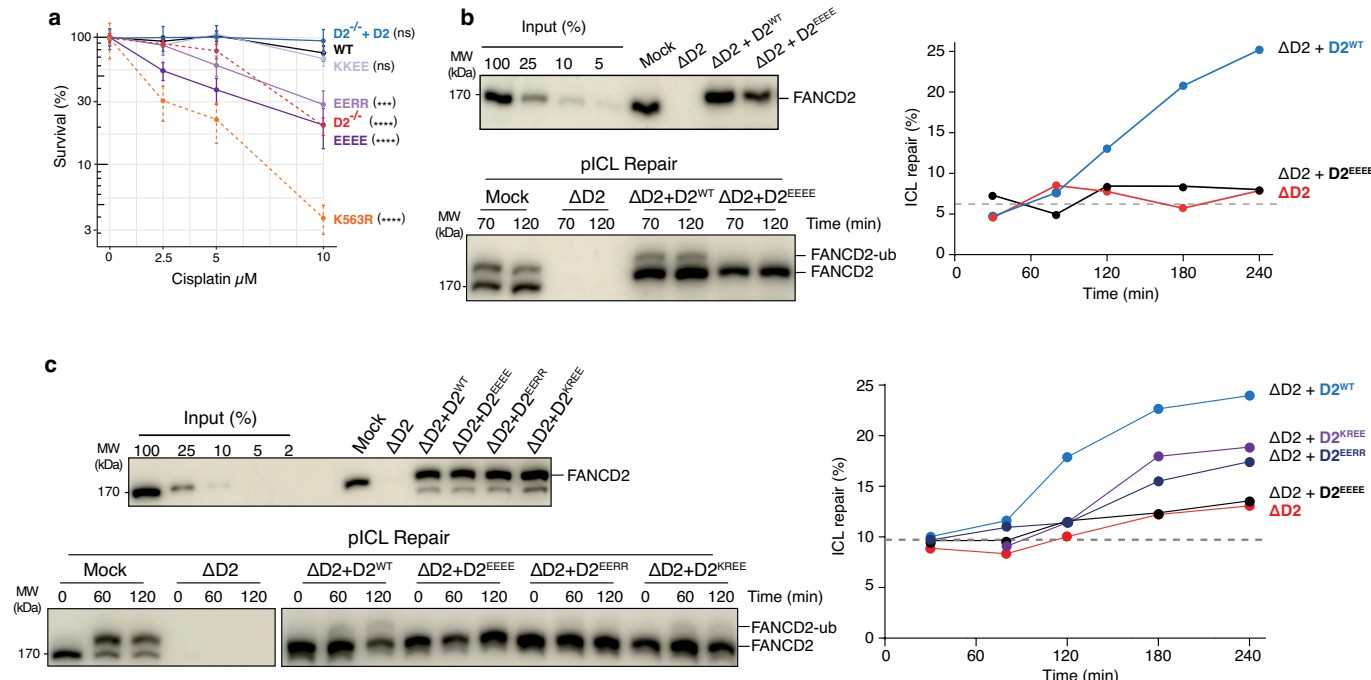

**Extended Data Fig. 9 | Cisplatin sensitivity assay in DT40 cells and ICL repair in *Xenopus* egg extracts. a**, Clonogenic survival assay assessing cisplatin sensitivity of DT40 FANCD2$^{-/-}$ cells complemented with either FANCD2$^{WT}$ (WT), FANCD2$^{K563R}$ (ubiquitination deficient), or KR helix variants (KKEE, EERR, EEEE). Results are based on 2 independent clones of each mutant and 6 replicates per clone, with mean and standard deviation plotted. P-values enclosed in brackets are for samples compared to WT (ns = not significant, **** P < 0.000001; see Supplementary Table 2 for all p values). The arginine mutants (KKEE and KKAA) fully rescue cisplatin sensitivity but the lysine mutants (EERR and AARR) do not (see also Fig. 6a). This is consistent with the lysine mutants having a stronger effect on stalling (Fig. 5c). However, since the EEEE and AAAA mutants show high sensitivity to cisplatin (compare EEEE to EERR and KKEE), it is likely that all four residues contribute to ss-dsDNA junction binding. Interestingly, cell lines expressing alanine substitutions were more sensitive to cisplatin than glutamic acid substitutions. We hypothesize that the glutamic acid substitutions do not bind DNA efficiently in cells and therefore act as nulls (like D2$^{-/-}$) whereas the alanine substitutions act as dominant negatives (like FANCD2$^{K563R}$) that bind DNA but are non-functional. For the colony survival assay, three cell dilutions (1:1, 1:10, and 1:100) were plated in duplicate for each cisplatin concentration. This resulted in a maximum of six replicates per concentration. Additionally, two clones were tested for each mutant. This brought the theoretical maximum number of observations to N = 12 for each cisplatin concentration, including the untreated control. However, some conditions were excluded due to

limitations in counting at very high or very low colony numbers. For example, the WT DT40 at dilution 1:1 (untreated) was too confluent for accurate counting, and no colonies were found in the replicates of DT40 K563R at a 10 μM cisplatin for cell dilution of 1:100. Consequently, the final number of observations for these examples is N = 8. Statistical testing was performed by the R package CFAssay using two way analysis of the cell survival data by fitting a linear quadratic model with maximum likelihood (all values reported in Supplementary Table 2). **b**, ICL repair in *Xenopus* egg extracts. Mock-depleted (Mock), and FANCD2-depleted (ΔD2) NPE complemented with wild-type FANCD2 (ΔD2 + D2$^{WT}$) or mutant FANCD2 (ΔD2 + D2$^{EEEE}$), were analyzed by western blot using α-FANCD2 antibody (top left blot). These extracts were used to replicate pICL. Absolute ICL repair efficiency was calculated and plotted (right panel). Dotted line indicates SapI fragments from contaminating uncrosslinked plasmid present in pICL preparations. Total extract samples were collected at indicated timepoints and analyzed by western blot using α-FANCD2 (bottom left blot). Note that the basic KR helix residues tested in the chicken system (KKRR) are KRRR in *xl*FANCD2. Representative of 2 independent experiments (n = 2). **c**, Repeat of experiment shown in panel b and Fig. 6b including all FANCD2 mutants (ΔD2 + D2$^{EEEE}$, ΔD2 + D2$^{EERR}$ and ΔD2 + D2$^{KREE}$). Note that the 30-minute timepoint of ΔD2 + D2$^{KREE}$ is not shown due to poor digestion of this sample; we can assume that no repair has taken place at 30 min since this is the case at 60 min. Representative of 2 independent experiments (n = 2). For gel source data see Supplementary Fig. 1.

**Extended Data Table 1 | Structural data statistics**

| | Sliding D2-I on dsDNA (EMDB-50355) (PDB 9FFF) | Stalled D2-I on ss-dsDNA (EMDB-50353) (PDB 9FFB) |
|---|---|---|
| **Data collection and processing** | | |
| Magnification | 105,000 X | 105,000 X |
| Voltage (kV) | 300 | 300 |
| Electron exposure (e–/Å²) | ~40 | ~40 |
| Defocus range (µm) | -1.2 to -2.8 | -1.2 to -2.8 |
| Pixel size (Å) | 0.825 | 0.831 |
| Symmetry imposed | none | none |
| Initial particle images (no.) | ~1,900,000 | ~2,400,000 |
| Final particle images (no.) | 165,469 | 319,378 |
| Map resolution (Å) FSC threshold | 3.68 | 3.59 |
| Map resolution range (Å) | 0.143 | 0.143 |
| | 3.68 to >10 | 3.59 to >10 |
| **Refinement** | | |
| Initial model used (PDB code) | 8A2Q | 8A2Q |
| Model resolution (Å) FSC threshold | 3.4 | 3.4 |
| Model resolution range (Å) | 0.143 | 0.143 |
| Map sharpening $B$ factor (Å²) | -90 | -120 |
| Model composition Non-hydrogen atoms Protein residues | 15045 2069 | 17285 2067 |
| $B$ factors (Å²) | | |
| Protein | 151.42 | 154.87 |
| DNA | 448.32 | 410.35 |
| R.m.s. deviations | | |
| Bond lengths (Å) | 0.003 | 0.013 |
| Bond angles (°) | 0.640 | 0.833 |
| Validation | | |
| MolProbity score | 1.90 | 2.16 |
| Clashscore | 11.74 | 17.34 |
| Poor rotamers (%) | 0.50 | 0.16 |
| Ramachandran plot | | |
| Favored (%) | 95.51 | 93.51 |
| Allowed (%) | 4.39 | 6.44 |
| Disallowed (%) | 0.10 | 0.00 |
| Map-model correlation coefficients | | |
| CCmask | 0.72 | 0.82 |
| CCbox | 0.83 | 0.87 |

Statistics for sliding D2–I on dsDNA and stalled D2–I on ss–dsDNA cryoEM data.

# Reporting Summary

## Statistics

For all statistical analyses, confirm that the following items are present in the figure legend, table legend, main text, or Methods section.

| n/a | Confirmed | |
|---|---|---|
| ☐ | ☒ | The exact sample size (*n*) for each experimental group/condition, given as a discrete number and unit of measurement |
| ☐ | ☒ | A statement on whether measurements were taken from distinct samples or whether the same sample was measured repeatedly |
| ☐ | ☒ | The statistical test(s) used AND whether they are one- or two-sided<br>*Only common tests should be described solely by name; describe more complex techniques in the Methods section.* |
| ☒ | ☐ | A description of all covariates tested |
| ☒ | ☐ | A description of any assumptions or corrections, such as tests of normality and adjustment for multiple comparisons |
| ☐ | ☒ | A full description of the statistical parameters including central tendency (e.g. means) or other basic estimates (e.g. regression coefficient) AND variation (e.g. standard deviation) or associated estimates of uncertainty (e.g. confidence intervals) |
| ☐ | ☒ | For null hypothesis testing, the test statistic (e.g. *F*, *t*, *r*) with confidence intervals, effect sizes, degrees of freedom and *P* value noted<br>*Give P values as exact values whenever suitable.* |
| ☒ | ☐ | For Bayesian analysis, information on the choice of priors and Markov chain Monte Carlo settings |
| ☒ | ☐ | For hierarchical and complex designs, identification of the appropriate level for tests and full reporting of outcomes |
| ☒ | ☐ | Estimates of effect sizes (e.g. Cohen's *d*, Pearson's *r*), indicating how they were calculated |

*Our web collection on statistics for biologists contains articles on many of the points above.*

## Software and code

Policy information about availability of computer code

| Data collection | EPU version 3.4.0 (Thermo Fisher Scientific) was used for all cryo electron microscopy data collection. A Typhoon Imaging System (GE Healthcare) or Gel Doc XR+ system (Bio-Rad) was used for gel imaging. Single-molecule data was collected using the commercial software Bluelake v. 1.6 (LUMICKS) that is fully compatible with the instrument (correlative optical tweezer and confocal microscope C-TRAP). Measurements of individual DNA molecules were saved as .H5 Hierarchical Data Format. All H5-files contain force and position data of optically trapped beads, as well as pixel values of the recorded 3-color confocal images. The exported files contain additional meta-data such as: experimental description, status of the microfluidic system, laser powers, laser coordinates, camera settings. |
|---|---|
| Data analysis | Relion v4.0, MotionCor2, CTFFIND4, crYOLO 1.7.6, 3DFSC, Coot, Phenix 1.20.1, UCSF Chimera 1.15, ChimeraX-1.16.1, ISOLDE, DeepEMhancer, Prism 10, switchANALYSIS 1.9.0.33 were used for data analysis.<br>Single-molecule  data were processed and analyzed using custom-made scripts written in Python v. 3.9 using Pylake 1.2.1/Numpy 1.26.0/ Matplotlib 3.7.2/Scipy 1.11.3/Peakutils 1.3.4. packages<br>Each H5-file was processed in a separate Jupyter Notebook - an interactive platform to visualize the results of the Python script. Within a single Jupyter Notebook, the fluorescent data was rendered into an RGB image, was correlated with the force measurement, subsequently underwent the single-particle tracking and mean-square-displacement (MSD) analysis. A representative Jupyter Notebook that includes all the above features is available at www.github.com/singlemoleculegroup.<br>Selected images rendered in Jupyter Notebook into .TIFF images were cropped in ImageJ 2.1<br>Commercial software Wavemetrics IGOR 8 was used to generate final plots (force-distance curves, 1D trajectories, MSD plots and histograms). To do so, the output of each Jupyter Notebook (in the .CSV format) was imported to IGOR's workspace. |

For manuscripts utilizing custom algorithms or software that are central to the research but not yet described in published literature, software must be made available to editors and reviewers. We strongly encourage code deposition in a community repository (e.g. GitHub). See the Nature Portfolio guidelines for submitting code & software for further information.

# Data

Policy information about <u>availability of data</u>

All manuscripts must include a <u>data availability statement</u>. This statement should provide the following information, where applicable:
- Accession codes, unique identifiers, or web links for publicly available datasets
- A description of any restrictions on data availability
- For clinical datasets or third party data, please ensure that the statement adheres to our <u>policy</u>

The model of FANCD2 in complex with FANCI (PDB 6TNG) was used as initial reference for model building into cryoEM maps. The models and maps generated in this study have been deposited to PDB and EMDB and assigned the following accession codes: PDB ID 9FFF, EMD-50355 for dsDNA-D2-I; PDB ID 9FFB, EMD-50353 for ss-dsDNA-D2-I. All original gels shown in this study are provided as Source Data. Other raw data is accessible on Zenodo at DOI: 10.5281/zenodo.11521474. Code for analysis of single molecule data are freely available (https://github.com/singlemoleculegroup). Correspondence and requests for materials should be addressed to L.A.P. or D.S.R.

# Research involving human participants, their data, or biological material

Policy information about studies with <u>human participants or human data</u>. See also policy information about <u>sex, gender (identity/presentation), and sexual orientation</u> and <u>race, ethnicity and racism</u>.

| | |
|---|---|
| Reporting on sex and gender | Not applicable |
| Reporting on race, ethnicity, or other socially relevant groupings | Not applicable |
| Population characteristics | Not applicable |
| Recruitment | Not applicable |
| Ethics oversight | Not applicable |

Note that full information on the approval of the study protocol must also be provided in the manuscript.

# Field-specific reporting

Please select the one below that is the best fit for your research. If you are not sure, read the appropriate sections before making your selection.

☒ Life sciences          ☐ Behavioural & social sciences          ☐ Ecological, evolutionary & environmental sciences

For a reference copy of the document with all sections, see <u>nature.com/documents/nr-reporting-summary-flat.pdf</u>

# Life sciences study design

All studies must disclose on these points even when the disclosure is negative.

| | |
|---|---|
| Sample size | Sample sizes were selected based on previous experience and published studies to evaluate reproducibility of assays. For quantified results, the number of replicates is indicated in the Figure legends. The number of micrographs in our cryoEM data collection was chosen accordingly to obtain the required resolution. |
| Data exclusions | 1) Kymographs that contained too many diffusing species were excluded from quantifications because the protein molecules were colliding with each other, obscuring the results of a single-particle tracking algorithm.<br>2) Trajectories of protein diffusing on the DNA were not included in the statistics if the diffusion was shorter than 5 seconds (not enough datapoints to fit linear mean-square-displacement).<br>3) For the colony survival assay (DT40), some conditions were excluded due to limitations in counting at very high or very low colony numbers. For example, the WT DT40 at dilution 1:1 (untreated) was too confluent for accurate counting, and no colonies were found in the two biological replicates of DT40 K563R at a 10 µM cisplatin for cell dilution of 1:100. Consequently, these points were excluded.<br>4) For the Xenopus egg extract experiment in Extended Data Fig 9c, the 30-minute timepoint of ΔD2+D2KREE is not shown due to poor digestion of this sample |
| Replication | All protein expression, purification, labelling, monoubiquitination assays and DNA binding essays, were performed at least three times, as indicated in the text. CryoEM data sets for dsDNA-D2-I and ss-dsDNA-D2-I were collected and processed two or three times, respectively, obtaining similar results. The largest data set for each complex was selected for obtaining the final reconstructions.<br>For single-molecule imaging, on each experimental day, we captured ~10-30 DNA molecules (one by one) that had a proper contour length (16.5 um for lambda DNA) and appropriate mechanical properties (persistence length, stretch modulus characteristic to dsDNA). Each experiment was replicated in at least 3 independent sessions. The resulting force-distance curves are practically identical, and are easily reproducible. The number of replicates is indicated for each experiment. |
| Randomization | Randomization is not relevant to the assays and biochemical experiments performed in this study. For experiments in Xenopus egg extracts, |

| Randomization | we subjected common extracts to the treatments indicated in the text. Cellular studies were conducted on cells from common pools that were treated with increasing amounts of DNA-damaging agent. For biochemical assays, the same protein stock was used for a given experiment. For calculation of the Fourier Shell Correlation using RELION, particles were automatically split into two random halves by the software. Replicates of all experiments show the results are reproducible and are not subjected to researcher's bias. |
| --- | --- |
| Blinding | Blinding is not relevant to the experiments presented in this study. Cryo-EM, biochemical data and cellular data were collected and processed identically under the same experimental conditions in an unbiased manner, and sample information did not lead to bias on any sample during the analysis. |

# Reporting for specific materials, systems and methods

We require information from authors about some types of materials, experimental systems and methods used in many studies. Here, indicate whether each material, system or method listed is relevant to your study. If you are not sure if a list item applies to your research, read the appropriate section before selecting a response.

## Materials & experimental systems

| n/a | Involved in the study |
| --- | --- |
| ☐ | ☒ Antibodies |
| ☐ | ☒ Eukaryotic cell lines |
| ☒ | ☐ Palaeontology and archaeology |
| ☐ | ☒ Animals and other organisms |
| ☒ | ☐ Clinical data |
| ☒ | ☐ Dual use research of concern |
| ☒ | ☐ Plants |

## Methods

| n/a | Involved in the study |
| --- | --- |
| ☒ | ☐ ChIP-seq |
| ☒ | ☐ Flow cytometry |
| ☒ | ☐ MRI-based neuroimaging |

## Antibodies

| Antibodies used | Histone H3 (ab1791, Abcam) used at 1:4000 dilution.<br>xlFANCD2 against N-terminal peptide (1-172aa) of xlFANCD2, described in https://doi.org/10.1016/ j.cell.2008.08.030. For western blot we used the serum as described in the reference (used in a 1:6000 dilution). For immundepletions we affinity purified the serum. This was a custom antibody raised at the Pocono Rabbit farm and Laboratory and a gift of the Walter lab (Harvard Medical School). |
| --- | --- |
| Validation | The Histone H3 antibody is a commercial antibody validated for Xenopus laevis Histone H3.<br>The xlFANCD2 antibody was previously validated in Xenopus egg extract (Knipscheer et al. Science 2009). |

## Eukaryotic cell lines

Policy information about cell lines and Sex and Gender in Research

| Cell line source(s) | Sf9, Oxford Expression Technologies Ltd, Cat No. 600100.<br><br>DT40 is an avian leukosis virus (ALV) induced bursal lymphoma cell line derived from a Hyline SC chicken. Obtained from Dr John Young at the Institute of Animal Health, Compton, Berkshire UK. Also commercially available at https://www.atcc.org/products/crl-2111 |
| --- | --- |
| Authentication | No authentication of the Sf9 cell line was performed.<br>DT40 cell line was authenticated by Bu-1a and b expression and immunoglobulin gene sequences. |
| Mycoplasma contamination | All cell lines were tested for mycoplasma. No mycoplasma contamination was detected. |
| Commonly misidentified lines (See ICLAC register) | No commonly misidentified cell lines were used in this study. |

## Animals and other research organisms

Policy information about studies involving animals; ARRIVE guidelines recommended for reporting animal research, and Sex and Gender in Research

| Laboratory animals | Female Xenopus laevis frogs used in this study were older than 2 years and obtained from Nasco |
| --- | --- |
| Wild animals | The study did not involve wild animals. |
| Reporting on sex | Xenopus egg extracts were produced from the eggs of Xenopus laevis frogs and therefore this study only involved the use of female frogs. |
| Field-collected samples | This study did not involve field-collected samples. |

Ethics oversight

All animal procedures and experiments were performed in accordance with national animal welfare laws and were reviewed by the Animal Ethics Committee of the Royal Netherlands Academy of Arts and Sciences (KNAW). All animal experiments were conducted under a project license granted by the Central Committee Animal Experimentation (CCD) of the Dutch government and approved by the Hubrecht Institute Animal Welfare Body (IvD), with project license number AVD80100202216633.

Note that full information on the approval of the study protocol must also be provided in the manuscript.

