## [Peer Review File · Nature]

Manuscript Title: FANCD2-FANCI surveys DNA and recognizes double to single-stranded junctions

Reviewer Comments & Author Rebuttals

Reviewer Reports on the Initial Version:

Referees' comments:

Referee #1:

In this study, Alcon et al. investigate how the FANCD2-FANCI (D2-I) protein complex executes two critical functions, the recognition of DNA inter-strand crosslinks and the protection of stalled replication forks. The authors employ single-molecule imaging to investigate DNA binding by D2-I and find that the complex acts as a sliding clamp on double-stranded DNA, but then stalls upon encountering single-stranded–double-stranded (ss-ds) DNA junctions. Such junctions arise at DNA inter-strand crosslinks during replication and are also present at stalled replication forks. Therefore, the authors data suggest a mechanism, wherein D2-I specifically clamps onto ss-dsDNA junctions to identify sites of DNA damage, offering a unified explanation for its various functions. The other proceed by investigating the mechanistic basis of how D2-I binds to ss-ds DNA-junctions using cryoEM and reveal specific interactions between D2-I and the single-stranded portion of the junction. Based on this structural information, the authors generated D2-I mutant variants with amino acid replacements in the ssDNA interface, which were however only analysed superficially.

Overall, this is a great manuscript, and I am highly enthusiastic about it. The writing is clear and thoughtful. The proposed model consolidates apparently different functions of D2-I into a coherent mode-of-action. However, while the model is certainly elegant, it unfortunately remains untested. But once my comments have been addressed, I think this manuscript will be a great candidate for publication in Nature.

Major points

Guided by their structural analysis, the authors generated two mutant variants of FANCD2 in which key basic residues within the ssDNA-binding interface were replaced. Two sets of EMSA experiments are conducted with these mutants. The authors conclude that the mutant variants show a specific reduction in affinity towards ss-ds DNA-junctions. I cannot agree with this conclusion based on the presented gels and the rather high error bars in their quantifications do not provide confidence in these results as well. I suggest that the authors employ these mutant variants to actually test their model, first in vitro using their beautiful single-molecule assays and then in physiologically relevant model systems, cells and frog egg extracts. Only this way can the authors test whether the proposed recognition ss-ds DNA-junctions has any role in DNA inter-strand crosslink repair and replication fork protection.

1. Do the D2-I mutant variants (KKAA and AARR) fail to stall at ss-ds DNA-junctions in single molecule experiments? Is their residence time on dsDNA affected?
2. Can the sensitivity to crosslinking agents and the loss of fork protection in FANCD2 KO cells be rescued by expression of the KKAA and AARR mutant variants? Ideally, using ubiquitylation-defective FANCD2 and WT as controls.
3. Are these mutant variants supporting inter-strand crosslink repair in frog egg extracts upon depletion of endogenous FANCD2? If repair fails, at what step of the FA pathway?
4. The authors speculate that "By simply physically holding the duplex together at ss-dsDNA junctions, it could prevent the double-stranded region from fraying and from further resection." This is an interesting thought that should be directly tested. Does D2-I protect DNA junctions against resection by an exonuclease in in vitro assays? If yes, is this ability lost in KKAA and AARR mutant variants?
5. The authors show that FANCI also binds DNA by itself. The authors' model predicts that FANCI would not stall at junctions in the absence of FANCD2. Is this the case? Vice versa, FANCD2 is shown to not bind dsDNA on its own. However, does it bind to junctions?

Minor points

1. Ext. Data Fig. 1d: Immuno-blots for FANCI and FANCD2 should be used to determine whether there is any difference in ubiquitylation. The Coomassie-stained gel may suggest that there is one, but it is difficult to judge.
2. Does D2-I show any preference for 5' or 3' ends at the junction?
3. Page 4, "Photobleaching was observed as a single-step event, consistent with the analyzed trajectories comprising single D2-I heterodimers (Extended Data Fig. 2b). Therefore, D2-I moves along the DNA duplex via thermal-energy driven random one-dimensional diffusion.": I suggest rephrasing, as the concluding sentence seems to reference information discussed above rather than the preceding sentence.
4. Fig. 3b: One FANCI molecule binds to the ssDNA region. How often are such events observed?
5. Page 5, "Due to random photobleaching events, a given D2-I particle may have green fluorescence, red fluorescence or both. This allowed us to differentiate complexes from each other.": Could red fluorescent particles also represent FANCI homodimers?
6. The number of replicates performed per experiment is sometimes mentioned in the respective figure legend but is not always provided.

Referee #2:

The authors present a series of experiments showing that the FANCD1-FANCI (D2-I) complex is capable of detecting and binding to discontinuities in the DNA backbone, specifically double-strand to single-strand transitions. They present a comprehensive set of single-molecule experiments using a Lumicks C-TRAP, supporting this hypothesis. Generally, the manuscript is well written, with a logical structure and clear narrative. There are no major concerns with the authors' findings, but some additional detail and clarification is required (see points below), particularly with respect to the mode of DNA-interaction.

MAJOR:

From the main text of the manuscript and accompanying figures/illustrations, it is not clear as the nature of the ds-ssDNA junction; i.e. does it represent a 3'- or 5'-recessed end? Does FANCD1-FANCI require a 3'-OH, or 5'-P for binding (depending on the overhang/recess)? This aspect of the D2-I / DNA interaction should be formally demonstrated.

Electrophoretic Mobility Shift Assays (EMSA); Ext. Data Figs. 6 and 10:

A significant amount of the D2-I complex, at higher concentrations, does not migrate into the gel, signifying either that: a higher-order complex is formed; there are significant "end-binding effects"; or the complex has precipitated out of solution. As this can affect the analysis and interpretation of nucleic-acid interactions they should be mitigated, either by a change in protocol or use of an alternate methodology (for example, fluorescence polarisation, SPR or similar).

The effects of the D2KKAA-I and D2AARR-I mutation sets on binding to ss-dsDNA require revisiting, as the data, whilst indicative, is not conclusive — especially given the very wide error bars at higher protein concentrations (see comments above). Ideally, values for K_d should be measured and reported. Contrary to the authors' suggestion that the 'smears' observed in the EMSA (Ext. Data Fig. 10) are due to dissociation of DNA, they rather reflect more transient interactions with the DNA by the mutated D2 proteins; i.e. as a consequence of altered k_{on} / k_{off} parameters.

Ideally, the authors should support their observations with some cellular experiments; demonstrating that the D2 mutations have a demonstrable effect on function of the D2-I complex.

A commentary on how the recognition of ss-dsDNA junctions might interplay with the action of the 9-1-1 complex would also be useful/welcomed.

MINOR:

Please clearly label 5'- and 3'-ends of DNA cartoons.

Convert all plots corresponding to the quantification of DNA binding to % bound (rather than % free) as this is the standard representation across the scientific literature.

Please state clearly what error bars represent; presumably 1 (one) standard deviation.

Referee #3:

In this paper, Alcon et al. applied single-molecule imaging and cryo-EM to address the interaction of FANCD2-FANCI (D2-I) with full dsDNA and ss/dsDNA, aiming to understand its broader role in DNA crosslink repair and fork protection. This is a very important question and the combination of methodologies applied are well-suited to address it. By tracking fluorescently labeled D2-I, the authors demonstrated that it could slide on dsDNA in random one-dimensional diffusion (Fig. 1d-g), and further validated this by showing its inability to bypass another D2-I molecule upon their collision (Fig. 2e) and when encountering a Holliday Junction (HJ) (Fig. 2f). They also characterized in these contexts the effect of phosphorylation and ubiquitination. The authors subsequently focused on D2-I sliding upon encountering three obstacles: HJ, ssDNA and ssDNA-coated with RPA. The authors made a claim that D2-I bounces back when encountering the HJ but stalls at the other two.

I have some questions regarding the experiments that support these findings (see below). Nonetheless, comparing the cryo-EM structures of D2-I bound to full dsDNA and to ss/dsDNA showed the duplex region in the ss/dsDNA displaying more defined electron density. The structures highlighted few key interactions with the DNA that stabilizes the duplex region although a weak density of the ssDNA is observed. These new structures are important for understanding the role of D2-I in fork protection.

In addition to addressing my main comments below, it is a must that the authors characterize the sliding behavior on a DNA containing a crosslink block to increase the broader impact of the study to both DNA crosslink repair and fork protection.

Main comments:

1) In the experiment presented in Fig. 2f, it is unclear from the kymograph how the authors reached the conclusion that D2-I didn't pause at the HJ. It is hard to see this just by looking at the figure (the scale of motion is confined due to the blockage of diffusion by the HJ) and in the absence of quantitative analysis. It appears to me by eye that the highlighted green particle displays pausing behavior at the HJ. The authors must provide quantitative analysis that supports their conclusion of no pausing at the HJ. This analysis must include characterization of the spatial resolution of the assay, by comparing the Brownian motion of the DNA near the HJ location and the displacement of D2-I that show pausing or sliding backward (as shown in Ext. Data Fig. 2c); a control experiment that can address this would be to fluorescently label the DNA near the HJ's location. The authors must also include a control experiment showing that the applied stretching force doesn't affect the structure of the HJ and consequently the findings from this control experiment.

2) In the comparison of the behavior of D2-I upon encountering the HJ (Fig. 2f) versus ssDNA (Fig. 3a), quantitative analysis of the lifetime of pausing must be performed on both cases.

3) In the kymograph in Fig. 3a, I noticed by eye the green and orange particles are located within the vicinity of the ssDNA region. I understand that I could be misinterpreting the position of the ssDNA region at the lower regime forces, since when higher force was applied at the end of the time trace, the green and the orange particles showed pausing at the junction with the green one showing longer lived pausing and maybe the orange particle colliding with it. What is the effect of force on the pausing kinetics at ssDNA? Can the authors provide analysis of lifetime of pausing at ssDNA at different stretching forces and report if they observe any D2-I binding/hopping on the ssDNA region?

4) Related to comment (3), in the experiment presented in Fig. 3b, I understand that the authors used RPA to map the position of the ssDNA. The fact that RPA stretches the ssDNA to similar length as dsDNA (Lewis, J.S., 2017 PNAS, 114:10630-10635), the authors might be able to locate the ssDNA region and perform the experiment at much lower stretching forces. I couldn't find the concentration of RPA used in this experiment in the text, but if lowering the fluorescence signal from RPA at the ssDNA is required, the authors could use a mixture of labeled and unlabeled RPA. On another note, quantitative analysis of pausing lifetime must be also provided in this experiment as well.

5) In the experiment presented in Fig. 3b, at the end of the time trace where the pausing of D2-I at the junction was indicated, there is a D2-I particle (pink color) that is in the vicinity of the ssDNA. What is the explanation for this?

6) Based on the cryo-EM structure, the authors created point mutations that minimize the interaction with the ssDNA (Ext. Data Fig. 10). These mutations provide an excellent control experiment to test their role in recognizing ssDNA at the single molecule level.

7) Can the author add Ext. Data figures with more examples of time traces for each experimental condition?

Author Rebuttals to Initial Comments:

FANCD2-FANCI is a sliding clamp that surveys DNA and recognizes single-stranded gaps

Nature 2023-08-14076

Thank you for the opportunity to revise our manuscript. We appreciate the valuable feedback and have now performed additional *in vitro*, in cell and extract-based studies to understand D2-I stalling at ss-dsDNA junctions. Together, these revisions enhance the quality and impact of our work. Please find below a point-by-point response to each of the referees' comments.

Referee #1 (ICL repair biochemistry):

In this study, Alcon et al. investigate how the FANCD2-FANCI (D2-I) protein complex executes two critical functions, the recognition of DNA inter-strand crosslinks and the protection of stalled replication forks. The authors employ single-molecule imaging to investigate DNA binding by D2-I and find that the complex acts as a sliding clamp on double-stranded DNA, but then stalls upon encountering single-stranded–double-stranded (ss-ds) DNA junctions. Such junctions arise at DNA inter-strand crosslinks during replication and are also present at stalled replication forks. Therefore, the authors data suggest a mechanism, wherein D2-I specifically clamps onto ss-dsDNA junctions to identify sites of DNA damage, offering a unified explanation for its various functions. The other proceed by investigating the mechanistic basis of how D2-I binds to ss-ds DNA-junctions using cryoEM and reveal specific interactions between D2-I and the single-stranded portion of the junction. Based on this structural information, the authors generated D2-I mutant variants with amino acid replacements in the ssDNA interface, which were however only analysed superficially.

Overall, this is a great manuscript, and I am highly enthusiastic about it. The writing is clear and thoughtful. The proposed model consolidates apparently different functions of D2-I into a coherent mode-of-action. However, while the model is certainly elegant, it unfortunately remains untested. But once my comments have been addressed, I think this manuscript will be a great candidate for publication in Nature.

Major points

Guided by their structural analysis, the authors generated two mutant variants of FANCD2 in which key basic residues within the ssDNA-binding interface were replaced. Two sets of EMSA experiments are conducted with these mutants. The authors conclude that the mutant variants show a specific reduction in affinity towards ss-ds DNA-junctions. I cannot agree with this conclusion based on the presented gels and the rather high error bars in their quantifications do not provide confidence in these results as well. I suggest that the authors employ these mutant variants to actually test their model, first *in vitro* using their beautiful single-molecule assays and then in physiologically relevant model systems, cells and frog egg extracts. Only this way can the authors test whether the proposed recognition ss-ds DNA-junctions has any role in DNA inter-strand crosslink repair and replication fork protection.

We thank the referee for their thorough evaluation of our manuscript and their enthusiasm for our study. We acknowledge the referee's constructive feedback regarding our model and have performed an array of additional experiments to further test it *in vitro*, in cells and in extracts, as detailed in the response to the points below. Given the non-quantitative nature of the EMSAs, we have removed them from the manuscript.

1. Do the D2-I mutant variants (KKAA and AARR) fail to stall at ss-ds DNA-junctions in single molecule experiments? Is their residence time on dsDNA affected?

To investigate the hypothesis that substitution of basic residues in the “KR helix” of FANCD2 (**Response Fig. 1a**) affects the ability of D2-I to recognise and stall at ss-dsDNA junctions, we performed additional experiments. First, we tested point mutants in the C-trap single-molecule assay. We generated three FANCD2 constructs with charge-reversal mutations (K400E, K404E, R407E, and R411E), resulting in 'D2^{EERR}' (K400E and K404E), 'D2^{KKEE}' (R407E and R411E), and 'D2^{EEEE}' (K400E, K404E, R407E, and R411E). We reasoned that charge-reversal mutants would be more compromised than alanine mutants. The mutants were purified, Cy3-labeled and incubated with Cy5-labeled FANCI to form double-labelled D2-I complexes.

Using our single-molecule assay, we then investigated the ability of these complexes to recognize and stall at ss-dsDNA junctions using λ DNA with a defined ssDNA gap. To visualize the position of the ssDNA gap, we included eGFP-labelled RPA. As previously shown, D2^{WT}-I displays one-dimensional bi-directional random diffusion on the dsDNA regions of the DNA construct, but it specifically stalls at ss-dsDNA junctions. Stalling occurs at both junctions (3' and 5') flanking the ssDNA gap (**Response Fig. 1b-c**). To quantify the stalling efficiency of D2-I at ss-dsDNA junctions, we collected additional kymographs (N = 80) and scored them based on the presence or absence of static traces at ss-dsDNA junctions. D2^{WT}-I exhibited efficient stalling at ss-dsDNA junctions, observed in 78% of recorded kymographs. In contrast, D2^{KKEE}-I and D2^{EERR}-I showed reduced stalling efficiency, which was observed in a four-fold reduction (N = 35) and or a 10-fold reduction (N = 36) compared to wild-type, respectively (**Response Fig. 1c-f**).

We also tested the ability of D2^{EEEE}-I to bind and stall at ss-dsDNA junctions. However, we did not observe efficient loading of D2^{EEEE}-I on DNA, suggesting that mutation of all four basic residues impairs DNA binding.

Response Fig. 1. *a*, Detail of KR helix in the ss-dsDNA–D2-I structure. The conserved basic residues that interact with the ss-dsDNA are coloured in magenta. *b*, Schematic representation of D2-I (green and red labelled) sliding on dsDNA and stalled at the junctions of RPA (blue labelled)-coated ssDNA gaps. *c-e*, Representative kymographs of D2^{WT}-I, D2^{KKEE}-I, and D2^{EERR}-I on λ DNA with a ssDNA gap. Stalled D2-I complexes are indicated

with yellow triangles on the kymographs. Orange triangles indicate the ss-dsDNA junctions on the DNA. Fluorescent labels are as indicated in panel b. The colored bars under the kymograph indicate the channel position (grey=buffer channel, green-yellow-red=D2-I channel). **f**, Quantification of the number of kymographs displaying stalled D2-I traces at ss-dsDNA junctions. N = number of independent kymographs. The results of pairwise comparisons using Fisher's exact test are shown (****, $p < 0.0001$; ns = non-significant).

To analyze the stalling events in more detail and to classify whether molecules are static or mobile in a quantitative manner, we tracked individual particles in kymographs and employed a rolling diffusion analysis approach to calculate a rolling diffusion coefficient (D_{roll}) with a 16-frame window (further explanation of the method provided in response to Referee #3, point 1). We then plotted this alongside the displacement. For reference, we used Cy3-labelled Cas9_{D10A} (nCas9), which is localized at a specific site (i.e. is static) (Newton et al. 2019). These trajectories enable us to establish a threshold diffusion coefficient $D_{th} = 6.4 \times 10^{-3} \mu\text{m}^2/\text{s}$ ($3 \times D_{roll}[\text{nCas9}]$) for a static molecule, which is indicative of the localisation error and bead diffusion in the trap (**Response Fig. 2a**). We then classified instances where D2-I molecules exhibited a $D_{roll} \leq D_{th}$. Interestingly, this analysis revealed two different types of D2-I stalls. About half ($n = 39/76$ or $\sim 51\%$) of D2^{WT}-I molecules stalled at ss-dsDNA junctions are essentially immobile, and remain static with a lifetime of 52 ± 3 s, which is likely limited by photo-bleaching (**Response Fig. 2b**). We classify these as "persistent stalls". The remaining $\sim 49\%$ of stalled complexes remained static for a short time (mean residence lifetime of 7.3 ± 0.1 s), before diffusing away from the ss-dsDNA junction. We classify these as "transient stalls" (**Response Fig. 2b** and also see response to Referee #3, points 1-3).

We also analysed the mutant complexes and confirmed that they do not stall efficiently at ss-dsDNA junctions (**Response Fig. 2c-d**). Although there were very few stalled mutant complexes, we observed both transient and persistent stalls.

a Static trace from nCas9 (Cy3) and its calculated D_{roll}

b $D2^{WT-I}$ stalls at ss-dsDNA junctions (N = 63/80)

c $D2^{KKEE-I}$ has reduced stalling efficiency at ss-dsDNA junctions (N = 7/35)

d $D2^{EERR-I}$ has reduced stalling efficiency at ss-dsDNA junctions (N = 3/36)

Response Fig. 2. a, (Left) Kymograph collected at ~ 15 pN force of a static (Cy3)-Cas9_{D10A} (nCas9) on λ DNA. Dye position was tracked over time (white rectangle). (Right) Rolling diffusion analysis was performed over the track, yielding the short-range diffusion coefficients of these molecules for 16-point time windows (~ 1 s). **b-d**, Rolling diffusion analysis of $D2^{WT-I}$, $D2^{KKEE-I}$, and $D2^{EERR-I}$ on DNA. Tracks on gapped dsDNA (top) show drops in the rolling diffusion coefficient, D_{roll} (below) below the threshold (dashed line) in all $D2^{WT-I}$ tracks (shown in pink) and in a minority of $D2^{KKEE-I}$ and $D2^{EERR-I}$ tracks.

Transient stalls may represent instances where D2-I molecules sense the ss-dsDNA junction but fail to engage stably. The reason that approximately half of the analyzed molecules had persistent stalls and the other half had transient stalls is not clear, but one possibility is related to the orientation of D2-I when approaching the ss-dsDNA junction. D2-I moves bidirectionally on DNA, but the cryoEM structure of D2-I stalled at a ss-dsDNA junction is directional: In our structure, the dsDNA is always bound to the C-terminal arch of FANCI and not to the C-terminal arch of FANCD2 (**Response Fig. 3**). Since D2-I loads onto dsDNA in a random orientation, it could approach the ss-dsDNA junction with FANCD2 leading, or with FANCI leading. It is possible that approaching the junction with FANCD2 results in a persistent stall (as seen in cryoEM) whereas approaching the junction with FANCI results in a transient stall. This hypothesis is consistent with approximately half of the stalls being persistent and half being transient. This also brings about the interesting idea that D2-I loading onto ss-dsDNA junctions in cells could be controlled to promote persistent stalls.

Orientation 1 (observed): C-terminal arch (C-arch) of FANCI contacts dsDNA and KR helix in FANCD2 contacts ss-dsDNA junction

Orientation 2 (not observed): C-terminal arch (C-arch) of FANCD2 contacts dsDNA and ss-dsDNA junction contacts FANCI

Response Fig. 3. Stalling directionality of D2-I at ss-dsDNA junctions.

In summary, the conserved basic residues in the KR helix of FANCD2 contribute to efficient stalling of D2-I at ss-dsDNA junctions: Mutation of these residues leads to a major reduction in stalling. We have added these data to the manuscript as **Fig. 5c** and **Extended Data Fig. 5-6**.

2. Can the sensitivity to crosslinking agents and the loss of fork protection in FANCD2 KO cells be rescued by expression of the KKAA and AARR mutant variants? Ideally, using ubiquitination-defective FANCD2 and WT as controls.

To assess the role of basic residues within the KR helix in cells, we generated chicken DT40 cell lines stably expressing either alanine (D2^{AARR}, D2^{KKAA}, D2^{AAAA}) or glutamic acid (D2^{EEERR}, D2^{KKEE}, D2^{EEEE}) substitutions in a D2^{-/-} background. We then compared the cisplatin sensitivities of these KR mutant cell lines to cell lines expressing wild-type FANCD2, a mono-ubiquitination deficient FANCD2 (D2^{K563R}), or no FANCD2 (full knock-out D2^{-/-} DT40 cell line). WT DT40 and D2^{-/-} cells complemented with WT FANCD2 showed no sensitivity to cisplatin within the tested dose range (**Response Fig. 4a**). In contrast, cell lines lacking FANCD2 (D2^{-/-}) or complemented

with D2^{K563R} were sensitive to cisplatin, with the point mutant showing greater sensitivity than the full knockout (**Response Fig. 4a**). Thus, a complete lack of FANCD2 is less toxic than carrying a non-functional protein that is capable of binding DNA, as previously reported (Seki et al. 2007).

Interestingly, the arginine mutants, D2^{KKAA} and D2^{KKEE}, fully rescue cisplatin sensitivity of the D2^{-/-} cell line (**Response Fig. 4b**) but the lysine mutants, D2^{AARR} and D2^{EERR}, do not (**Response Fig. 4c**). Consistent with this, the lysine mutants had a stronger effect on stalling (10-fold reduction) than the arginine mutants (4-fold reduction) in single molecule assays (**Response Fig. 1f**). Cell lines containing quadruple point mutations (D2^{AAAA} and D2^{EEEE}) are also unable to rescue (**Response Fig. 4d**). The quadruple mutants are more sensitive to cisplatin than the double lysine mutants. Therefore, these data show that all four analysed residues within the KR helix of FANCD2 play an important role in DNA repair in cells, although the importance of the arginines was only revealed in the lysine mutant background.

Interestingly, cell lines expressing alanine substitutions were more sensitive to cisplatin than glutamic acid substitutions. For example, cell lines expressing D2^{EEEE} had a similar phenotype to D2^{-/-}, whereas D2^{AAAA} was more similar to D2^{K563R} (**Response Fig. 4d**). We hypothesize that the charge-reversal glutamic acid mutants cannot bind DNA efficiently in cells (similar to the inability to bind DNA efficiently in single molecule experiments) and act as nulls. In contrast, the alanine mutants bind DNA but do not function in DNA repair and act as dominant negatives, similar to D2^{K563R}.

Response Fig. 4. The KR helix is important for the response to cisplatin in DT40 cells. Clonogenic survival assays were conducted to assess the sensitivity of DT40 FANCD2^{-/-} cells complemented with chicken FANCD2 and the K563R mutant, residues K400, K404, R407 and R411 mutated to either Ala (panel a) or Glu (panel b). Cisplatin treatment was performed at the indicated doses for 1 hour. Results are based on 2 independent clones of each mutant and 8-12 technical replicates, with mean (point) and standard deviation (error bar) plotted. P-values were calculated using a two-way analysis for cell survival data fitting the LQ model with maximum likelihood (CFAssay R package), with significance codes as follows: * $P < 0.05$; ** $P < 0.01$, *** $P < 0.001$, **** $P < 0.000001$, ns: not significant. P-values enclosed in brackets are for samples compared to WT DT40.

In conclusion, the colony survival assay shows that basic residues in the KR helix are important for an efficient cellular response to cisplatin treatment. This is consistent with a role for the KR helix in recognizing the ss-dsDNA junction in DNA repair and replication stress. These data are now included in our revised manuscript as **Fig. 6a** and **Extended Data Fig. 11a**.

3. Are these mutant variants supporting inter-strand crosslink repair in frog egg extracts upon depletion of endogenous FANCD2? If repair fails, at what step of the FA pathway?

To further test our model that ss-dsDNA junctions are recognized by the KR helix in FANCD2 during interstrand crosslink (ICL) repair, we utilized the *Xenopus* egg extract system that supports FA pathway-mediated ICL repair of a plasmid containing a site-specific cisplatin interstrand crosslink (Raschle et al. 2008). To this end, we purified recombinant *Xenopus laevis* (xl) D2-I complexes containing charge-reversal mutations in the KR helix resulting in 'D2^{EERR}' (K399E and R403E), 'D2^{KREE}' (R406E and R410E), and 'D2^{EEEE}' (K399E, R403E, R406E, and R410E) and tested their abilities to rescue ICL repair upon FANCD2 depletion. (Note that the basic KR helix residues tested in the chicken system (KKRR) are KRRR in xl/FANCD2). We added D2^{WT}-I, D2^{KREE}-I, D2^{EERR}-I, or D2^{EEEE}-I complex

to FANCD2-depleted *Xenopus* egg extract (**Response Fig. 5a, top panels, depletion blots**) and assessed ICL repair over time by monitoring the regeneration of a restriction enzyme site on the plasmid, which is blocked until ICL repair occurs (Raschle et al. 2008). ICL repair was severely inhibited in the presence of all mutant D2-I complexes (**Response Fig. 5a, bottom panels, repair graphs**). D2^{EEEE}-I shows the most drastic defect, while the double mutants (D2^{KREE}-I, D2^{EERR}-I) retain a low level of ICL repair, just above the Δ D2 control. The D2^{KREE}-I mutant seems to support a slightly higher level of ICL repair compared to the D2^{EERR}-I mutant, which is consistent with the higher level of stalling observed for this mutant in the single molecule assay.

Since we and others have previously shown that FANCD2 ubiquitination is required for stable retention of the D2-I complex onto DNA, and for ICL repair (Budzowska et al. 2015; Klein Douwel et al. 2014), we next assessed FANCD2 ubiquitination of the mutant proteins during ICL repair in extract. While recombinant D2^{WT}-I was efficiently ubiquitinated in extracts, no FANCD2 ubiquitination was observed for D2^{EEEE}-I (**Response Fig. 5b, ubiquitination during repair blots**), consistent with the drastic defect in ICL repair (**Response Fig. 5a**). Interestingly, D2^{KREE}-I, and to a lesser extent D2^{EERR}-I, show residual FANCD2 ubiquitination, consistent with their residual ICL repair activities. (**Response Fig. 5b, ubiquitination during repair blots**).

To confirm that the double mutant complexes can be recruited to the DNA during ICL repair in extracts, we performed plasmid pull-down assays and blotted for FANCD2. This shows that both double mutants still bind to the plasmid during repair, albeit with lower efficiency compared to the wildtype FANCD2 protein (**Response Fig. 5c, plasmid pull down blots**), consistent with the lower levels of FANCD2 ubiquitylation.

Response Fig. 5. a, ICL repair efficiency in *Xenopus* egg extracts is compromised with KR helix mutants. Mock-depleted (Mock) and FANCD2-depleted (Δ D2) nucleoplasmic egg extract (NPE) complemented with wild-type FANCD2 (Δ D2 + D2^{WT}) or FANCD2 mutants (Δ D2 + D2^{EEEE}, Δ D2 + D2^{EERR} and Δ D2 + D2^{KREE}), were analyzed by western blot using α -FANCD2 antibody to show FANCD2 levels. Lines within blots indicate positions where

irrelevant lanes were removed. These extracts were used to replicate pICL. Absolute ICL repair efficiency was calculated using regeneration of Sap1 restriction enzyme site and plotted. Dotted line indicates Sap1 fragments from contaminating uncrosslinked plasmid present in pICL preparations. Separate graphs represent independent experiments. **b**, FANCD2 monoubiquitination is compromised in KR helix mutants. pICL was replicated using NPE that was depleted of FANCD2 and complemented with recombinant proteins as described in panel a. Total extract samples were collected at indicated timepoints and analyzed by western blot using α -FANCD2 to visualize ubiquitination efficiency. Lines within blots indicate positions where irrelevant lanes were removed. **c**, FANCD2 KR mutants have reduced DNA retention. Mock-depleted (Mock) and FANCD2-depleted (Δ D2) NPE complemented with wild-type FANCD2 (Δ D2+D2^{WT}) or FANCD2 mutants (Δ D2+D2^{EERR} and Δ D2+D2^{KREE}) were used to replicate pICL. At indicated time points plasmids were isolated by LacI pull-down. Proteins bound to the plasmids were analyzed by Western blot analysis with the indicated antibodies along with a 1% input sample (top blot, first four lanes). Total extract samples were collected at indicated timepoints and analyzed by western blot using α -FANCD2 (bottom blot).

We conclude that the double D2^{KREE}-I and D2^{EERR}-I mutants show severely reduced ICL repair, with residual FANCD2 ubiquitination and damage site recruitment, while D2^{EERE}-I has a drastic defect in ICL repair and deficient FANCD2 ubiquitination, consistent with poor recruitment to DNA. Thus, the KR helix is essential for efficient ICL repair in *Xenopus* extracts. These data are now included in our manuscript in **Fig. 6b-c and Extended Data Fig. 11b-c**.

4. The authors speculate that "By simply physically holding the duplex together at ss-dsDNA junctions, it could prevent the double-stranded region from fraying and from further resection." This is an interesting thought that should be directly tested. Does D2-I protect DNA junctions against resection by an exonuclease in in vitro assays? If yes, is this ability lost in KCAA and AARR mutant variants?

This is an excellent suggestion. A challenge with this experiment is to identify a system that tests exonuclease activity on the double-stranded DNA, starting at a ss-dsDNA junction (and not at the ssDNA end or the blunt dsDNA end). Given the interplay between FANCD2 and DNA2 (e.g. (Liu et al. 2023)), we purified DNA2 to test whether FANCD2 prevents resection by this exonuclease *in vitro*. Unfortunately, our purified DNA2 was not active. We next tried several (non-physiologically relevant) nucleases, and we tried blocking the single-stranded DNA end. However, the results were inconclusive as the nucleases were active on most DNA ends, making it difficult to assay for a specific effect on the D2-I bound junction.

We hope to continue this work by reconstituting the relevant vertebrate nuclease complexes, but this is beyond the scope of the current work. Nevertheless, there is substantial existing evidence that FANCD2 protects against resection both *in vitro*, in cells and in mice [e.g. (Garaycochea et al. 2018; Liu et al. 2023)] and this is consistent with our hypothesis. We have changed the text to: "By simply physically holding the duplex together at ss-dsDNA junctions, it could prevent the double-stranded region from fraying and from further resection (...) Further testing of this hypothesis will be the subject of future work."

5. The authors show that FANCI also binds DNA by itself. The authors' model predicts that FANCI would not stall at junctions in the absence of FANCD2. Is this the case? Vice versa, FANCD2 is shown to not bind dsDNA on its own. However, does it bind to junctions?

Using our single-molecule assay, we have shown that FANCI alone binds to dsDNA, albeit with lower loading efficiency, slower diffusion and shorter dwell times (Extended Data Fig. 2d-f). As per the referee's suggestion, we have now investigated the binding of FANCI alone to λ DNA containing a defined ssDNA gap. We did not observe FANCI stalling at ss-dsDNA junctions (**Response Fig. 6a**). Interestingly, we observed a small number of cases of FANCI within the ssDNA gaps, which diffuse slowly but do not localise to ss-dsDNA junctions. Consistent with this, it has been previously shown that FANCI binds to ssDNA *in vitro* (Longerich et al. 2009; Yuan et al. 2009). It is unclear if this is physiologically relevant, i.e. whether any monomeric FANCI would be available for binding DNA in the nucleus.

Using the same setup, we also find that FANCD2 alone does not efficiently bind to dsDNA, and does not bind or stall at ss-dsDNA junctions (**Response Fig. 6b**).

In conclusion, FANCI binds and diffuses on dsDNA (albeit inefficiently), and it can also bind to ssDNA *in vitro*. However, neither FANCI nor FANCD2 on their own specifically stall at ss-dsDNA junctions, in contrast to the D2-I heterodimer, which slides proficiently on dsDNA and recognises ss-dsDNA junctions (**Response Fig. 1c**). These data are now included in **Extended Data Fig. 7**.

Response Fig. 6. a, Representative kymographs showing FANCI on λ DNA containing a defined ssDNA gap. Red triangles indicate instances where FANCI is observed within the ssDNA gap. **b**, Representative kymographs showing FANCD2 on λ DNA containing a defined ssDNA gap. A small number of FANCD2 traces were observed on DNA (green triangle). The colored bars indicate the channel position (grey=buffer channel, red=FANCI channel, green=FANCD2 channel) **c**, Schematic representation of FANCD2 and FANCI monomers and their interaction with DNA. FANCD2 is labelled in green, FANCI is in magenta and RPA is in blue.

Minor points

6. Ext. Data Fig. 1d: Immuno-blots for FANCI and FANCD2 should be used to determine whether there is any difference in ubiquitination. The Coomassie-stained gel may suggest that there is one, but it is difficult to judge.

We have now conducted time-course reactions to directly compare the ubiquitination rates of untagged, non-labelled D2-I complexes with those of ybbR-tagged and fluorescently labelled D2(Cy3)-I(Cy5) complexes. The ubiquitination rates are comparable, and therefore, we conclude that ybbR-tagging and fluorescent labelling do not substantially affect the ubiquitination efficiency of D2-I (**Response Fig. 7**). This experiment is now included in **Extended Data Fig. 1d**.

Response Fig. 7. Time-course ubiquitination assay of untagged D2-I and ybbR-tagged, fluorescently labelled D2-I. Ubiquitination reactions were set up as described in Methods, samples were taken at the indicated time points and loaded on a 3-8% NuPAGE Tris-Acetate gel (Invitrogen). Top panel shows Coomassie-stained gel and bottom panel shows merged scan in the green (Cy3) and red (Cy5) channels. This is a representative gel of an experiment repeated three times.

7. Does D2-I show any preference for 5' or 3' ends at the junction?

Our single-molecule data (C-trap) shows that D2-I binds and stalls at both 3'- and 5'-recessed ends flanking a ssDNA gap. To generate this ssDNA gap on lambda DNA, we used a nickase version of dCas9_{D10A} (nCas9). After cleavage by nCas9, we remove the nCas9 by Proteinase K digestion and release the intervening ssDNA by force stretching. The resultant gap has a 3'-OH and a 5'-phosphate. Because the gap position is asymmetric within the λ DNA, we can differentiate the 3'- and 5'-recessed ends of the ss-dsDNA junctions (**Response Fig. 10a**).

To determine whether D2-I preferentially stalls at the 5' or 3' end of a ssDNA gap, we collected additional kymographs to perform statistical analysis of its stalling preference. In kymographs where static traces of D2-I are observed (78% of all kymographs recorded for D2^{WT}-I), ~22% displayed stalling on both ends, ~41% on the 5' end, and ~37% on the 3' end. Therefore, our single-molecule analysis does not show a significant preference for either 5' or 3' junctions (**Response Fig. 8**).

Response Fig. 8. a, Schematic representation of the ssDNA gap generation. b, Representative kymographs of D2-I on gapped λ DNA displaying stalling at either both, 5'- or 3'-ended ss-dsDNA junctions. b, Percentage of kymographs of D2-I on gapped λ DNA displaying stalling at either the 5'-end of ss-dsDNA gaps, the 3'-end, or both. c, Representative kymographs of D2-I on gapped λ DNA displaying stalling at the 5'-end of ss-dsDNA junctions (indicated by light purple arrow), the 3'-end (indicated by dark purple arrow), or both ends. The colored bars indicate the channel position (grey=buffer channel, red, yellow and green=D2-I channel)

In addition, we compared the binding affinities and binding kinetics of D2-I towards DNA constructs containing a 3'-OH, 5'-OH, or 5'-phosphorylated ss-dsDNA junction using SwitchSENSE. SwitchSENSE employs dsDNA nanolevers labelled with fluorescent markers, attached to a gold surface, which oscillate in response to alternating electric fields. Upon introducing protein (D2-I) onto the chip, alterations in the speed of DNA nanolever switching enable the analysis of DNA-protein interaction kinetics. Our analysis shows that D2-I has comparable association and dissociation constants regardless of the polarity of the DNA end, resulting in comparable binding affinities. Therefore, and in agreement with our single-molecule analysis, SwitchSENSE shows that D2-I has no preference for either 3'- or 5'-ended ss-dsDNA junctions. We note that SwitchSENSE gives only a two-fold lower binding affinity for blunt ended DNA than for DNA with a ssDNA gap. Thus, it is likely that it is reading out the affinity to dsDNA and not necessarily to the ss-dsDNA junction. We include both the single molecule and SwitchSENSE data in our manuscript as Fig. 3c and Extended Data Fig 4c-d.

Response Fig. 9. *a*, Schematic representation of the SwitchSENSE experiment. Data points from association (upon addition of protein) and dissociation (upon addition of buffer) are used to fit association and dissociation curves for D2-I. *b*, Schematic representation of the different DNA substrates used in SwitchSENSE experiments. *c*, Table with calculated k_{on} , k_{off} and K_d values (mean and standard deviations) from two independent runs. *d*, Mean K_d values for D2-I on different DNA substrates. Error bars represent standard deviations. Pairwise comparisons between the calculated affinities to each different DNA substrates were not significant (ns) in all cases.

8. Page 4, "Photobleaching was observed as a single-step event, consistent with the analyzed trajectories comprising single D2-I heterodimers (Extended Data Fig. 2b). Therefore, D2-I moves along the DNA duplex via thermal-energy driven random one-dimensional diffusion.": I suggest rephrasing, as the concluding sentence seems to reference information discussed above rather than the preceding sentence.

We have adjusted this sentence to improve clarity: "Photobleaching was observed as a single-step event, consistent with the analyzed trajectories comprising single D2-I heterodimers (Extended Data Fig. 1f). Analysis of these data show that D2-I moves along the DNA duplex via thermal-energy driven random one-dimensional diffusion (Extended Data Fig. 1g)."

9. Fig. 3b: One FANCI molecule binds to the ssDNA region. How often are such events observed?

Out of 80 kymographs of D2-I on gapped λ DNA, we observe 6 kymographs (7%) where a single trace of FANCI appears in the ssDNA gap, while 2 kymographs (2%) show 2-3 FANCI traces within the same gap. Notably, the vast majority (91%) show no FANCI presence within the gap. The observed FANCI molecules show slow diffusion and short lifespan, typically lasting less than 5 seconds. Importantly, we did not observe FANCI at ssDNA junctions.

Furthermore, we directly investigated the binding of FANCI alone to gapped λ DNA, as addressed in our response to point 5. We show that FANCI binds to dsDNA and ssDNA gaps, both in the absence and presence of RPA (see Response Fig. 6a), consistent with earlier reports (Longerich et al. 2009; Yuan et al. 2009). Notably, FANCI binding to DNA is significantly less efficient: using the same concentration (5 nM FANCI), we observed approximately ten times fewer instances of FANCI bound to DNA compared to D2-I. Moreover, this binding

required longer incubation time in the protein channel. Thus, while FANCI can indeed bind to DNA, it does so less efficiently than D2-I and it leads to short, slow traces on both ss and dsDNA but not at ss-dsDNA junctions.

10. Page 5, “Due to random photobleaching events, a given D2-I particle may have green fluorescence, red fluorescence or both. This allowed us to differentiate complexes from each other.”: Could red fluorescent particles also represent FANCI homodimers?

Although we have shown that FANCD2 can form homodimers (Alcon et al. 2020), there is no evidence of FANCI homodimerization. Thus, we do not believe that the red-only traces are homodimeric FANCI. They are most likely FANCI monomers or D2-I heterodimers where FANCD2 is either non-labelled or the Cy3 fluorophore has been photobleached.

11. The number of replicates performed per experiment is sometimes mentioned in the respective figure legend but is not always provided.

We have corrected this in the revised manuscript.

Referee #2 (cryoEM of DNA repair):

The authors present a series of experiments showing that the FANCD1-FANCI (D2-I) complex is capable of detecting and binding to discontinuities in the DNA backbone, specifically double-strand to single-strand transitions. They present a comprehensive set of single-molecule experiments using a Lumicks C-TRAP, supporting this hypothesis. Generally, the manuscript is well written, with a logical structure and clear narrative. There are no major concerns with the authors' findings, but some additional detail and clarification is required (see points below), particularly with respect to the mode of DNA-interaction.

We appreciate the referee's comments and now provide additional detail and clarifications, as described below.

MAJOR:

1. From the main text of the manuscript and accompanying figures/illustrations, it is not clear as the nature of the ds-ssDNA junction; i.e. does it represent a 3'- or 5'-recessed end? Does FANCD1-FANCI require a 3'-OH, or 5'-P for binding (depending on the overhang/recess)? This aspect of the D2-I / DNA interaction should be formally demonstrated.

We thank the referee for pointing this out and we have clarified the nature of the junctions for each experiment. We have also performed additional single molecule analyses to test whether there is a preference for the 5' or 3' junction. In addition, we have used SwitchSENSE to analyse the binding kinetics of D2-I to DNA substrates with different ss-dsDNA junctions. These experiments are described in detail in our response to Referee #1, point 7 (**Response Fig. 8**). In summary, our single-molecule assays and analysis of binding kinetics on different DNA substrates do not show a strong preference for stalling on either end. We hypothesize that D2-I is found at the ss-dsDNA junction because: 1) it recognizes the dsDNA region with the C-terminal arch of FANCI; 2) it recognizes the ss-dsDNA junction with the KR helix; and 3) because there is no energetic penalty for bending the dsDNA in this configuration, it stalls at the junction rather than continuing to slide. This provides D2-I with the flexibility to recognize many different types of ss-dsDNA junctions. These data are now included in the manuscript (Figure 3c and Extended Data Fig. 4c-d).

2. Electrophoretic Mobility Shift Assays (EMSA); Ext. Data Figs. 6 and 10:

A significant amount of the D2-I complex, at higher concentrations, does not migrate into the gel, signifying either that: a higher-order complex is formed; there are significant "end-binding effects"; or the complex has precipitated out of solution. As this can affect the analysis and interpretation of nucleic-acid interactions they should be mitigated, either by a change in protocol or use of an alternate methodology (for example, fluorescence polarisation, SPR or similar).

The effects of the D2KKAA-I and D2AARR-I mutation sets on binding to ss-dsDNA require revisiting, as the data, whilst indicative, is not conclusive — especially given the very wide error bars at higher protein concentrations (see comments above). Ideally, values for K_d should be measured and reported. Contrary to the authors' suggestion that the 'smears' observed in the EMSA (Ext. Data Fig. 10) are due to dissociation of DNA, they rather reflect more transient interactions with the DNA by the mutated D2 proteins; i.e. as a consequence of altered k_{on} / k_{off} parameters.

We agree that EMSAs are not ideal for quantitative analyses of DNA-protein interactions. We have therefore removed the EMSAs from the manuscript and now include SwitchSENSE experiments. These allowed us to determine the DNA binding affinities, as well as association and dissociation rates for D2^{WT}-I, D2^{EERR}-I, D2^{KKEE}-I and D2^{EEEE}-I. SwitchSENSE employs dsDNA nanolevers labelled with fluorescent markers, attached to a gold surface, which oscillate in response to alternating electric fields. Upon introducing protein (D2-I) onto the chip, alterations in the speed of DNA nanolever switching enable the analysis of DNA-protein interaction kinetics.

Using a DNA containing a 3'-ended ss-dsDNA junction, we obtained the SwitchSENSE-derived binding affinities for wild-type and mutant D2-I complexes. D2^{WT}-I displays the tightest binding (~ 0.4 nM). The double mutants have slightly lower affinities (~ 0.67 nM for D2^{KKEE} and ~ 2.1 nM for D2^{EERR}-I). Interestingly, D2^{EEEE}-I shows the lowest binding affinity (~ 3.6 nM), about 10-fold lower than D2^{WT}-I. These results agree with our single-molecule experiments where WT and double mutants readily bind and slide on dsDNA, whereas D2^{EEEE}-I does not efficiently load onto DNA. These experiments also agree with function in DT40 cells where cisplatin sensitivity

followed the order WT=KKEE>EERR>EEEE. The same trend was observed with the mutants in *Xenopus* egg extracts. In summary, the function of D2-I correlates with its ability to bind to a DNA containing a ss-dsDNA junction (**Response Fig. 10**).

In addition, we used SwitchSENSE to compare the binding affinities of KR helix mutants towards different DNA substrates. The trend for DNA binding was similar to that discussed above with the 3' end ss-dsDNA junction. Overall, each mutant binds each DNA substrate (5'-end ss-dsDNA junction, 3'-end ss-dsDNA junction or blunt end dsDNA) with a similar affinity, although they have the weakest affinities for blunt-end dsDNA. Moreover, in each case, wild-type D2-I binds the DNA substrate with a slightly higher (but reproducible) affinity than D2^{KKEE}-I, which in turn binds each DNA substrate with a slightly higher affinity than D2^{EERR}-I (**Response Fig. 10**). We note that all of our SwitchSENSE substrates have dsDNA regions and it will therefore also measure the affinity to dsDNA. Importantly, it does not read out the dynamic behavior that we observe in single molecule assays.

Response Fig. 10. a, Schematic representation the different DNA substrates used in SwitchSENSE assays (top) and the calculated K_d values for each D2-I complex (bottom). **b**, Table containing the calculated k_{on} , k_{off} and K_d values and their corresponding standard deviations, for each D2-I complex on the different DNA substrates.

3. Ideally, the authors should support their observations with some cellular experiments; demonstrating that the D2 mutations have a demonstrable effect on function of the D2-I complex.

We agree and have therefore performed experiments in DT40 cells and *Xenopus* egg extracts to show that the KR mutants are defective in ICL repair. Please see our response to **Referee #1, points 2 and 3**. These data are now included in the manuscript as Fig. 6a-c and Extended Data Fig. 11.

4. A commentary on how the recognition of ss-dsDNA junctions might interplay with the action of the 9-1-1 complex would also be useful/welcomed.

Comparison of the activities and recognition modes of ss-dsDNA junctions by D2-I and the Rad9-Hus1-Rad1 (9-1-1) complex reveals intriguing parallels and distinctions between these sliding DNA clamps in the DNA damage response.

Structurally, the 9-1-1 complex, like PCNA, forms a heterotrimeric ring-shaped DNA clamp, while D2-I, a heterodimer, initially adopts an extended (open) structure, transitioning to a closed clamp upon DNA binding. However, their loading mechanisms vary significantly: 9-1-1 loads onto 5'-ended ss-dsDNA junctions via the Rad17-Replication like complex (Rad17-RLC) in an ATP-dependent manner, akin to PCNA loading onto 3'-ended ss-dsDNA junctions by Replication Factor C (RFC). The 9-1-1 clamp is loaded onto ssDNA, and structural analysis of the yeast 9-1-1 showed that the preference for 5'-ended ss-dsDNA junctions is imparted by the Rad24 subunit (Rad17 in humans) in the clamp loader. In contrast, unlike PCNA and 9-1-1, D2-I loads onto dsDNA without the requirement of a clamp loader (Castaneda et al. 2022; Schrecker et al. 2022; Zheng et al. 2022).

Both complexes play roles in sensing DNA damage and stalled replication forks, triggered by the accumulation of ssDNA. Mechanistically, these DNA clamps could recruit additional repair factors, protect the damaged site, coordinate resection, and/or prepare the DNA substrates for repair enzymes. Future investigation of the interplay between D2-I and other junction-signalling sliding clamps like 9-1-1 and PCNA promises valuable insights into DNA repair coordination and regulation.

We have added the following to the main text “Other DNA clamps also function as platforms to recruit repair enzymes to sites of DNA damage (e.g. 9-1-1 complex and PCNA). Interestingly, these clamps are dependent on ATP-driven loaders to assemble on DNA (Castaneda et al. 2022; Schrecker et al. 2022), whereas D2-I loads and clamps dsDNA on its own, but is regulated by post-translational modifications..”

MINOR:

Please clearly label 5'- and 3'-ends of DNA cartoons.

We have added labels on the figures to clarify the nature of the junctions for each experiment.

Convert all plots corresponding to the quantification of DNA binding to % bound (rather than % free) as this is the standard representation across the scientific literature.

In the revised version, we have removed the EMSAs and replaced them with C-trap (Fig. 5c) and SwitchSENSE data (Extended Data Fig. 4c).

Please state clearly what error bars represent; presumably 1 (one) standard deviation.

We have corrected this in the revised version.

Referee #3 (single molecule analysis of DNA repair/replication):

In this paper, Alcon et al. applied single-molecule imaging and cryo-EM to address the interaction of FANCD2-FANCI (D2-I) with full dsDNA and ss/dsDNA, aiming to understand its broader role in DNA crosslink repair and fork protection. This is a very important question and the combination of methodologies applied are well-suited to address it. By tracking fluorescently labeled D2-I, the authors demonstrated that it could slide on dsDNA in random one-dimensional diffusion (Fig. 1d-g), and further validated this by showing its inability to bypass another D2-I molecule upon their collision (Fig. 2e) and when encountering a Holliday Junction (HJ) (Fig. 2f). They also characterized in these contexts the effect of phosphorylation and ubiquitination. The authors subsequently focused on D2-I sliding upon encountering three obstacles: HJ, ssDNA and ssDNA-coated with RPA. The authors made a claim that D2-I bounces back when encountering the HJ but stalls at the other two.

I have some questions regarding the experiments that support these findings (see below). Nonetheless, comparing the cryo-EM structures of D2-I bound to full dsDNA and to ss/dsDNA showed the duplex region in the ss/dsDNA displaying more defined electron density. The structures highlighted few key interactions with the DNA that stabilizes the duplex region although a weak density of the ssDNA is observed. These new structures are important for understanding the role of D2-I in fork protection.

In addition to addressing my main comments below, it is a must that the authors characterize the sliding behavior on a DNA containing a crosslink block to increase the broader impact of the study to both DNA crosslink repair and fork protection.

We thank the referee for their comments and suggestions and address their concerns in detail below.

With respect to crosslinked DNA, it is often assumed that D2-I recognizes the DNA crosslink since it localises to stalled replication forks and is required for ICL repair. However, direct evidence for this is lacking. Moreover, ICLs are heterogeneous, and it is not clear how D2-I (a DNA clamp that encircles dsDNA) would recognize them. The Pavletich lab previously characterized structures of D2-I in the presence of crosslinked DNA, but they were unable to directly visualize the ICL (Wang 2020). It, therefore, remained unclear whether D2-I directly recognises crosslinks.

To characterize the sliding behaviour of D2-I on dsDNA containing a crosslink using single-molecule assays, we attempted to create a DNA construct containing a single inter-strand crosslink at the centre of a dsDNA molecule large enough to be captured in our C-trap. Unfortunately, despite repeated attempts, we were unable to efficiently ligate the crosslinked substrate to biotinylated DNA handles. We used the same protocol as for generating the ICL plasmid used in *Xenopus* egg extracts (Enoiu et al. 2012). This is already an inefficient ligation reaction with a circular template, which is compounded by the additional challenges in ligating it into the two separate linear DNA constructs required for single molecule studies.

As an alternative, we created multiple crosslinks on λ DNA by incubating it with cisplatin (following (Cai et al. 2023)). We observed diffusion of D2-I on these DNA molecules, albeit with a lower diffusion coefficient on average (~ 10 -fold lower; mean $D = 0.01 \pm 0.02 \mu\text{m}^2 \text{s}^{-1}$, $N = 14$) when compared to untreated dsDNA. Interestingly, we did not observe static D2-I molecules on this substrate, suggesting that D2-I does not bind directly to crosslinks. We confirmed the presence of cisplatin-induced crosslinks on the DNA by measuring the force-distance curve of the cisplatin-treated DNA, but a major challenge is that we are unable to ascertain the number and location of these crosslinks. We are also unable to determine the identity of these crosslinks, and whether they were inter-strand or intra-strand (or indeed if they were some other cisplatin (Crisafuli et al. 2012; Hou et al. 2009) adduct). In light of these drawbacks, we are unable to conclude if the observed slower diffusion was due to interactions between D2-I and the crosslinks, or due to frustrated diffusion in the presence of roadblocks caused by the cisplatin adducts. Nonetheless, when we investigated the behaviour of a mutant variant of D2-I (D2^{EERR-I}) on the same DNA construct, we also observed a similar lower-diffusion coefficient. Unlike D2^{WT-I}, the mutant D2^{EERR-I} shows negligible ICL repair *in vivo* (Response Fig. 5).

In summary, although our single-molecule data with crosslinked DNA are not conclusive, they are consistent with D2-I recognizing ss-dsDNA junctions – the consequence of ICLs – and not necessarily ICLs themselves.

Main comments:

1) In the experiment presented in Fig. 2f, it is unclear from the kymograph how the authors reached the conclusion that D2-I didn't pause at the HJ. It is hard to see this just by looking at the figure (the scale of motion is confined due to the blockage of diffusion by the HJ) and in the absence of quantitative analysis. It appears to me by eye that the highlighted green particle displays pausing behavior at the HJ. The authors must provide quantitative analysis that supports their conclusion of no pausing at the HJ. This analysis must include characterization of the spatial resolution of the assay, by comparing the Brownian motion of the DNA near the HJ location and the displacement of D2-I that show pausing or sliding backward (as shown in Ext. Data Fig. 2c); a control experiment that can address this would be to fluorescently label the DNA near the HJ's location. The authors must also include a control experiment showing that the applied stretching force doesn't affect the structure of the HJ and consequently the findings from this control experiment.

We thank the referee for pointing out this. Our data show that D2-I neither bypasses Holliday junction-like structures (formed by a double 15-nt loop and referred to as 'HJ') nor stalls upon reaching these sites. Instead, the HJ acts as a roadblock that causes sliding D2-I to bounce off and continue sliding. To further investigate a possible transient retention or pausing behaviour at HJs, we collected additional kymographs of D2-I on DNAs containing HJ (N=10). Additionally, as the reviewer suggested, we further characterised the spatial resolution of our assays.

Utilizing our single-particle tracking algorithm, we detected the trajectories of three fiduciary static molecules on DNAs tethered to the optical traps: a Cy3-labelled nCas9 on λ DNA, a single Atto-647N dye on a completely double-stranded 17.8kb DNA, and an Atto-647N dye on a 17.8kb DNA containing a ssDNA gap (**Response Fig. 11, left panels**). We plotted the histograms of displacements over consecutive time points for these trajectories and calculated Gaussian fits. As expected, the means of these distributions were nearly zero, showing that there was no systematic bias in our tracking assays. The widths, *i.e.*, the standard deviations of the distributions, were ~ 20 nm, reflecting the precision in our detection of consecutive static positions. Thus, we could estimate that the spatial resolution of our assays is ~ 20 nm on average (**Response Fig. 11, central panels**).

The analysis described above showed that, over short timescales, bead diffusion in the trap (limited by the trap stiffness) can lead to apparent displacements within the entire bead-DNA tether. Thus, to distinguish D2-I stalling from small displacements caused by frustrated diffusion of D2-I around obstacles, we employed a rolling diffusion analysis approach. Briefly, we computed mean squared displacements (MSDs) over short windows (16-time points) rather than over the entire trajectory, yielding the rolling diffusion coefficients (D_{roll}) for these specific windows. Employing rolling windows over the entire trajectory, we could estimate how the diffusion of D2-I changed over the course of the trajectory. For our fiduciary static molecules, we observed that D_{roll} is consistently low ($< 6.4 \times 10^{-3} \mu\text{m}^2/\text{s}$). We could, therefore, use these trajectories to set a threshold ($D_{th} = 6.4 \times 10^{-3} \mu\text{m}^2/\text{s}$) below which we could consider D2-I to be static in our assays (**Response Fig. 11, right panel**).

Response Fig. 11 Tracking of fiduciary static molecules. (Left panels) Kymographs collected at ~ 15 pN force of (a) static (Cy3)-nCas9 on λ DNA, (b) Atto-647N on double-stranded 17.8 kb DNA, and (c) Atto-647N on 17.8 kb DNA containing a ssDNA gap. Dye positions were tracked over time (white rectangle). (Middle panels) Histograms of displacements over single time points for the tracks were fit to Gaussian distributions. (Right panels) Rolling diffusion (D_{roll}) analysis was performed over the tracks, yielding the short-range diffusion coefficients of these molecules for 16-point time windows (~ 1 s). We define a static molecule as any trace having a D_{roll} of $< 6.4 \times 10^{-3} \mu\text{m}^2/\text{s}$ (dotted line).

Upon analysing the interaction of D2-I with ss-dsDNA junctions, we confirmed that we can detect stalling of D2-I at these junctions using this rolling analysis. The rolling diffusion coefficient D_{roll} remained below D_{th} ($6.4 \times 10^{-3} \mu\text{m}^2/\text{s}$) over the entire trajectory, confirming that D2-I is essentially static (**Response Fig. 12a** – but also see response to point 2 below for further analysis). In contrast, we observed no such stalling at HJs (**Response Fig. 12c**). Together, these data support our conclusion that D2-I does not pause at HJs, which instead act as obstacles to D2-I diffusion. These data are included in the revised manuscript (Extended Data Fig. 5).

a Persistent $D2^{WT-I}$ stalling at ss-dsDNA junctions (N = 39/76)

b Transient $D2^{WT-I}$ stalling at ss-dsDNA junctions (N = 37/76)

c No stalling of $D2^{WT-I}$ at Holliday junctions (N = 10)

Response Fig. 12. Rolling diffusion analysis of $D2-I$ on $dsDNA$. $D2-I$ tracks on $dsDNA$ with a $ssDNA$ gap (a–b, top) shows a rolling diffusion coefficient, D_{roll} (below) below the threshold of $6.4 \times 10^{-3} \mu m^2/s$ (dashed line), signifying persistent (a) and transient (b) stalls at the $ss-dsDNA$ junction (shown in pink). No such stalls are seen for $D2-I$ upon encountering Holliday junctions (c).

Regarding the effect of the applied stretching force on the HJ, we have previously characterized the stability of this HJ DNA construct, containing two 15-bp hairpins flanked by 7.5 kbp $dsDNA$. This showed that the HJ is stable at forces below 23 ± 1 pN (Kaczmarczyk et al. 2022). Based on this result, the forces of 5 pN and 15 pN used in our experiments should not affect the stability of the HJ. We could verify this stability in our own assays by performing force-ramp experiments which also showed the HJ structure disrupts at forces around 23 pN.

2) In the comparison of the behavior of $D2-I$ upon encountering the HJ (Fig. 2f) versus $ssDNA$ (Fig. 3a), quantitative analysis of the lifetime of pausing must be performed on both cases.

Our rolling diffusion analysis above showed that $D2-I$ exhibits two types of behaviour upon encountering an $ss-dsDNA$ junction. In approximately half of the cases (N = 39/76), we observed a $D2-I$ molecule persistently stalling at the $ss-dsDNA$ junction, while in the remaining cases (N = 37/76), we observed $D2-I$ molecules transiently stalling at and then diffusing away from the $ss-dsDNA$ junction.

Separately, we performed an independent kinetic analysis of the dwells of D2-I at the ss-dsDNA junctions. We saw that at least two kinetic phases were needed to accurately describe the observed kinetics of D2-I stalling at these junctions. This agreed well with our observation of the two different kinds of D2-I stalling. The longer-lived dwells, which had a lifetime of 52 ± 3 s (which is likely to be photobleaching limited), could be attributed to persistent D2-I stalls, while the comparatively shorter-lived dwell, with a lifetime of 7.3 ± 0.1 s, could be attributed to transient D2-I stalls (**Response Fig. 13**). As our rolling diffusion analysis showed, D2-I did not exhibit either persistent or transient stalling when encountering the HJ (**Response Fig. 11**). These data are included in the revised manuscript as Extended Data Fig. 7.

Response Fig. 13. Quantitative analysis of the lifetime of D2-I stalls. The survival probability plot for the distribution of D2-I dwells from 54 D2-I molecules (purple) along with the corresponding double exponential decay fit (black). The lifetimes of the two phases of the fit are shown.

3) In the kymograph in Fig. 3a, I noticed by eye the green and orange particles are located within the vicinity of the ssDNA region. I understand that I could be misinterpreting the position of the ssDNA region at the lower regime forces, since when higher force was applied at the end of the time trace, the green and the orange particles showed pausing at the junction with the green one showing longer lived pausing and maybe the orange particle colliding with it. What is the effect of force on the pausing kinetics at ssDNA? Can the authors provide analysis of lifetime of pausing at ssDNA at different stretching forces and report if they observe any D2-I binding/hopping on the ssDNA region?

In the figure highlighted (reproduced below as Response Fig. 14), the first third of the kymograph contains a double-stranded λ DNA molecule with nicks at two pre-defined locations (“Nicked DNA” in Response Fig. 14). D2-I appears to diffuse through the nicks in the dsDNA, highlighting that nicks do not serve as roadblocks to D2-I diffusion. Subsequently, we generated a ssDNA gap in the region between the two nicks (following (Belan et al. 2021) by increasing the tension in the DNA in two steps. The two steps can be followed by observing the distance between the two beads. During the first step (the middle region of the kymograph, labelled “Partial ssDNA gap in Response Fig. 14), the dsDNA is likely partially melted between the two nicks, resulting in a ss-dsDNA junction at the top where a “yellow” molecule stalls (labelled “1”). During the second step (the right-hand side of the kymograph, labelled “Fully formed ssDNA gap”), the nicked DNA strand likely fully melts and dissociates, leaving a complete ssDNA gap. Once this gap is fully formed, D2-I stalls at both ss-dsDNA junctions and is unable to diffuse into the gap (molecules “1” and “2”). We have clarified this in the legend in our manuscript.

Response Fig. 14. Generation of ssDNA gap and visualisation of D2-I stalling at ss-dsDNA junctions *in situ*.

We thank the referee for highlighting the different behaviours between D2-I molecules. First, we find that D2-I exhibits two types of stalling at the ss-dsDNA junction, a “persistent stall” (lifetime of 52 ± 3 s – limited by photobleaching) and a “transient stall” (lifetime of 7.3 ± 0.1 s), which is demonstrated by D2-I molecules labelled “1” (persistent) and “2” (transient) respectively in Response Fig. 12. We believe that this difference is consistent with D2-I binding the dsDNA with two different orientations, only of which yields a persistent stall (see Response Fig. 3 and accompanying discussion in response to Referee 1, point 1). The molecule labelled “3” in Response Fig. 14 exhibits a different behavior – it is primarily diffusing but collides with molecule “2”.

The high force used in this experiment is a consequence of our procedure for generating a ssDNA gap *in situ* on a D2-I bound DNA molecule. In all other experiments, we formed the gap before incubation of the DNA with D2-I. Specifically, we normally incubated the gapped DNA with D2-I at very low forces (<1 pN) to enable efficient loading of D2-I on DNA (see Extended Data Fig. 1h of revised manuscript showing force dependence on D2-I loading) before stretching it to ~15 pN for imaging. In a substantial proportion of these kymographs (Supplementary Figure 2), we observe that D2-I is already stalled at the ss-dsDNA junction at the onset of imaging, highlighting that these stalls may be formed at low forces. Diffusing D2-I can also stall at junctions that are not already occupied at the beginning of the experiment, at the imaging force of 15 pN. To investigate the force dependence further, we have additionally imaged D2-I on gapped DNA at a ~5 pN (instead of ~15 pN), where we also observe similar stalling behaviour (Response Fig. 15).

Response Fig. 15. D2-I stalls at ss-dsDNA junctions at lower force (~5 pN).

In summary, we have not seen any major differences in the stalling behavior of D2-I at different forces, and we do not observe D2-I in the ssDNA gap.

4) Related to comment (3), in the experiment presented in Fig. 3b, I understand that the authors used RPA to map the position of the ssDNA. The fact that RPA stretches the ssDNA to similar length as dsDNA (Lewis, J.S., 2017 PNAS, 114:10630-10635), the authors might be able to locate the ssDNA region and perform the experiment at much lower stretching forces. I couldn't find the concentration of RPA used in this experiment in the text, but if lowering the fluorescence signal from RPA at the ssDNA is required, the authors could use a mixture of labeled and unlabeled RPA. On another note, quantitative analysis of pausing lifetime must be also provided in this experiment as well.

We thank the referee for pointing this out. All of the experiments with RPA were performed using 800 pM of RPA-eGFP in the RPA channel. We have added the experimental RPA concentration to the manuscript.

While the majority of our experiments with gapped DNA were imaged at ~15 pN, following the protocol described by (Belan et al. 2021) as described above, we also imaged gapped DNA at ~5 pN and observed similar stalling behaviour (see above **Response Fig. 15**).

To assess whether these traces at ss-dsDNA junctions corresponded to truly static D2-I complexes, we performed rolling diffusion analysis at 5 pN confirming that D2-I stalls at ss-dsDNA junctions even when DNA is held at lower forces (**Response Fig. 16**). Moreover, similar to our observations at 15 pN, we found that 50% of stalls are transient and 50% are persistent at 5 pN (N=10).

Response Fig. 16. Rolling diffusion analysis of a fiduciary static molecule (Atto647N) on 17.8 kb gapped dsDNA held at 5 pN (a) and stalled D2-I at ss-dsDNA junctions on a gapped DNA held at 5 pN (b).

5) In the experiment presented in Fig. 3b, at the end of the time trace where the pausing of D2-I at the junction was indicated, there is a D2-I particle (pink color) that is in the vicinity of the ssDNA. What is the explanation for this?

We thank the referee for drawing our attention to this. Using our single-molecule assay (C-trap), we had shown that FANCI alone binds to dsDNA, but with lower loading efficiency, and displays slower diffusion and shorter dwell times (Extended Data Fig. 2d-f of revised manuscript). As detailed in the response to Referee 1's point 5, we have now investigated the binding of FANCI alone to λ DNA containing a defined ssDNA gap. These data show that FANCI alone does not stall at ss-dsDNA junctions (**Response Fig. 6a**). Instead, we observed sparse, slow-diffusing traces of FANCI within the ssDNA gaps, consistent with previous reports of FANCI binding to ssDNA (Longerich et al. 2009; Yuan et al. 2009). These data are now in Extended Data Fig. 6.

6) Based on the cryo-EM structure, the authors created point mutations that minimize the interaction with the ssDNA (Ext. Data Fig. 10). These mutations provide an excellent control experiment to test their role in recognizing ssDNA at the single molecule level.

We have now performed additional experiments to test these mutants in our single-molecule assay, which are detailed in the response to Referee 1's point 1 (**Response Fig. 1-3**). In summary, the conserved basic residues in FANCD2 contribute to efficient stalling of D2-I at ss-dsDNA junctions: Mutation of the basic residues in the KR helix leads to a major reduction in stalling. We have added these data to the manuscript as Fig 5c.

7) Can the author add Ext. Data figures with more examples of time traces for each experimental condition?

We now include Extended Data Fig. 12 with a gallery of representative kymographs for each single experimental condition.

Response letter references

- Alcon, P., S. Shakeel, Z. A. Chen, J. Rappsilber, K. J. Patel, and L. A. Passmore. 2020. 'FANCD2-FANCI is a clamp stabilized on DNA by monoubiquitination of FANCD2 during DNA repair', *Nat Struct Mol Biol*, 27: 240-48.
- Belan, O., G. Moore, A. Kaczmarczyk, M. D. Newton, R. Anand, S. J. Boulton, and D. S. Rueda. 2021. 'Generation of versatile ss-dsDNA hybrid substrates for single-molecule analysis', *STAR Protoc*, 2: 100588.
- Budzowska, M., T. G. Graham, A. Sobock, S. Waga, and J. C. Walter. 2015. 'Regulation of the Rev1-pol zeta complex during bypass of a DNA interstrand cross-link', *EMBO J*, 34: 1971-85.
- Cai, T., J. Zhang, J. Deng, Q. Tian, R. Zhang, H. Geng, Y. Liu, K. Zhong, L. Wen, C. Liu, X. Hu, Y. Li, Z. Li, S. Hao, K. Yang, and H. Xu. 2023. 'Single-Molecule Force Spectroscopy of Deoxyribonucleic Acid and Deoxyribonucleic Acid Polymerase Activity Impacted by Alkylating Agents', *J Phys Chem Lett*, 14: 4842-49.
- Castaneda, J. C., M. Schrecker, D. Remus, and R. K. Hite. 2022. 'Mechanisms of loading and release of the 9-1-1 checkpoint clamp', *Nat Struct Mol Biol*, 29: 369-75.
- Crisafulli, F. A., E. C. Cesconetto, E. B. Ramos, and M. S. Rocha. 2012. 'DNA-cisplatin interaction studied with single molecule stretching experiments', *Integr Biol (Camb)*, 4: 568-74.
- Enoiu, M., T. V. Ho, D. T. Long, J. C. Walter, and O. D. Scharer. 2012. 'Construction of plasmids containing site-specific DNA interstrand cross-links for biochemical and cell biological studies', *Methods Mol Biol*, 920: 203-19.
- Garaycochea, J. I., G. P. Crossan, F. Langevin, L. Mulderrig, S. Louzada, F. Yang, G. Guilbaud, N. Park, S. Roerink, S. Nik-Zainal, M. R. Stratton, and K. J. Patel. 2018. 'Alcohol and endogenous aldehydes damage chromosomes and mutate stem cells', *Nature*, 553: 171-77.
- Hou, X. M., X. H. Zhang, K. J. Wei, C. Ji, S. X. Dou, W. C. Wang, M. Li, and P. Y. Wang. 2009. 'Cisplatin induces loop structures and condensation of single DNA molecules', *Nucleic Acids Research*, 37: 1400-10.
- Kaczmarczyk, A. P., A. C. Declais, M. D. Newton, S. J. Boulton, D. M. J. Lilley, and D. S. Rueda. 2022. 'Search and processing of Holliday junctions within long DNA by junction-resolving enzymes', *Nature Communications*, 13: 5921.
- Klein Douwel, D., R. A. Boonen, D. T. Long, A. A. Szypowska, M. Raschle, J. C. Walter, and P. Knipscheer. 2014. 'XPF-ERCC1 acts in Unhooking DNA interstrand crosslinks in cooperation with FANCD2 and FANCP/SLX4', *Molecular Cell*, 54: 460-71.
- Liu, W., P. Polaczek, I. Roubal, Y. Meng, W. C. Choe, M. C. Caron, C. A. Sedgeman, Y. Xi, C. Liu, Q. Wu, L. Zheng, J. Y. Masson, B. Shen, and J. L. Campbell. 2023. 'FANCD2 and RAD51 recombinase directly inhibit DNA2 nuclease at stalled replication forks and FANCD2 acts as a novel RAD51 mediator in strand exchange to promote genome stability', *Nucleic Acids Research*, 51: 9144-65.
- Longerich, S., J. San Filippo, D. Liu, and P. Sung. 2009. 'FANCI binds branched DNA and is monoubiquitinated by UBE2T-FANCL', *Journal of Biological Chemistry*, 284: 23182-6.
- Newton, M. D., B. J. Taylor, R. P. C. Driessen, L. Roos, N. Cvetesic, S. Allyjaun, B. Lenhard, M. E. Cuomo, and D. S. Rueda. 2019. 'DNA stretching induces Cas9 off-target activity', *Nat Struct Mol Biol*, 26: 185-92.
- Raschle, M., P. Knipscheer, M. Enoi, T. Angelov, J. Sun, J. D. Griffith, T. E. Ellenberger, O. D. Scharer, and J. C. Walter. 2008. 'Mechanism of replication-coupled DNA interstrand crosslink repair', *Cell*, 134: 969-80.
- Schrecker, M., J. C. Castaneda, S. Devbhandari, C. Kumar, D. Remus, and R. K. Hite. 2022. 'Multistep loading of a DNA sliding clamp onto DNA by replication factor C', *Elife*, 11.
- Seki, S., M. Ohzeki, A. Uchida, S. Hirano, N. Matsushita, H. Kitao, T. Oda, T. Yamashita, N. Kashihara, A. Tsubahara, M. Takata, and M. Ishiai. 2007. 'A requirement of FancL and FancD2 monoubiquitination in DNA repair', *Genes Cells*, 12: 299-310.
- Yuan, F., J. El Hokayem, W. Zhou, and Y. Zhang. 2009. 'FANCI protein binds to DNA and interacts with FANCD2 to recognize branched structures', *Journal of Biological Chemistry*, 284: 24443-52.
- Zheng, F., R. E. Georgescu, N. Y. Yao, M. E. O'Donnell, and H. Li. 2022. 'DNA is loaded through the 9-1-1 DNA checkpoint clamp in the opposite direction of the PCNA clamp', *Nat Struct Mol Biol*, 29: 376-85.

Reviewer Reports on the First Revision:

Referees' comments:

Referee #1:

The authors have addressed all my concerns and suggestions. I am happy to support publication of this manuscript.

Minor comments:

1. It seems to me that replacing 'single-stranded gaps' with 'single-/double-stranded DNA junctions' in the title would be a better representation of the data.

2. Given that the authors did not investigate whether KR-helix mutant variants affect fork protection, I suggest to remove this claim from the last sentence of the abstract.

Referee #2:

In my opinion, the authors have fully addressed each of the questions posed by the independent reviewers, providing a comprehensive set of additional supporting experiments.

I would however, ask that the authors check their calculations of binding affinity as determined by SwitchSENSE, as these indicate extremely tight binding (low nM) — values which are a little at odds with the previously supplied EMSA data.

Referee #3:

I would like to thank the authors for addressing all my comments in details. The newly performed analysis and experiments significantly strengthen the conclusions in this landmark study. I am pleased to endorse the publication of this excellent work in Nature.

Author Rebuttals to First Revision:

Response to Reviewer's Comment

Manuscript 2023-08-14076A

Reviewer's Comment: I would however, ask that the authors check their calculations of binding affinity as determined by SwitchSENSE, as these indicate extremely tight binding (low nM) — values which are a little at odds with the previously supplied EMSA data.

Our response: We have consistently observed higher affinities using SwitchSENSE compared to EMSAs and we addressed this in a previous publication (Sijacki et al., *Nat Struct Mol Biol*, 2022). There are several possible reasons for the discrepancies in affinity. First, while EMSAs could be thought of as a solution-based method, SwitchSENSE is surface-based. Second, because the readout of an EMSA is obtained by performing electrophoresis, it does not reflect true equilibrium binding. Thus, the different apparent Kds could be due to intrinsic differences in the methods. We prefer to use EMSAs as qualitative assays and rely on SwitchSENSE to derive kinetic parameters and obtain dissociation constants.

In the Sijacki manuscript, we used a SwitchSENSE DNA with a stem loop to prevent D2-I from sliding off the end of the DNA, but we did not do that here. Nevertheless, one end of the DNA is always tethered to the surface in SwitchSENSE and this would prevent D2-I from sliding off that end of the DNA. We included the following statement in the published version of the previous manuscript:

The higher affinities measured on SwitchSENSE compared to EMSA could be due to differences in the DNA substrate: the DNA used in EMSA had free DNA ends whereas the ends were blocked in SwitchSENSE to prevent D2-I from sliding off².

Reference:

Sijacki T, Alcón P, Chen ZA, McLaughlin SH, Shakeel S, Rappsilber J, Passmore LA. (2022) The DNA-damage kinase ATR activates the FANCD2-FANCI clamp by priming it for ubiquitination. *Nat Struct Mol Biol* 29, 881–890